# LieDynNet: Learning Lie Symmetries from Spatiotemporal Data

## Abstract

Continuous symmetries of dynamical systems—transformations that map solution trajectories or spatiotemporal fields to new, valid solutions—are powerful tools for analysis, reduction, and control. Prior work on symmetry discovery broadly falls into two categories: methods that prioritize Lie-algebraic structure but operate on static datasets rather than dynamical systems, and methods that discover symmetries for dynamical systems but often do not enforce algebraic structure. Across both threads, most approaches also neglect the infinitesimal invariance condition (IIC)—that prolonged generators annihilate the governing equations. To fill this gap, we introduce *LieDynNet*, which learns Lie symmetry generators directly from data by pairing differentiable ODE/PDE surrogates with two families of constraints: *dynamical validity*, enforced both via IIC (via generator prolongations) and under finite flows; and *algebraic soundness*, enforcing closure, antisymmetry, Jacobi identity, and bilinearity so the generators form a Lie algebra. The framework is model-agnostic and applies to both ODEs and PDEs without hand-crafted priors. On canonical dynamical systems, LieDynNet recovers symmetry algebras and associated invariants from data, showing that learned symmetries can be simultaneously algebraically consistent and dynamically faithful. These results provide a practical, data-driven route to uncovering the symmetry structure of complex dynamical phenomena. Our codes are available as part of the supplementary material.

## 1 Introduction

Symmetries of *dynamical systems*—transformations that carry solution trajectories or spatiotemporal fields to other valid solutions—are a cornerstone of analysis, reduction, and control (Gross, 1995; Gross and Wilczek, 1973). They reveal invariants, constrain qualitative behaviors, and organize families of solutions (Alet et al., 2021; Greydanus et al., 2019). Yet, outside textbook models, the relevant symmetry structure is rarely known a priori and is difficult to recover directly from raw data. This creates a gap between the central role of symmetry in theory and its practical use for learned models of real systems. In this paper, we address the problem of discovering the Lie symmetry structure of an unknown ODE or PDE directly from raw data without imposing any priors. The problem of learning symmetries for dynamical systems is different from learning static invariances (e.g., image rotations), in that these transformations must additionally preserve the underlying dynamical structure. This requires two ingredients that are often pursued separately: (i) *algebraic soundness*—the learned transformations should form a Lie algebra with closure, antisymmetry, Jacobi consistency, and bilinearity; and (ii) *dynamical validity*—their flows should map solutions to solutions, at least locally and for finite steps.

**Contributions.** We present *LieDynNet*, a two-stage framework for discovering continuous Lie point symmetries directly from spatiotemporal data. Our approach couples differentiable neural surrogates of ODEs and PDEs with principled invariance and algebraic constraints. Specifically, we (i) formulate symmetry discovery as learning infinitesimal generators whose flows preserve the surrogate dynamics while forming a valid Lie algebra; (ii) propose a completely *prior-free*, *unsupervised* method that learns continuous symmetries without templates, canonical coordinates, or physics priors; (iii) introduce a practical objective that enforces both infinitesimal and finite-flow invariance, together with closure, antisymmetry, Jacobi consistency, bilinearity and functional independence, enabling recovery of algebraically sound and dynamically valid symmetries; (iv) automatically recover the jet order by sweeping prolongation order and selecting the minimizer of the infinitesimal invariance loss $L_{\text{inv}}$; and (v) identify the true Lie-algebra dimension by increasing the number of generators and selecting the first nontrivial minimum of the closure loss $L_{\text{clo}}$, then validating alignment via

principal-angle comparisons. On canonical benchmarks—including free particle, simple harmonic oscillator, Van der Pol oscillator, Lotka–Volterra system, and viscous Burgers equation—LieDynNet recovers Lie algebras and associated invariants, selects the correct jet order and algebra dimension, and maintains dynamical validity under finite flows. Ablation studies further confirm the necessity of combining algebraic soundness with dynamical consistency.

**Related work.** Prior research on symmetry in machine learning has followed two complementary paths. The first emphasizes *equivariant architectures*, where symmetry is built into the network design. Examples include group-equivariant CNNs for images (Cohen and Welling, 2016) and the general theory of equivariant CNNs on homogeneous spaces (Cohen et al., 2019), spherical CNNs on $SO(3)$ (Kondor et al., 2018), Tensor Field Networks and SE(3)-Transformers in 3D (Thomas et al., 2018; Fuchs et al., 2020), and EGNNs for higher-dimensional settings (Satorras et al., 2021). In physics, gauge-equivariant flows have been developed for efficient sampling in lattice gauge theories with $U(1)$ and $SU(N)$ symmetry (Boyda et al., 2021; Kanwar et al., 2020). See (Wang and Yu, 2021) for a survey of physics-guided deep learning for dynamical systems. A second line of work aims at *symmetry discovery*, ranging from Liu & Tegmark's hidden-symmetry framework (Liu and Tegmark, 2022) to classical Lie group methods of Olver (Olver, 1993).Relatedly, symmetry has been used to guide governing-equation discovery (Yang et al., 2024). Our work builds on both traditions by learning generators directly from trajectories, enforcing both invariance and closure without hard-coding the symmetry group.

Early efforts in symmetry discovery have largely taken two directions. One stresses **Lie algebraic structure**, ensuring that learned generators close under the Lie bracket. For example, Forestano et al. proposed deep-learning methods that recover continuous groups by explicitly enforcing closure, yielding a valid Lie algebra basis from data (Forestano et al., 2023). Similarly, LaLiGAN and its latent-space extension constrain learned transformations to lie within general linear groups, producing interpretable generators (Yang et al., 2023a). These approaches excel at preserving algebraic consistency but generally lack mechanisms to guarantee that the discovered symmetries map solutions of a dynamical system to other valid solutions.

A second direction stresses **flow-based invariance**. Here, candidate generators are validated by integrating their flows and checking whether transformed trajectories remain valid solutions. Ko et al. pursue this perspective by parameterizing infinitesimal generators as vector fields integrated via Neural ODEs, with losses penalizing deviations from invariance (Ko et al., 2024). LieGAN likewise frames symmetry discovery as a generative adversarial task, with a discriminator enforcing that transformed samples are indistinguishable from the original distribution (Yang et al., 2023b). These methods provide strong dynamical guarantees but do not enforce closure of the learned generators.

| Paper | Generator type | | Lie algebra? | IIC? | Dynamical System? |
|---|---|---|---|---|---|
| Ko et al. (Ko et al., 2024) | VF ■ | M □ | □ | □ | ■ |
| Forestano et al. (Forestano et al., 2023) | VF □ | M ■ | ■ | □ | □ |
| LieGAN (Yang et al., 2023b) | VF □ | M ■ | ■ | □ | □ |
| Augerino (Benton et al., 2020) | VF □ | M ■ | □ | □ | □ |
| LaLiGAN (Yang et al., 2023a) | VF □ | M ■ | ■ | □ | □ |
| Shaw et al. (Shaw et al., 2024) | VF ■ | M □ | □ | □ | □ |
| LieDynNet (Ours) | VF ■ | M □ | ■ | ■ | ■ |

Table 1: **Comparison of LieDynNet with related symmetry discovery works.** Here, ■ denotes yes and □ denotes no. "VF" = vector field generator, "M" = matrix/linear parameterization. Columns indicate whether methods use a Lie algebra structure, IIC, or dynamical system validation.

LieDynNet bridges these two threads: It learns infinitesimal generators directly from trajectory data while simultaneously enforcing (i) *Lie bracket closure and Lie algebra axioms consistency*, so that the generators form a valid Lie algebra, and (ii) *invariance losses* at both infinitesimal and finite-flow levels, ensuring preservation of the dynamics. By combining the algebraic guarantees of Lie-algebra–based approaches with the dynamical guarantees of flow-based approaches, LieDynNet recovers full Lie group structure from data, advancing beyond the limitations of earlier frameworks. A further advantage is that LieDynNet is truly prior-free: our training never accesses any physical priors of the governing equation behind the dynamical process. We learn purely from trajectories via

algebraic closure and solution-preservation tests, so the same recipe applies even when the dynamics are unknown.

## 2 MATHEMATICAL PRELIMINARIES

In this section, we present the basic definitions of relevant concepts of a one-parameter Lie group. We treat each symmetry transformation as a (local) group action on the variables of a dynamical system. A *Lie point symmetry* of a differential equation is a local Lie group $G$ acting smoothly on the space of independent and dependent variables $(x, u) \mapsto (x', u')$ such that the action maps solutions to solutions. Infinitesimally, each one–parameter ($\epsilon$) subgroup $\{\exp(\epsilon v)\}_\epsilon \subset G$ is generated by a vector field (the *generator*):

$$v = \sum_{\mu=1}^{p} \xi^\mu(x, u) \, \partial_{x^\mu} + \sum_{a=1}^{q} \phi^a(x, u) \, \partial_{u^a}.$$

for $p$ independent variables and $q$ dependent variables with $\mu, a$ being the indices. Given two generators $v_i, v_j$, their *Lie bracket* $[v_i, v_j] := v_i v_j - v_j v_i$ is again a vector field and measures the first non-commutative correction to composing the two flows. A finite family $\{v_1, \ldots, v_m\}$ spans a (finite-dimensional) *Lie algebra* $\mathfrak{g}$ if it is closed under the bracket and the bracket is bilinear, antisymmetric, and satisfies Jacobi identity:

$$[v_i, v_j] = -[v_j, v_i], \quad [v_i, [v_j, v_k]] + [v_j, [v_k, v_i]] + [v_k, [v_i, v_j]] = 0.$$

Equivalently, there exist constants $c_{ij}{}^k$ (*structure constants*) with $[v_i, v_j] = \sum_{k=1}^{m} c_{ij}{}^k v_k$. A generator and a group action is connected through the *exponential map*, which sends each Lie generator $v \in \mathfrak{g}$ to a group action on states: a one–parameter subgroup $\{\exp(\epsilon v)\}_{\epsilon \in \mathbb{R}} \subset G$ acting on a point (a "state") $z = (x, u)$ (where $x$ denotes the independent variables and $u$ denotes the dependent variables) by $\exp(\epsilon v) \cdot z = \Phi_v^\epsilon(z)$, where the flow $\Phi_v^\epsilon$ solves:

$$\frac{d}{d\epsilon} z(\epsilon) = v\big(z(\epsilon)\big), \qquad z(0) = z_0.$$

A *prolongation* extends a vector field from $(x, u)$ to the *jet space*—the space of variables and their derivatives up to a fixed order $n$, known as the *jet order* (with jet order chosen to match the highest derivative order appearing in the equations as written). The $n$-jet $J^n$ has coordinates $\big(x, u, \{u_J^a\}_{|J| \leq n}\big)$. The prolonged vector field $\mathrm{pr}^{(n)} v$ is the unique lift that acts on these coordinates via the chain rule. To demonstrate how the prolongation is calculated, we use a second-order ODE with one independent variable and one dependent variable as an example (since they are the most relevant orders in this paper; refer to *Theorem 2.36* (Olver, 1993) for the general prolongation formula). For a single second-order equation $F(t, x(t), \dot{x}(t), \ddot{x}(t)) = 0$ with a point–symmetry generator $v = \xi(t, x) \, \partial_t + \phi(t, x) \, \partial_x$. and total derivative $D_t = \partial_t + \dot{x} \, \partial_x + \ddot{x} \, \partial_{\dot{x}}$, the first and second prolongations are given by

$$\mathrm{pr}^{(1)} v = \xi \, \partial_t + \phi \, \partial_x + \phi^{(1)} \, \partial_{\dot{x}}, \quad \phi^{(1)} = D_t \phi - \dot{x} \, D_t \xi$$

$$\mathrm{pr}^{(2)} v = \mathrm{pr}^{(1)} v + \phi^{(2)} \, \partial_{\ddot{x}}, \quad \phi^{(2)} = D_t \phi^{(1)} - \ddot{x} \, D_t \xi$$

**Lemma 2.1** (*Infinitesimal Invariance Condition (IIC) cf. Theorem 2.31 (Olver, 1993)*). *Suppose* $\Delta_\nu(x, u^{(n)}) = 0$, $\nu = 1, \ldots, l$ *is a system of differential equations of maximal rank defined over* $M \subset X \times U$. *If $G$ is a local group of transformations acting on $M$, and:*

$$pr^{(n)} v[\Delta_\nu(x, u^{(n)})] = 0, \ \nu = 1, \ldots, l, \ \text{whenever} \ \Delta(x, u^{(n)}) = 0,$$

*for every infinitesimal generator $v$ of $G$, then $G$ is a symmetry group of the system.*

Here, $X \subset \mathbb{R}^p$ is the open "independent-variable" manifold with coordinates $x = (x^1, \ldots, x^p)$, $U \subset \mathbb{R}^q$ is the open set of "dependent-variable" values $u = (u^1, \ldots, u^q)$, and $M \subset X \times U$ is the configuration (state) manifold of admissible pairs $(x, u)$ on which the point transformations and equations are defined. Therefore, for the second-order ODE example we mentioned above, the equation is invariant under $v$ if and only if $\mathrm{pr}^{(2)} v\big(F(t, x, \dot{x}, \ddot{x})\big) = 0$ whenever $F = 0$.

## 3 LIEDYNNET FRAMEWORK

**Goal.** Our aim is to learn a set of infinitesimal generators (vector fields) that span the Lie algebra of a system's symmetries. Exponentiating and composing these generators yields the full connected Lie group: any continuous symmetry in the identity component can be expressed as a product of exponentials, $\exp(\epsilon_1 v_{i_1}) \cdots \exp(\epsilon_r v_{i_r})$. A key challenge is ensuring that the learned transformations form a *genuine* symmetry group rather than a loose collection of local invariances. Infinitesimal checks such as the invariance condition $\mathrm{pr}^{(n)} v[\Delta]\big|_{\Delta=0} = 0$ certify only that dynamics are preserved to first order. Without algebraic constraints, finite flows may drift, compositions may generate directions outside the span of learned generators, and the resulting set is not closed under group operations. Enforcing closure, antisymmetry, and the Jacobi identity upgrades local invariance into a consistent Lie algebra, which integrates via the exponential and Baker–Campbell–Hausdorff correspondence into a connected symmetry group. This guarantees a stable composition law, a well-defined parameterization, and interpretable generators. In practice, this distinction is crucial: conservation laws in physical and biophysical systems are indexed by the Lie algebra of continuous symmetries, and inconsistent algebraic structure leads to spurious or contradictory invariants. By enforcing both dynamical validity and algebraic soundness, LieDynNet ensures that the recovered generators correspond to true symmetries rather than approximate invariances, laying the foundation for the workflow described next.

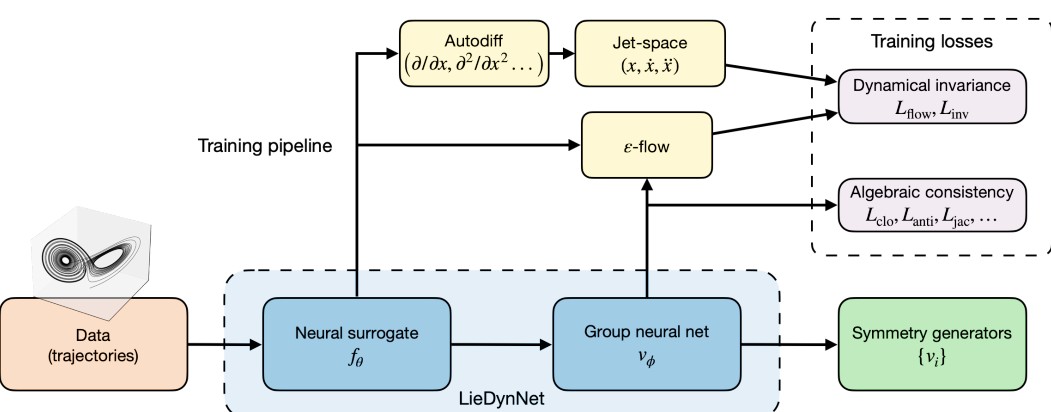

Figure 1: **Architecture and training pipeline of LieDynNet.** Trajectory data are first modeled by a neural surrogate $f_\theta$, which supplies jet-space samples through automatic differentiation $(\partial/\partial t, \partial/\partial x, \dots)$. Candidate infinitesimal generators $\{v_i\}$ are produced by the group neural network and trained under two classes of objectives: *dynamical invariance*, which enforces both the infinitesimal invariance condition $L_{\mathrm{inv}}$ and flow-based validity $L_{\mathrm{flow}}$, and *algebraic consistency*, which penalizes violations of closure, antisymmetry, bilinearity, and Jacobi relations ($L_{\mathrm{clo}}, L_{\mathrm{anti}}, L_{\mathrm{jac}}, \dots$). Together, these losses ensure that the learned generators form a Lie algebra and map trajectories to trajectories, recovering full symmetry structure. Validation proceeds via the principal-angle comparison between generators, in which the alignment between the learned and the ground-truth Lie algebras is measured.

**Learning objectives.** We parameterize each generator in the current presentation for scalar ODEs with one independent variable and one dependent variable as

$$v_k = \xi_k(t, u)\, \partial_t + \phi_k(t, u)\, \partial_u \quad \text{(shorthand } v_k = (\xi_k, \phi_k)\text{)}$$

for $k \in \{1, \dots, m\}$, $m \in \mathbb{N}^+$. For PDEs or ODEs with multiple dependent variables, we adopt the standard multi-component point-symmetry generator form:

$$v_k = \sum_{\mu=1}^{p} \xi_k^\mu(x, u)\, \partial_{x^\mu} \ + \ \sum_{a=1}^{q} \phi_k^a(x, u)\, \partial_{u^a},$$

with $p$ independent variables $x = (x^1, \dots, x^p)$ and $q$ dependent variables $u = (u^1, \dots, u^q)$. We then optimize a set of complementary losses described below. Our dynamics-consistency losses follow

the general spirit of dynamical invariance objectives used in prior symmetry-discovery work such as (Ko et al., 2024; Hu et al., 2025). Our algebra-structure losses are inspired by recent approaches that enforce Lie-algebraic structure in learned representations (Forestano et al., 2023; Yang et al., 2023a;b).

**Dynamics-consistency losses.**

*1. Infinitesimal invariance (prolongation).* On-shell, for a differential equation with order $a$, we penalize violation of the infinitesimal invariance condition:

$$L_{\text{inv}} = \mathbb{E}\left[\frac{1}{m}\sum_{i=1}^{m}\left\|\text{pr}^{(a)}v_i(f_\theta)\right\|^2\right].$$

*2. Flow-based validity (small $\epsilon$).* Let $\Phi_\epsilon^{(i)}$ denote the pseudo-time $\epsilon$-flow of $v_i$. Advance a small step along each generator's flow and require that transformed trajectories remain solutions in the data-supported sense:

$$L_{\text{flow}} = \mathbb{E}\left[\frac{1}{m}\sum_{i=1}^{m}\left\|f_\theta \circ \Phi_\epsilon^{(i)}\right\|^2\right].$$

**Algebra-structure losses.**

*3. Closure.* We project Lie brackets onto the span of generators and penalize the residual:

$$L_{\text{clo}} = \mathbb{E}\left[\sum_{i<j}\left\|[v_i, v_j] - \sum_{k=1}^{m}c_{ij}^k v_k\right\|^2\right].$$

*4. Constancy.* We encourage structure coefficients to be sample-independent:

$$L_{\text{const}} = \mathbb{E}\left[\sum_{i,j,k}\left\|c_{ij}^k - \bar{c}_{ij}^k\right\|^2\right].$$

*5. Independence.* We stack generator evaluations over samples to form a Gram matrix $G$ and regularize its spectrum using $\tau$ as a floor and $\beta$ as a weight ($\lambda_{\max}, \lambda_{\min}$ respectively are the maximum and minimum eigenvalues):

$$L_{\text{ind}} = \left[\max(0, \tau - \lambda_{\min})\right]^2 + \beta\left[\log(\lambda_{\max}/\lambda_{\min})\right]^2.$$

*6. Antisymmetry.* We enforce $c_{ij}^k = -c_{ji}^k$ via

$$L_{\text{anti}} = \sum_{i,j,k}\left(c_{ij}^k + c_{ji}^k\right)^2.$$

*7. Jacobi identity.* We penalize violations of $\sum_m(c_{ij}^m c_{mk}^\ell + c_{jk}^m c_{mi}^\ell + c_{ki}^m c_{mj}^\ell) = 0$:

$$L_{\text{jac}} = \mathbb{E}\left[\sum_{i,j,k,\ell}\left(\sum_m c_{ij}^m c_{mk}^\ell + c_{jk}^m c_{mi}^\ell + c_{ki}^m c_{mj}^\ell\right)^2\right].$$

*8. Bilinearity.* We check linearity in each slot on random triples and random coefficients:

$$r_1(i, j, k; c, c') := [cv_i + c'v_j, v_k] - c[v_i, v_k] - c'[v_j, v_k],$$
$$r_2(i, j, k; c, c') := [v_i, cv_j + c'v_k] - c[v_i, v_j] - c'[v_i, v_k].$$

With $(i, j, k)$ sampled uniformly without replacement from $\{1, \ldots, m\}$ (all distinct) and $(c, c') \sim \mathcal{C} \subset [-1, 1]^2$,

$$L_{\text{bilin}} = \mathbb{E}_{(i,j,k),(c,c')}\left[\|r_1(i, j, k; c, c')\|_1 + \|r_2(i, j, k; c, c')\|_1\right].$$

The total objective is:

$$L = w_{\text{inv}}L_{\text{inv}} + w_{\text{flow}}L_{\text{flow}} + w_{\text{ind}}L_{\text{ind}} + w_{\text{clo}}L_{\text{clo}} + w_{\text{const}}L_{\text{const}} + w_{\text{anti}}L_{\text{anti}} + w_{\text{jac}}L_{\text{jac}} + w_{\text{bilin}}L_{\text{bilin}}.$$

---

**Algorithm: LieDynNet Workflow** (*details provided in Appendix A*)

---

**Input:** Trajectories / spatiotemporal samples $\mathcal{D}$; generator-count schedule $m = 1, 2, \ldots$
**Output:** Jet order $\hat{n}$, algebra dimension $\hat{m}$, generators $\{v_i\}_{i=1}^{\hat{m}}$, diagnostics

1: **Surrogate & jet order.** Train a differentiable surrogate $f_\theta$ on $\mathcal{D}$. For $n = 1, 2, \ldots$, train with a small $m$ and record post-training $L_{\text{inv}}(n)$. Set $\hat{n} \leftarrow \arg\min_n L_{\text{inv}}(n)$ and freeze $\hat{n}$.
2: **Jet-space assembly.** From $f_\theta$, form $J^{\hat{n}}(X, U)$ (states/fields and all derivatives up to order $\hat{n}$) to evaluate $\text{pr}^{(\hat{n})}$ for $L_{\text{inv}}$ and to draw samples for finite-flow checks $L_{\text{flow}}$.
3: **Generator learning (fixed $m$).** Parameterize $v_i$ for $i = 1, \ldots, m$. Optimize the composite objective: $w_{\text{inv}}L_{\text{inv}} + w_{\text{flow}}L_{\text{flow}} + w_{\text{ind}}L_{\text{ind}} + w_{\text{clo}}L_{\text{clo}} + w_{\text{const}}L_{\text{const}} + w_{\text{anti}}L_{\text{anti}} + w_{\text{jac}}L_{\text{jac}} + w_{\text{bilin}}L_{\text{bilin}}$.
4: **Dimension selection.** Increase $m$ and repeat Step 3; choose $\hat{m}$ at the first nontrivial minimum of $L_{\text{clo}}(m)$.
5: **Report.** At $(\hat{n}, \hat{m})$, compute principal-angle alignment/cosine similarity with analytic spans (when available), evaluate/plot after-flow residuals, and release $\{v_i\}$.

---

**Hyperparameter selection.** There are two hyperparameters that require careful selection: jet order $n$ (which is also the order of the dynamical system) and Lie algebra dimension $m$ (equivalent to the number of generators). We first recover the jet order by running the training while iterating the jet order starting from 1 while fixing the dimension. The jet order at which the training eventually yields the smallest IIC loss $L_{\text{inv}}$ is chosen and fixed. Since we work with Lie point symmetry, the dimension $m$ is constrained by bounds in classical Lie theory. For scalar ODEs of order $n \geq 3$,

$$\dim \mathfrak{sym} \leq n + 4,$$

where $\mathfrak{sym}$ denotes the Lie algebra of point symmetries of the system, with equality attained by the flat model $x^{(n)} = 0$. For second-order scalar ODEs, $\dim \mathfrak{sym} \leq 8$. These bounds guide the maximum $m$ we attempt. In practice we start with $m = 1$, grow it up to the bound (or a user cap), and stop when the Lie bracket closure loss $L_{\text{clo}}$ is minimized. For PDEs, bounds are equation-class dependent thus have no universal bounds on dimension; in practice, we adopt the same progressive strategy with validation. With $n$ and $m$ thus identified, we then run the symmetry-discovery training pipeline to recover the generators. (Additional details on hyperparameter selection are provided in Appendix A.)

## 4 EXPERIMENTAL SETUP AND RESULTS

**Setup: systems and neural surrogates** We discover symmetries from LieDynNet utilizing spatiotemporal data from five canonical equations — two second-order ODEs (free particle/FP and simple harmonic oscillator/HO), a nonlinear oscillator (Van der Pol/VdP), a two-species model (Lotka–Volterra/LV), and a 1D viscous Burgers PDE. In every case, we first learn a neural surrogate of the dynamics from synthetically generated, noisy observations, and then train the symmetry generators by enforcing the loss terms described above using that learned surrogate. Training data for the neural surrogates are synthesized from the analytic equations, after which the learning is entirely data-driven; we do not impose any hand-crafted structure, coordinates, or group forms during surrogate training.

| System | Sampling / size | Noise $\sigma$ |
|---|---|---|
| Free Particle | $64 \times 200$ (trajectories $\times$ time steps) | 0.03 |
| Simple Harmonic Oscillator | $64 \times 200$ (trajectories $\times$ time steps) | 0.05 |
| Van der Pol | $64 \times 300$ (trajectories $\times$ time steps) | 0.03 |
| Lotka–Volterra | train $128 \times 801$;  val $32 \times 801$ (trajectories $\times$ time steps) | 0.03 |
| Viscous Burgers (1D) | $256 \times 201$ (space points $\times$ time steps) | 0.03 |

Table 2: **Systems, data, and sampling for neural surrogates.** Van der Pol: RK4 sub-steps $= 5$ per data interval. Lotka–Volterra: windowed training with *30-step* segments ("multi-shooting": train on many short rollout segments, each initialized at the observed state, to reduce long-horizon error compounding and stabilize gradients). Burgers: at each optimization step, we draw $8,192$ random space–time points from the $256 \times 201$ grid to compute the loss.

**Methods of evaluation.**     To interpret our results, we first justify the three diagnostics we report.

**(i) Algebra alignment via principal angles.** Because a Lie algebra is a vector space, any invertible linear recombination (and rescaling) of a valid generating set yields an equally valid basis; there is no unique "canonical" choice of generators, and the structure constants change under such basis changes. Element-wise matching of generators is therefore ill-posed, so we compare *algebra spaces* instead. Let $\{v_i\}_{i=1}^m$ be the learned generators and $\{w_j\}_{j=1}^r$ the analytic generators, with

$$v_i(t,x) = \xi_i(t,x)\,\partial_t + \phi_i(t,x)\,\partial_x, \qquad w_j(t,x) = \tilde{\xi}_j(t,x)\,\partial_t + \tilde{\phi}_j(t,x)\,\partial_x,$$

illustrated here for an ODE with one independent and one dependent variable, $t$ and $x$. We evaluate all generators on a grid $\Omega = \{(t_r, x_s) : r = 1, \ldots, R;\ s = 1, \ldots, S\}$ (for PDEs such as Burgers, we use a $(t, x, u)$ grid) and form data matrices by stacking the sampled components:

$$B_\ell \in \mathbb{R}^{(2RS) \times m}, \qquad B_g \in \mathbb{R}^{(2RS) \times r},$$

whose $i$-th (resp. $j$-th) column collects the values of $[\xi_i, \phi_i]$ (resp. $[\tilde{\xi}_j, \tilde{\phi}_j]$) on $\Omega$. Using the Euclidean inner product on $\mathbb{R}^{2RS}$, we compute reduced QR factorizations $B_\ell = Q_\ell R_\ell$ and $B_g = Q_g R_g$, and define the principal angles $\{\theta_k\}_{k=1}^d$ between the column spans via

$$\cos \theta_k = \sigma_k\big(Q_g^\top Q_\ell\big), \qquad d = \min\{\text{rank } B_\ell, \text{rank } B_g\}, \quad \theta_k \in [0, \tfrac{\pi}{2}],$$

where $\sigma_k(\cdot)$ are singular values in descending order. Angles near $0°$ indicate close span alignment (all $\theta_k = 0°$ in the ideal case where the sampled spans coincide under this metric). In practice, once the correct dimension is identified ($m = r$ and both matrices are full rank), we have $d = m$ and report the full spectrum (in degrees) together with a scalar summary $\max_k \theta_k$ across candidate dimensions $r$. Detailed practical notes on the principal-angle comparison are provided in Appendix A.

**(ii) Dimension selection via closure loss.** When we fit fewer than the true number of generators, some Lie brackets necessarily leave the learned span, yielding a non-vanishing closure residual that cannot be removed by optimization. As we increase $m$, this closure loss decreases and reaches a minimum precisely when the learned span is rich enough to contain all brackets, i.e., at $m = \dim \mathfrak{sym}$. Beyond this point, adding extra generators does not further reduce the closure loss and is discouraged by the independence and Jacobi penalties, leading to a clear elbow or minimum at the correct algebra dimension.

**(iii) Jet order identification via IIC.** The infinitesimal invariance condition must be enforced at the PDE or ODE's jet order. If we use a prolongation order that is *lower* than the true differential order, key derivatives are omitted and the invariance residual cannot be driven to zero; if we use a *higher* order, we introduce unnecessary derivatives that mainly amplify estimation noise. As a result, the IIC loss attains its minimum at the true jet order, providing a data-driven check of the correct differential order that is consistent with Lie theory.

**Results.**     We describe the results of the experiments below.

Table 3: Principal angles for dynamical systems with $\dim \mathfrak{sym} > 1$ at the ground-truth dimension.

| Systems | Gen 1 | Gen 2 | Gen 3 | Gen 4 | Gen 5 | Gen 6 | Gen 7 | Gen 8 |
|---|---|---|---|---|---|---|---|---|
| FP | $15.390°$ | $7.052°$ | $0.781°$ | $0.260°$ | $0.092°$ | $0.082°$ | $0.005°$ | $0.002°$ |
| HO | $10.044°$ | $7.568°$ | $5.554°$ | $3.419°$ | $1.433°$ | $0.757°$ | $0.328°$ | $0.176°$ |
| 1D Burgers | $21.081°$ | $17.129°$ | $13.778°$ | $7.343°$ | $5.696°$ | — | | |
| 2D Burgers | $19.237°$ | $14.593°$ | $13.642°$ | $6.418°$ | $5.483°$ | $2.539°$ | — | |

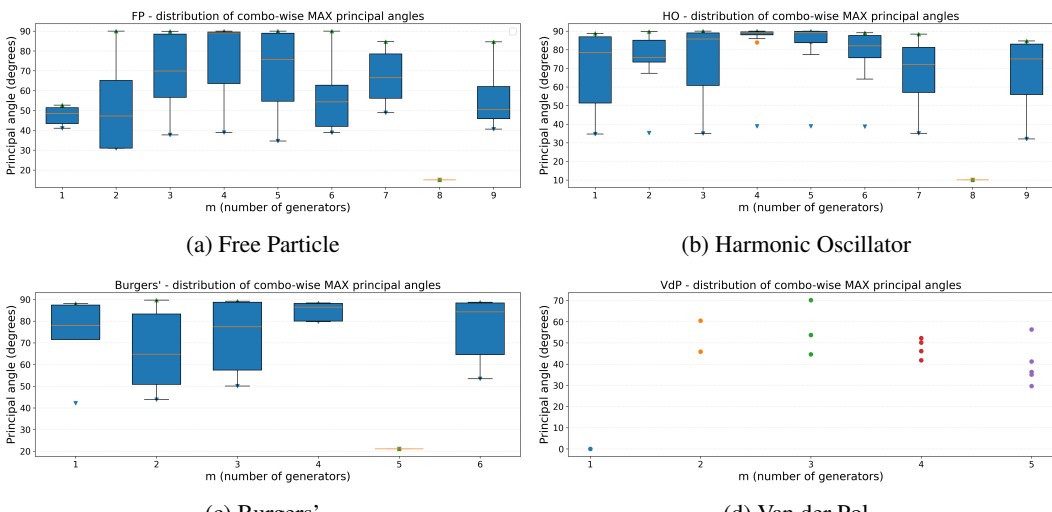

(a) Free Particle

(b) Harmonic Oscillator

(c) Burgers'

(d) Van der Pol

Figure 2: **Distribution of combination-wise maximum principal angles.** From (a) to (c): the blue boxes denote the interquartile range; the orange horizontal lines inside the box denote the median; the green up triangles and the blue down triangles denote the maximum and minimum of MAX principal angles, respectively; the orange dots denote the mean. In (d), the dots denote the principal angle values — it's not presented in a box plot because there are very few principal angles to plot.

Following the workflow above, we first sweep the jet order starting from 1. As shown in Fig. 3, the invariance loss $L_{\text{inv}}$ attains its minimum at jet order 2. Importantly, this minimum is consistently achieved at the same jet order even when the number of generators $m$ is varied. Thus, the correct jet order can be identified without knowing $m$ in advance (it is 2 in our experiments). We then fix the jet order at 2 and sweep the number of generators $m = 1, 2, \ldots$, selecting $m$ where the Lie-bracket closure loss $L_{\text{clo}}$ reaches its first nontrivial minimum. (Note that $m = 1$ yields $L_{\text{clo}} = 0$ by construction and is therefore not informative.) Fig. 4 shows that this minimum occurs at the ground-truth dimension (8 for FP and HO; 5 for Burgers'; 1 for VdP), indicating that LieDynNet recovers the correct Lie algebra dimension.

For each candidate $m$, we quantify alignment with the analytic algebra using principal angles between the corresponding generator spans. Let $r$ be the number of ground-truth generators. If $m \leq r$, we compare the learned $m$-dimensional span to all $\binom{r}{m}$ $m$-element subsets of the ground-truth basis; if $m > r$, we compare all $\binom{m}{r}$ $r$-element subsets of the learned generators to the $r$-dimensional ground-truth span. For each comparison we record the largest principal angle, and for each $m$ we aggregate the distribution of these maximal angles. As shown in Fig. 2, the minimum (over $m$) of the maximal principal angle occurs at the true dimension $r$. At that selected dimension, Table 3 reports the principal angles for systems with $r > 1$, which are uniformly close to zero, confirming that the learned generators reconstruct the ground-truth Lie algebra. Although there is no universal threshold for how small principal angles should be, the fact that the maximal principal angle attains its minimum at the true dimension provides a validity check of the pipeline. To aid interpretation of these angles, we additionally benchmark them against the symmetry-discovery method of Ko et al. (Ko et al., 2024) on the 1D viscous Burgers equation. In this baseline, the maximal principal angle achieved by our method is substantially smaller (on the order of $20°$) than that of Ko et al. (exceeding $80°$), indicating a much closer alignment with the ground-truth symmetry algebra; see Appendix C

(Baseline Comparison) for a detailed comparison of the learned generators. (See Figures 19-20 for results on 2D Burgers' equation in Appendix C.)

In the Van der Pol and Lotka–Volterra experiments, the ground-truth Lie algebra is one-dimensional, spanned by the time-translation field $v^\star = \partial_t$. Accordingly, we evaluate alignment *directly* via a trajectory-averaged cosine similarity between the learned generator and $v^\star$. *(Side note: for illustration, we also report the maximum principal angle for VdP under different $m$ in Fig. 2d).*

For VdP with state $(t, x)$, the learned generator is $v_0 = \xi_0(t, x) \, \partial_t + \phi_0(t, x) \, \partial_x$. Given a solution $\gamma(t) = (t, x(t))$ sampled at $\{t_i\}_{i=1}^N$, the discrete trajectory-averaged cosine is

$$\cos(v^\star, v_0) = \frac{\frac{1}{N} \sum_{i=1}^N \langle (1,0), \, (\xi_0, \phi_0) \rangle |_{(t_i, x(t_i))}}{\sqrt{\frac{1}{N} \sum_{i=1}^N \|(1,0)\|^2} \sqrt{\frac{1}{N} \sum_{i=1}^N \|(\xi_0, \phi_0)\|^2} |_{(t_i, x(t_i))}}$$

$$= \frac{\frac{1}{N} \sum_{i=1}^N \xi_0(t_i, x(t_i))}{\sqrt{\frac{1}{N} \sum_{i=1}^N \left( \xi_0^2 + \phi_0^2 \right)(t_i, x(t_i))}}$$

For LV with state $(t, u, w)$, the learned generator is $v_0 = \tau_0(t, u, w) \, \partial_t + \xi_0(t, u, w) \, \partial_u + \phi_0(t, u, w) \, \partial_w$. Along a trajectory $\gamma(t) = (t, u(t), w(t))$ sampled at $\{t_i\}_{i=1}^N$, the cosine becomes

$$\cos(v^\star, v_0) = \frac{\frac{1}{N} \sum_{i=1}^N \tau_0(t_i, u(t_i), w(t_i))}{\sqrt{\frac{1}{N} \sum_{i=1}^N \left( \tau_0^2 + \xi_0^2 + \phi_0^2 \right)(t_i, u(t_i), w(t_i))}}$$

According to Table 4 below, the cosine similarities are close to 1 across diverse initial conditions, indicating strong alignment with $v^\star$ and validating recovery of time-translation symmetry.

Table 4: **Trajectory-averaged cosine similarities with the ground-truth time-translation generator $v^\star = \partial_t$. Higher is better (closer to 1).**

| System | IC 1 | IC 2 | IC 3 |
|---|---|---|---|
| VdP $(x_0, \dot{x}_0)$ | $(1, 0): \ 0.980$ | $(0.5, 0): \ 0.985$ | $(2, 0): \ 0.974$ |
| LV $(u_0, w_0)$ | $(1.5, 1.0): \ 0.880$ | $(1.2, 0.8): \ 0.950$ | $(2, 0.6): \ 0.930$ |

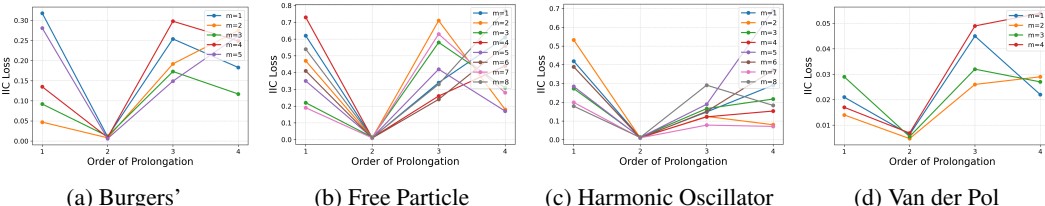

(a) Burgers'  (b) Free Particle  (c) Harmonic Oscillator  (d) Van der Pol

Figure 3: **Post-training IIC loss $L_{\text{inv}}$ across prolongation order $n$.** Colors indicate the number of generators $m$. Varying $m$ shows the loss minimum is consistently attained at the ground-truth jet order (2 in these experiments), rather than being a fluke tied to a particular $m$.

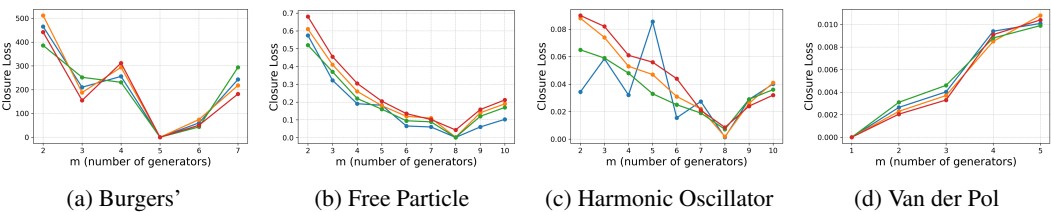

(a) Burgers'  (b) Free Particle  (c) Harmonic Oscillator  (d) Van der Pol

Figure 4: **Post-training Lie bracket closure loss $L_{\text{clo}}$ under different numbers of generators.** Colors denote independent runs with identical settings (different random seeds), showing the minimum is consistently attained at the ground-truth $m$ rather than a one-off.

To visualize representative outcomes, Fig. 15 shows the Van der Pol (VdP) trajectory after applying the single learned generator with $\epsilon = 4$; the transformed trajectory closely preserves the original solution's shape. For the free particle (FP), Fig. 16 displays the after-flow trajectories obtained by pushing the data through each of the eight learned generators; the resulting straight paths confirm that the FP solution structure is preserved. Additional qualitative and quantitative results are provided in Appendix C. Beyond these visuals, ablation studies in Appendix B show that disabling either the infinitesimal invariance term $L_{inv}$ or the finite-flow consistency term $L_{flow}$ consistently worsens alignment (larger maximum principal angles) and increases closure residuals, indicating both are necessary for stable recovery of the algebra. We also compare alternative neural surrogate parameterizations and obtain comparable principal angles and algebraic metrics across forms, supporting the model-agnostic nature of LieDynNet (in Appendix B).

## 5 CONCLUSION

We presented *LieDynNet*, a prior-free, unsupervised framework that learns Lie point symmetries directly from spatiotemporal data by coupling differentiable ODE/PDE surrogates with principled invariance and algebraic constraints. Concretely, we train neural surrogates and learn generators that (i) satisfy the infinitesimal invariance condition via prolongations evaluated on the surrogate residual, (ii) preserve dynamics under small finite flows, and (iii) obey Lie-algebra structure—closure with constant structure coefficients, antisymmetry, Jacobi, bilinearity, and functional independence. The pipeline also identifies the correct jet order (via the minimum of $L_{inv}$) and the algebra dimension (via the first nontrivial minimum of $L_{clo}$). Across canonical benchmarks—free particle and harmonic oscillator (rich 8-D algebras), Van der Pol and Lotka–Volterra (time translation), and viscous Burgers (multiple generators)—the learned spans align with analytic algebras via principal-angle diagnostics while maintaining finite-flow validity.

Overall, LieDynNet offers a practical, equation-agnostic path from raw data to usable symmetry structure for dynamical systems. Currently, our framework only includes *point* symmetries and the identity-connected component, which is the main limitation. Future work involves extending the framework to generalized or nonlocal symmetries that can be built on this presented pipeline.

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

# A   PRACTICAL NOTES ON TRAINING AND EVALUATION

**Workflow.**   Given trajectories or spatiotemporal samples, LieDynNet proceeds in three stages.

*(1) Learn a differentiable surrogate of the dynamics and pick the jet order.* For ODEs and PDEs, we fit a neural surrogate in any form on the given spatiotemporal data to recover the structure of the underlying differential equation of the data. For the purpose of identifying the true jet order, we fix a small non-negative number of generators, then run the training and iterate through each jet order starting from one until a clear minimum of infinitesimal invariance loss ($L_{\mathrm{inv}}$, explained later in the text) is identified. We then freeze the identified jet order $\hat{n}$ at which the minimum loss occurs.

*(2) Assemble jet-space samples from the surrogate.* From the trained neural surrogates $f_\theta$, we draw a grid of states/fields and their derivatives up to order $\hat{n}$. Concretely, we evaluate the on-shell (evaluated on the solution manifold of the learned surrogate—i.e., at states/fields where the surrogate ODE/PDE is satisfied and the governing-equation residual vanishes. ) after-flow residual $L_{\mathrm{flow}}$ together with the derivatives needed by $\mathrm{pr}^{(\hat{k})}$ for $L_{\mathrm{inv}}$ using the jet-space samples. Here, the $\hat{n}$-*jet space* $J^{\hat{n}}(X, U)$ is the set of tuples $(x, u^{(\hat{n})})$ collecting all partial derivatives of $u$ up to order $\hat{n}$. These jet-space tuples supply all inputs required to compute the symmetry losses.

*(3) Learn generator vector fields under dynamical invariance and algebraic structure.* We parameterize $m$ candidate infinitesimal generators $v_i$ (with multi-index components for fields). Starting from a small $m$, we optimize the generator network using a composite objective that (i) enforces IIC ($L_{\mathrm{inv}}$); (ii) checks finite-flow validity $L_{\mathrm{flow}}$ by pushing samples along $\epsilon$-flows of $v_i$ and re-evaluating $f_\theta$ to assure they still lie inside the solution space; (iii) imposes Lie-algebra structure via closure/antisymmetry/Jacobi/bilinearity terms ($L_{\mathrm{clo}}, L_{\mathrm{anti}}, L_{\mathrm{jac}}, L_{\mathrm{bilin}}$) with a constancy penalty on $c_{ij}^k$ ($L_{\mathrm{const}}$); and (iv) maintains functional independence through a stacked Gram penalty $L_{\mathrm{ind}}$. We gradually increase $m$ until the minimum Lie bracket closure loss ($L_{\mathrm{clo}}$) is identified (which marks the current number of generators $\hat{m}$ is the correct Lie algebra dimension), never exceeding classical bounds (e.g., $m \le n + 4$ for scalar $n$th-order ODEs; $m \le 8$ for second order).

**Finite-$\epsilon$ validity under numerical integration.**   Mathematically, a true Lie symmetry should map solutions to solutions for any $\epsilon$ in its one-parameter group. In practice, however, our network only enforces invariance over the finite $\epsilon$ values used during training. Numerical integration errors accumulate as $\epsilon$ grows, so the discovered generators behave as approximate symmetries that are valid only within a user-chosen $\epsilon$ range. This behavior is expected and not specific to our method: any discrete-time numerical integration of continuous flows incurs truncation and round-off errors that grow with step size and horizon.

**Inferring the Lie–algebra dimension from the closure–loss curve.**   Let $L_{\mathrm{clo}}(m)$ denote the post-training Lie–bracket closure loss when the model is instantiated with $m$ generators. Note that $L_{\mathrm{clo}}(1) = 0$ *by definition*: with a single generator $v_1$, the only bracket is $[v_1, v_1] = 0$, hence the algebra is vacuously closed. Therefore the point $m = 1$ is uninformative for dimension selection and should be excluded from the search for a nontrivial minimum.

Empirically, when the true dimension $m^\star > 1$, the curve $m \mapsto L_{\mathrm{clo}}(m)$ (for $m \ge 2$) is U-shaped with a unique interior minimum at $m^\star$, i.e.

$$L_{\mathrm{clo}}(m^\star) = \min_{m \ge 2} L_{\mathrm{clo}}(m), \qquad L_{\mathrm{clo}}(m) > L_{\mathrm{clo}}(m^\star) \text{ for all } m \ne m^\star,$$

so the correct dimension is read off as the argmin over $m \ge 2$. In contrast, when $m^\star = 1$, the curve exhibits a *right-hand* increase: $L_{\mathrm{clo}}(1) = 0$ while $\min_{m \ge 2} L_{\mathrm{clo}}(m)$ remains strictly positive (up to numerical tolerance), and typically grows with $m$.

A practical decision rule is

$$\hat{m} = \begin{cases} 1, & \text{if } L_{\mathrm{clo}}(1) = 0 \text{ and } \min_{m \ge 2} L_{\mathrm{clo}}(m) > \varepsilon, \\ \arg\min_{m \ge 2} L_{\mathrm{clo}}(m), & \text{otherwise,} \end{cases}$$

with a small tolerance $\varepsilon$ to absorb numerical noise. The only potentially ambiguous edge case is $m^\star = 2$: when $m^\star \in \{1, 2\}$, the closure–loss curve over $m \ge 2$ shows the same right–hand increase

(with $L_{\text{clo}}(1) = 0$), so the shape alone cannot distinguish $m^\star = 2$ from $m^\star = 1$. In that case we resolve by an auxiliary check: prefer $m = 2$ if the learned 2-generator model exhibits stable, nonzero structure constants (non-commuting brackets) and/or improved consistency/fit losses relative to $m = 1$; otherwise prefer $m = 1$.

**Neural surrogates and prolongation-based loss computation.** Across all ODEs, the surrogate is first-order in time (a neural vector field on state), trained via differentiable RK4. For Burgers, the surrogate remains first-order in time while using second-order spatial derivatives through autodiff; in all cases, the learning is *model-agnostic* and purely data-driven from the analytically generated datasets. For FP, HO, and VdP, the neural dynamics can be written as $[\dot{x}(t) \quad \dot{v}(t)]^\top = f_\theta(x(t), \dot{x}(t))$ for $v(t) \equiv \dot{x}(t)$ with one independent variable $t$ and two dependent variables $x, v$ [1], whereas LV can be written as $[\dot{u}(t) \quad \dot{w}(t)]^\top = g_\theta(u(t), w(t))$ with one independent variable $t$ with two dependent variables $u, w$ with $f_\theta, g_\theta$ denoting the neural vector fields for each case (note that this does not imply FP, HO, and VdP share the same neural surrogate). For Burgers equation, the machine first learns $u = u_\phi(x, t)$, then $\partial_t u_\phi = h_\theta(u, u_x, u_{xx})$ where $u_\phi, h_\theta$ represent two different neural scalar fields. We calculate the algebraic losses (loss 3 to 8) by plugging in the generators evaluated on the training time points. To calculate the dynamic symmetry losses (loss 1 and 2), we rewrite the above neural analytic forms as neural differential equations:

$$F_{\text{FP,HO,VdP}}(t, x, \dot{x}, \ddot{x}) := \begin{bmatrix} \dot{x}(t) \\ \dot{v}(t) \end{bmatrix} - \begin{bmatrix} f_{\theta,1} \\ f_{\theta,2} \end{bmatrix} = \mathbf{0}, \ F_{\text{LV}}(t, u, v, \dot{u}, \dot{w}) := \begin{bmatrix} \dot{u} \\ \dot{w} \end{bmatrix} - \begin{bmatrix} g_{\theta,1} \\ g_{\theta,2} \end{bmatrix} = \mathbf{0}$$

$$F_{\text{Burgers}}(t, x, u, u_t, u_x, u_{xx}) := u_t - h_\theta = 0$$

where the subscripts 1 and 2 under $f_\theta, g_\theta$ refer to the first and second component of each neural vector field. We can now evaluate $F$ on the $\epsilon$-flow of $v_i$ in loss 2, and write the IIC with prolongations as:

$$\begin{bmatrix} \text{pr}^{(1)} v_k(\dot{x} - f_{\theta,1}) \\ \text{pr}^{(2)} v_k(\dot{v} - f_{\theta,2}) \end{bmatrix} = \begin{bmatrix} 0 \\ 0 \end{bmatrix}, \ \begin{bmatrix} \text{pr}^{(1)} v_k(\dot{u} - g_{\theta,1}) \\ \text{pr}^{(1)} v_k(\dot{w} - g_{\theta,2}) \end{bmatrix} = \begin{bmatrix} 0 \\ 0 \end{bmatrix}, \ \text{pr}^{(2)} v_k(u_t - h_\theta) = 0$$

for $\text{pr}^{(a)} v_k$ being the $a$-th order prolongation of the generator vector field $v_k$, which then allows us to impose loss 1 without any known analytic equations as prior. Note that, because we are not treating $v = \dot{x}$ as a separate dependent variable but as the first derivative of $x$, the correct prolongation should be second-order since $\dot{v} \equiv \ddot{x}$.

Other than the neural vector field forms shown above, we have also trained the neural ODE surrogates in the form of scalar neural ODEs for FP, HO, and VdP ($f_\theta(t, x, \dot{x}) = \ddot{x}$). In this case, the prolonged vector field is applied on the equation as $\text{pr}^{(2)} v_i(f_\theta(t, x, \dot{x}) - \ddot{x})$. For the neural surrogate of Burgers' equation in the form $f_\theta(u, u_x, u_{xx}) = u_t$, the prolonged vector field is applied as: $\text{pr}^{(2)} v_i(f_\theta(u, u_x, u_{xx}) - u_t)$. The following table shows the neural surrogates we have trained and tested on for each experiment.

---

[1]We treated $x, v$ as two dependent variables when training the surrogate, but then we use second-order prolongation when applying the prolonged vector of $v = \xi \partial t + \phi \partial x$ on the differential equation since $\dot{v} = \ddot{x}$. It is also mathematically correct to apply first-order prolongation of $v = \tau \partial t + \xi \partial x + \phi \partial v$ on the same equation for some functions $\tau, \xi, \phi$.

Table 5: **Neural surrogates (neural analytic forms) used in each experiment.** Identical symbols across rows (e.g., $f_\theta$, $g_\theta$) denote neural models, not shared parameters. We trained two sets of neural surrogates - one set in the left column, the other in the right column, under "Neural Analytic Form." In the left surrogate for 1D viscous Burgers' equation, $\phi$ and $\theta$ respectively denotes two co-trained neural networks.

| Systems | Neural Analytic Form | |
| --- | --- | --- |
| Free Particle (FP) | $[\dot{x}, \dot{v}]^\top = [v, f_\theta(x, \dot{x})]^\top$ | $\ddot{x} = f_\theta(t, x, \dot{x})$ |
| Harmonic Oscillator (HO) | $[\dot{x}, \dot{v}]^\top = [v, f_\theta(x, \dot{x})]^\top$ | $\ddot{x} = f_\theta(t, x, \dot{x})$ |
| Van der Pol (VdP) | $[\dot{x}, \dot{v}]^\top = [v, f_\theta(x, \dot{x})]^\top$ | $\ddot{x} = f_\theta(t, x, \dot{x})$ |
| Lotka–Volterra (LV) | $[\dot{u}, \dot{w}]^\top = [g_{\theta,1}(u), g_{\theta,2}(w)]^\top$ | |
| 1D Viscous Burgers | $u = u_\phi(x,t), \ \partial_t u_\phi = h_\theta(u, u_x, u_{xx})$ | $u_t = f_\theta(u, u_x, u_{xx})$ |
| 2D Viscous Burgers | $u_t = f_\theta(u, u_x, u_y, u_{xx}, u_{yy})$ | |

**Evaluation method: principal angles between generator spans.** To assess whether the learned Lie algebra matches the ground truth, we compare *spans* of generators via principal angles rather than attempting one-to-one matches between basis elements. A Lie algebra is a vector space: any invertible linear recombination (and rescaling) of a valid generating set yields another equally valid basis. Consequently, there is no unique "canonical" set of generator vector fields for the ground-truth algebra, structure constants are basis-dependent (up to change of basis), and element-wise comparisons are ill-posed. Span comparison is basis-invariant and therefore the appropriate test.

Let $\{v_i\}_{i=1}^m$ be the learned generators and $\{w_j\}_{j=1}^r$ the ground-truth generators. For ODEs we consider

$$v_i(t, x) = \xi_i(t, x)\, \partial_t + \phi_i(t, x)\, \partial_x, \qquad w_j(t, x) = \tilde{\xi}_j(t, x)\, \partial_t + \tilde{\phi}_j(t, x)\, \partial_x,$$

while for PDEs (e.g., Burgers') we use

$$v_i(t, x, u) = \xi_i(t, x, u)\, \partial_t + \phi_i(t, x, u)\, \partial_x + \tau_i(t, x, u)\, \partial_u,$$

$$w_j(t, x, u) = \tilde{\xi}_j(t, x, u)\, \partial_t + \tilde{\phi}_j(t, x, u)\, \partial_x + \tilde{\tau}_j(t, x, u)\, \partial_u.$$

We evaluate all generators on a uniform grid $\Omega$ matched to the training domain:

$$\Omega_{\text{ODE}} = \{(t_r, x_s): \ r = 1, \ldots, R; \ s = 1, \ldots, S\} \subset [t_{\min}, t_{\max}] \times [x_{\min}, x_{\max}],$$

or

$$\Omega_{\text{PDE}} = \{(t_r, x_s, u_q): \ r = 1, \ldots, R; \ s = 1, \ldots, S; \ q = 1, \ldots, Q\}$$

$$\subset [t_{\min}, t_{\max}] \times [x_{\min}, x_{\max}] \times [u_{\min}, u_{\max}].$$

Let $p = 2$ for ODEs (components $[\xi, \phi]$) and $p = 3$ for PDEs (components $[\xi, \phi, \tau]$). We stack samples into

$$B_\ell \in \mathbb{R}^{(p|\Omega|) \times m}, \qquad B_g \in \mathbb{R}^{(p|\Omega|) \times r},$$

whose columns are the flattened component values over $\Omega$ (subscripts $\ell$ and $g$ denote *learned* and *ground truth*, respectively). Using the identity (Euclidean) inner product on $\mathbb{R}^{p|\Omega|}$, we compute reduced QR factorizations $B_\ell = Q_\ell R_\ell$ and $B_g = Q_g R_g$, then obtain the principal angles $\{\theta_k\}_{k=1}^d$ between $\text{span}(B_\ell)$ and $\text{span}(B_g)$ from

$$\cos \theta_k = \sigma_k(Q_g^\top Q_\ell), \qquad d = \min(\text{rank } B_\ell, \text{rank } B_g), \qquad \theta_k \in [0, \tfrac{\pi}{2}],$$

where $\sigma_k(\cdot)$ are singular values in descending order. We report $\{\theta_k\}$ in degrees (largest to smallest); smaller angles indicate closer span alignment, with $\theta_k = 0°$ for all $k$ iff the sampled spans coincide under the identity metric. In practice, once the correct number of generators is identified ($m = r$, full rank), $d = m$. As a scalar summary across candidate dimensions $r$, we also track $\max_k \theta_k$; the correct algebra dimension exhibits the smallest maximum principal angle and angles clustered near zero. Although there is no universal threshold or benchmark for how small principal angles must be, the fact that the maximum principal angle attains its minimum at the correct number of generators —

and that all angles at that dimension are near zero — indicates that the method has correctly identified both the dimension and the algebra.

Geometrically, the principal angles $\{\theta_k\}$ between two spaces $\mathcal{V}, \mathcal{W} \subset \mathbb{R}^N$ quantify how "aligned" the spaces are: $\theta_1$ is the smallest angle between any two unit vectors $v \in \mathcal{V}$ and $w \in \mathcal{W}$, $\theta_2$ is the smallest angle between the spaces after removing the first principal directions, and so on. Thus $\theta_k = 0°$ for all $k$ if and only if the spaces coincide, and all $\theta_k$ near $0°$ means that every direction in one space can be represented with very small error by directions in the other. In our setting, these spaces are the spans of sampled generator fields on the training grid, so small principal angles directly express *geometric agreement* between the learned and analytic symmetry directions as functions on spacetime.

**Evaluation method comparison: between principal-angle and structure constants.** We chose principal angles over other evaluation metrics - such as structure constants - as the primary metric for several reasons:

- *Basis invariance at the level of spans.* A Lie algebra is defined up to a change of basis: if $\{v_i\}$ and $\{\tilde{v}_j\}$ are two bases of the same algebra, their structure constants are related by a nontrivial change-of-basis transformation. To compare learned and ground-truth structure constants, one must first identify (or solve for) the correct linear isomorphism between the two bases, which is itself an optimization problem. Principal angles, by contrast, compare *spaces* $\text{span}\{v_i\}$ and $\text{span}\{\tilde{v}_j\}$ directly in the ambient function space, and are completely independent of the particular basis used within each span.

- *Interpretability and robustness.* Raw structure constants are sensitive to arbitrary rescalings and reorderings of generators, and their numerical values are not very intuitive to most readers: small perturbations in the basis can induce complicated, coupled changes in the table of $c_{ij}^k$. Principal angles, on the other hand, lie in $[0°, 90°]$ and have a clear geometric meaning: they measure the worst-case misalignment between the learned and true algebras on the sampled domain. This makes them a more transparent "strength-of-match" diagnostic.

- *Functional vs. purely algebraic agreement.* In our pipeline, closure and Jacobi are already enforced during training, so the learned structure constants are by construction close to constant and satisfy the Lie–algebra axioms. What is *not* guaranteed by these algebraic checks alone is that the resulting subspace of vector fields agrees with the *analytic* symmetry directions of the underlying dynamics on the domain of interest. Principal angles address exactly this point: they measure how well the Lie algebra spanned by the learned generators (as vector fields on spacetime) approximate the analytic ground-truth symmetry algebra.

**Prolongation Formulas.** In this section, we explicitly show the prolongation formulas used in training and the iteration of different jet orders shown in Figure 3.

- **Case 1**: generators of the form $v_i = \xi(t,x)\partial t + \phi(t,x)\partial x$ with $x = x(t)$.

$$D_t = \partial_t + x_t\partial_x + x_{tt}\partial_{x_t} + x_{ttt}\partial_{x_{tt}} + x_{tttt}\partial_{x_{ttt}} + \cdots$$

$$Q = \phi(t,x) - x_t\xi(t,x),\ \phi^{(1)} = D_t(\phi) - x_{tt}D_t(\xi),$$

The recursive formula gives:

$$\phi^{(k+1)} = D_t\big(\phi^{(k)}\big) - x_{t^{k+1}}D_t(\xi), \quad k \geq 1,$$

where $x_{t^k} = \frac{d^k x}{dt^k}$. Then we have the prolongations from order 1 to 4:

$$\text{pr}^{(1)}\,v = \xi\partial_t + \phi\partial_x + \phi^{(1)}\partial_{x_t},\ \text{pr}^{(2)}\,v = \text{pr}^{(1)}\,v + \phi^{(2)}\partial_{x_{tt}}$$

$$\text{pr}^{(3)}\,v = \text{pr}^{(2)}\,v + \phi^{(3)}\partial_{x_{ttt}},\ \text{pr}^{(4)}\,v = \text{pr}^{(3)}\,v + \phi^{(4)}\partial_{x_{tttt}}.$$

- **Case 2**: generators of the form $v_i = \tau(t,u,w)\partial_t + \xi(t,u,w)\partial_u + \phi(t,u,w)\partial_w$ with $u = u(t),\ w = w(t)$.

$$D_t = \partial_t + u_t\partial_u + w_t\partial_w + u_{tt}\partial_{u_t} + w_{tt}\partial_{w_t} + u_{ttt}\partial_{u_{tt}} + w_{ttt}\partial_{w_{tt}} + \cdots$$

Characteristics and first coefficients:

$$Q^u = \xi - u_t \tau, \quad Q^w = \phi - w_t \tau,$$

$$\eta^{u,(1)} = D_t(\xi) - u_{tt} D_t(\tau), \quad \eta^{w,(1)} = D_t(\phi) - w_{tt} D_t(\tau).$$

The recursive formulas give:

$$\eta^{u,(k+1)} = D_t\big(\eta^{u,(k)}\big) - u_{t^{k+1}} D_t(\tau), \quad \eta^{w,(k+1)} = D_t\big(\eta^{w,(k)}\big) - w_{t^{k+1}} D_t(\tau), \quad k \geq 1.$$

Then we have the prolongations from order 1 to 4:

$$\mathrm{pr}^{(1)} v = \tau \partial_t + \xi \partial_u + \phi \partial_w + \eta^{u,(1)} \partial_{u_t} + \eta^{w,(1)} \partial_{w_t},$$

$$\mathrm{pr}^{(2)} v = \mathrm{pr}^{(1)} v + \eta^{u,(2)} \partial_{u_{tt}} + \eta^{w,(2)} \partial_{w_{tt}},$$

$$\mathrm{pr}^{(3)} v = \mathrm{pr}^{(2)} v + \eta^{u,(3)} \partial_{u_{ttt}} + \eta^{w,(3)} \partial_{w_{ttt}},$$

$$\mathrm{pr}^{(4)} v = \mathrm{pr}^{(3)} v + \eta^{u,(4)} \partial_{u_{tttt}} + \eta^{w,(4)} \partial_{w_{tttt}}.$$

- **Case 3**: generators of the form $v_i = \xi(t,x,u)\partial t + \phi(t,x,u)\partial x + \tau(t,x,u)\partial u$ with $u = u(x,t)$.

$$D_t = \partial_t + u_t \partial_u + u_{tt} \partial_{u_t} + u_{tx} \partial_{u_x} + u_{ttt} \partial_{u_{tt}} + u_{ttx} \partial_{u_{tx}} + u_{txx} \partial_{u_{xx}} + \cdots$$

$$D_x = \partial_x + u_x \partial_u + u_{tx} \partial_{u_t} + u_{xx} \partial_{u_x} + u_{ttx} \partial_{u_{tt}} + u_{txx} \partial_{u_{tx}} + u_{xxx} \partial_{u_{xx}} + \cdots$$

Characteristic and order-1 coefficients:

$$Q = \tau - \xi u_t - \phi u_x,$$

$$\eta_t = D_t(Q) + \xi u_{tt} + \phi u_{tx} \quad \eta_x = D_x(Q) + \xi u_{tx} + \phi u_{xx}.$$

Order-2 coefficients:

$$\eta_{tt} = D_t^2(Q) + \xi u_{ttt} + \phi u_{ttx} \quad \eta_{tx} = D_t D_x(Q) + \xi u_{ttx} + \phi u_{txx} \quad \eta_{xx} = D_x^2(Q) + \xi u_{txx} + \phi u_{xxx}.$$

Order-3 coefficients:

$$\eta_{ttt} = D_t^3(Q) + \xi u_{tttt} + \phi u_{tttx} \quad \eta_{ttx} = D_t^2 D_x(Q) + \xi u_{tttx} + \phi u_{ttxx},$$

$$\eta_{txx} = D_t D_x^2(Q) + \xi u_{ttxx} + \phi u_{txxx} \quad \eta_{xxx} = D_x^3(Q) + \xi u_{txxx} + \phi u_{xxxx}.$$

Order-4 coefficients:

$$\eta_{tttt} = D_t^4(Q) + \xi u_{ttttt} + \phi u_{ttttx} \quad \eta_{tttx} = D_t^3 D_x(Q) + \xi u_{ttttx} + \phi u_{tttxx},$$

$$\eta_{ttxx} = D_t^2 D_x^2(Q) + \xi u_{tttxx} + \phi u_{ttxxx} \quad \eta_{txxx} = D_t D_x^3(Q) + \xi u_{ttxxx} + \phi u_{txxxx},$$

$$\eta_{xxxx} = D_x^4(Q) + \xi u_{txxxx} + \phi u_{xxxxx}.$$

Then we have the prolongations from order 1 to 4:

$$\mathrm{pr}^{(1)} v = \xi \partial t + \phi \partial x + \tau \partial u + \eta_t \partial_{u_t} + \eta_x \partial_{u_x},$$

$$\mathrm{pr}^{(2)} v = \mathrm{pr}^{(1)} v + \eta_{tt} \partial_{u_{tt}} + \eta_{tx} \partial_{u_{tx}} + \eta_{xx} \partial_{u_{xx}},$$

$$\mathrm{pr}^{(3)} v = \mathrm{pr}^{(2)} v + \eta_{ttt} \partial_{u_{ttt}} + \eta_{ttx} \partial_{u_{ttx}} + \eta_{txx} \partial_{u_{txx}} + \eta_{xxx} \partial_{u_{xxx}},$$

$$\mathrm{pr}^{(4)} v = \mathrm{pr}^{(3)} v + \eta_{tttt} \partial_{u_{tttt}} + \eta_{tttx} \partial_{u_{tttx}} + \eta_{ttxx} \partial_{u_{ttxx}} + \eta_{txxx} \partial_{u_{txxx}} + \eta_{xxxx} \partial_{u_{xxxx}}.$$

# B ABLATION STUDIES

## B.1 CONSTANCY LOSS

Let $m$ be the number of learned generators and let $p \in \mathcal{P}$ index samples in a (mini)batch, where each sample corresponds to a spacetime point $(t_p, x_p)$. Each generator has the form

$$v_i(t, x) = \xi_i(t, x)\,\partial_t + \phi_i(t, x)\,\partial_x, \qquad i = 1, \ldots, m.$$

Let $V_p \in \mathbb{R}^{2 \times m}$ collect the $(\xi, \phi)$ rows of the $m$ learned generators evaluated at sample $p$, i.e.,

$$V_p = \begin{bmatrix} \xi_1(t_p, x_p) & \cdots & \xi_m(t_p, x_p) \\ \phi_1(t_p, x_p) & \cdots & \phi_m(t_p, x_p) \end{bmatrix}.$$

Let $B_p^{(i,j)} \in \mathbb{R}^2$ denote the $(\xi, \phi)$-components of the Lie bracket $[v_i, v_j]$ evaluated at $(t_p, x_p)$. There are two natural ways to compute coefficients $c^{(i,j)} \in \mathbb{R}^m$ such that $[v_i, v_j] \approx \sum_{k=1}^{m} c_k^{(i,j)} v_k$ in the $(\xi, \phi)$ components:

(i) **Local projection (per-sample) + constancy.** For each $p$, solve the minimum-norm local least squares

$$c_p^{(i,j)} = \arg\min_{c \in \mathbb{R}^m} \left\| B_p^{(i,j)} - V_p c \right\|_2 = V_p^\top \left( V_p V_p^\top + \varepsilon I_2 \right)^{-1} B_p^{(i,j)}.$$

Here $I_2$ is the $2 \times 2$ identity matrix and $\varepsilon > 0$ is a small Tikhonov regularizer for numerical stability. The closure residual is $e_p^{(i,j)} = B_p^{(i,j)} - V_p c_p^{(i,j)}$. We minimize $\sum_{p \in \mathcal{P}} \| e_p^{(i,j)} \|_1$ for closure and add a constancy penalty on the dispersion of the local coefficients, e.g.

$$\sum_{p \in \mathcal{P}} \mathrm{Var}_p\big[c_p^{(i,j)}\big],$$

where $\mathrm{Var}_p[\cdot]$ denotes the elementwise variance over the batch $\mathcal{P}$, to encourage $c_p^{(i,j)}$ to be sample-independent.

(ii) **Global normal equations (constancy built-in).** Solve a single ridge-regularized global regression for each pair $(i, j)$:

$$c^{(i,j)} = \arg\min_c \sum_{p \in \mathcal{P}} \left\| B_p^{(i,j)} - V_p c \right\|_2^2 + \lambda \|c\|_2^2 = \left( \sum_{p \in \mathcal{P}} V_p^\top V_p + \lambda I_m \right)^{-1} \left( \sum_{p \in \mathcal{P}} V_p^\top B_p^{(i,j)} \right),$$

where $I_m$ is the $m \times m$ identity matrix and $\lambda > 0$ is a ridge parameter. This directly yields a single, sample-constant $c^{(i,j)}$, so no explicit constancy term is added.

**Why we prefer the current (local + constancy) formulation here:**

1. *Robustness to local rank loss and nonuniform coverage.* When some $V_p$ are ill-conditioned or nearly rank-1, the local projection keeps closure measured at each $p$ while the variance term softly aligns coefficients; the global normal equations can be biased by such regions because they aggregate $\sum_p V_p^\top V_p$ and may overfit dense/ill-conditioned parts of the state space.

2. *Optimization practicality with minibatches.* The local residuals and constancy penalty work seamlessly with stochastic training (no need to materialize $\sum_p V_p^\top V_p$ over the full dataset); they provide stable, per-batch gradients early in training when the span $\mathrm{col}(V_p)$ is still moving, whereas the global solve implicitly couples all samples and is harder to approximate well from small batches. Here $\mathrm{col}(V_p)$ denotes the column space of $V_p$.

We performed a controlled ablation on the free particle (FP) and the harmonic oscillator (HO) in which the *only* change was the estimator for the structure constants—(i) local per-sample projection with a constancy penalty versus (ii) a single global normal-equation solve. As shown in Table 6, approach (i) consistently produces smaller principal angles than (ii), indicating closer alignment to the ground-truth symmetry algebra.

Table 6: **Ablation of Constancy Loss.** Principal angles for FP (free particle) and HO (harmonic oscillator) at different implementations of structure constants. "Gen $\ell$" denotes the $\ell$-th principal angle (in degrees) between the learned and analytic generator spans, ordered nondecreasing.

| Systems | Gen 1 | Gen 2 | Gen 3 | Gen 4 | Gen 5 | Gen 6 | Gen 7 | Gen 8 |
|---|---|---|---|---|---|---|---|---|
| FP(i) | $15.390°$ | $7.052°$ | $0.781°$ | $0.260°$ | $0.092°$ | $0.082°$ | $0.005°$ | $0.002°$ |
| FP(ii) | $31.066°$ | $26.881°$ | $0.725°$ | $0.530°$ | $0.502°$ | $0.302°$ | $0.018°$ | $0.007°$ |
| HO(i) | $10.044°$ | $7.568°$ | $5.554°$ | $3.419°$ | $1.433°$ | $0.757°$ | $0.328°$ | $0.176°$ |
| HO(ii) | $28.042°$ | $21.087°$ | $14.417°$ | $9.196°$ | $8.262°$ | $2.976°$ | $0.409°$ | $0.069°$ |

## B.2 IIC AND FLOW-BASED VALIDITY

From a theoretical standpoint, the infinitesimal invariance condition (IIC) and flow-based validity are equivalent: on a smooth solution manifold, satisfying one implies the other. In practice, however, enforcing *both* losses yields markedly better numerical behavior. We verify this with an ablation in which we disable $L_{\mathrm{inv}}$, $L_{\mathrm{flow}}$, or both when training on Burgers' equation and on the harmonic oscillator. Figure 14 reports the post-training maximum principal angle between the learned and analytic generator spans. Removing either term consistently increases this angle relative to the "both on" setting, indicating poorer alignment with the ground-truth Lie algebra. These results support including both losses during training.

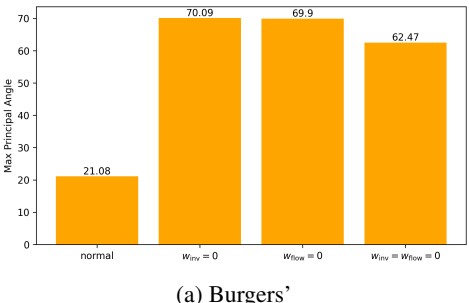
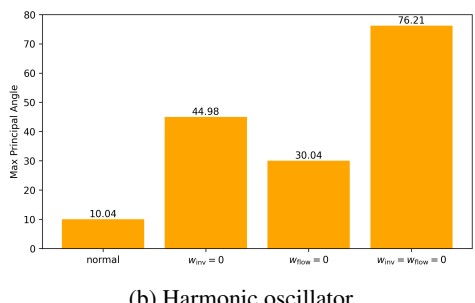

(a) Burgers'  (b) Harmonic oscillator

Figure 5: **Ablation of $L_{\mathrm{inv}}$ and $L_{\mathrm{flow}}$.** Maximum principal angle (lower is better) when different combinations of losses are used. "normal" uses both losses; $w_{\mathrm{inv}}$ and $w_{\mathrm{flow}}$ denote their weights.

## B.3 NEURAL SURROGATES

To assess the model-agnostic nature of LieDynNet, we repeated the full pipeline while swapping the neural surrogate used for the dynamics. We kept all other settings fixed and measured the post-training principal angles on the harmonic oscillator and Burgers' equation with different surrogates. The two surrogate parameterizations are listed in Table 5; we refer to them as (1) and (2), corresponding to the first and second columns of that table, respectively. As shown in Table 7, the angles are comparable across the two surrogates for each system, indicating similar alignment with the analytic symmetry subspace. This supports the claim that LieDynNet is not tied to a particular surrogate form.

Table 7: **Neural surrogates: post-training principal angles (degrees).** Comparison of principal angles for the harmonic oscillator (HO) and Burgers' equation under two surrogate parameterizations. "(1)" and "(2)" denote the first and second surrogate forms in Table 5. Lower is better.

| System | Gen 1 | Gen 2 | Gen 3 | Gen 4 | Gen 5 | Gen 6 | Gen 7 | Gen 8 |
|---|---|---|---|---|---|---|---|---|
| HO (1) | $11.437°$ | $9.152°$ | $4.708°$ | $3.364°$ | $2.891°$ | $1.823°$ | $1.574°$ | $1.605°$ |
| HO (2) | $10.044°$ | $7.568°$ | $5.554°$ | $3.419°$ | $1.433°$ | $0.757°$ | $0.328°$ | $0.176°$ |
| 1D Burgers' (1) | $19.812°$ | $18.731°$ | $12.644°$ | $8.907°$ | $6.953°$ | | — | |
| 1D Burgers' (2) | $21.081°$ | $17.129°$ | $13.778°$ | $7.343°$ | $5.696°$ | | — | |

## C ADDITIONAL EXPERIMENT RESULTS

**Heat maps.** In this section, we include the heatmaps of the learned generators in each experiment.

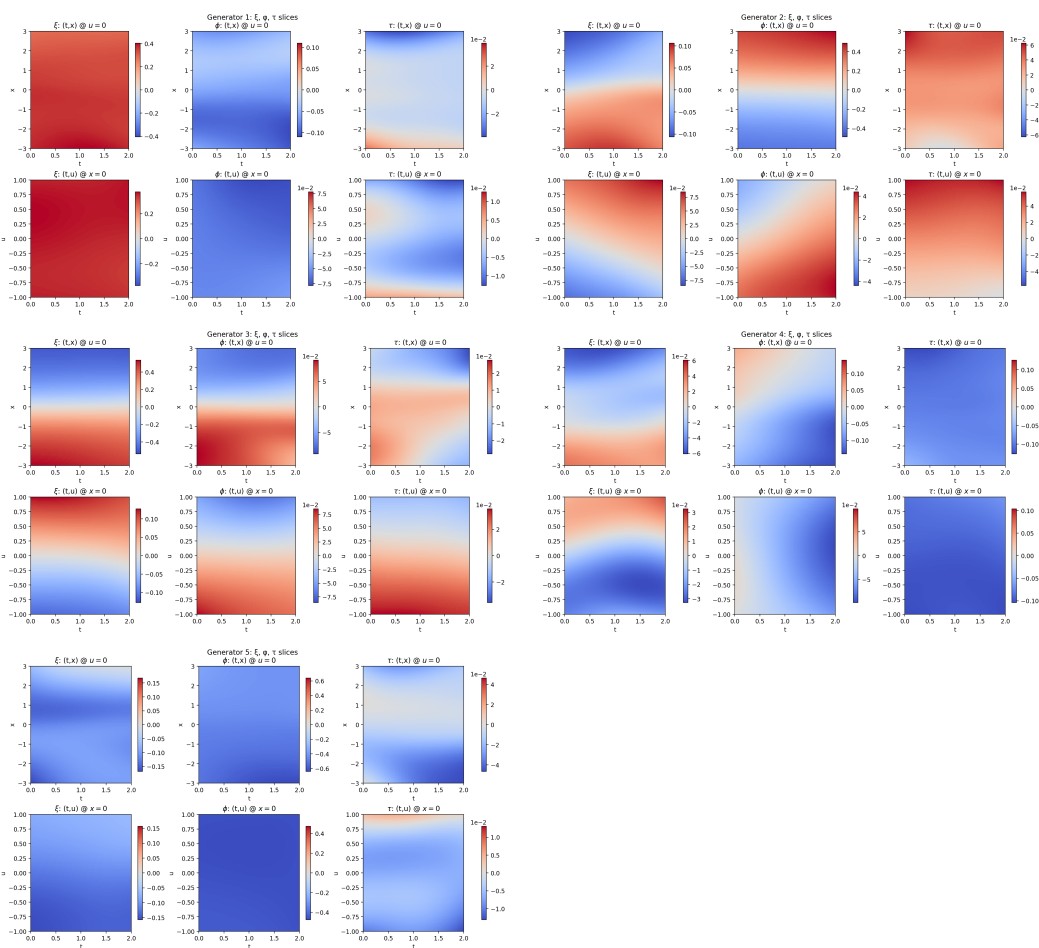

Figure 6: 1D Viscous Burgers' equation: heat maps of learned generators.

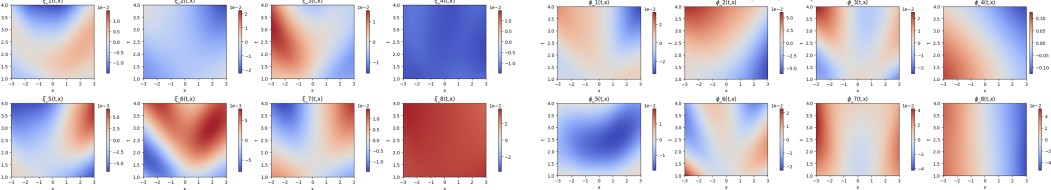

Figure 7: Harmonic Oscillator: heat maps of learned generators.

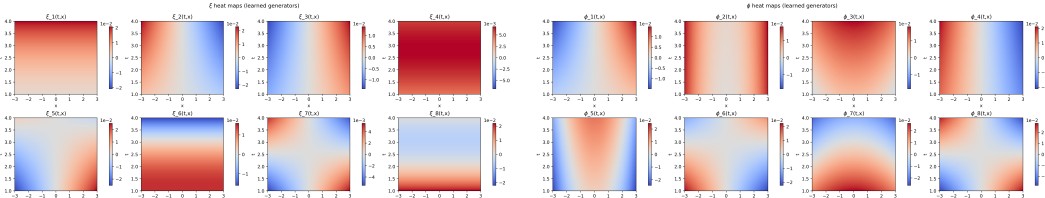

Figure 8: Free Particle: heat maps of learned generators.

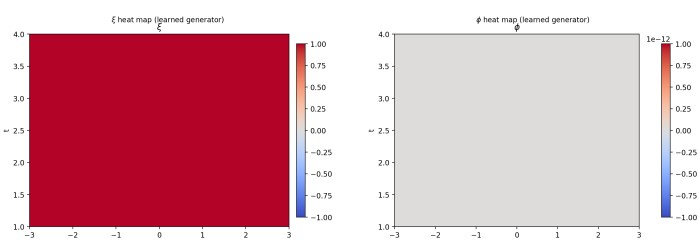

Figure 9: Van der Pol: heat maps of learned generators.

**LV Results.** In this section, we report the relevant results for the experiment done on Lotka-Volterra.

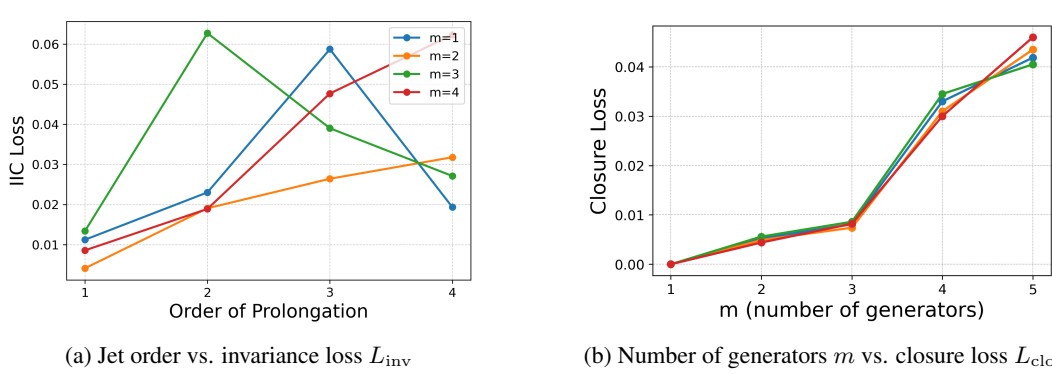

(a) Jet order vs. invariance loss $L_{inv}$

(b) Number of generators $m$ vs. closure loss $L_{clo}$

Figure 10: **Lotka–Volterra results.** (a) $L_{inv}$ vs. jet order $k$. (b) $L_{clo}$ vs. number of generators $m$.

**After-flow residuals and result visualizations.** In this section, we show the residuals of plugging the after-flow trajectories under each learned generator into the analytic equation of each experiment. For each experiment, we generated a solution trajectory directly from the analytic equation, and then integrated it along the pseudo-time parameter $\epsilon$ with Heun step (rk2) to get the flow. We then plugged the after-flow trajectories into the analytic equation to see whether they remain in the solution space, and plotted different $\epsilon$ values in $[0, 2]$ versus RMSE of residual. To visualize the results, we also chose FP and VdP as examples in Figure 15 and Figure 16 to show the classical shapes of solution trajectories are preserved after flowing along the learned generators.

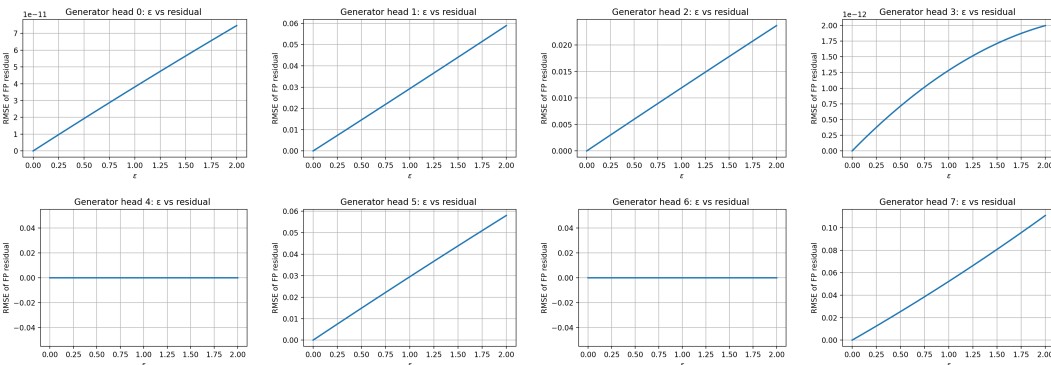

Figure 11: Free Particle: $\epsilon$ versus after-flow residuals.

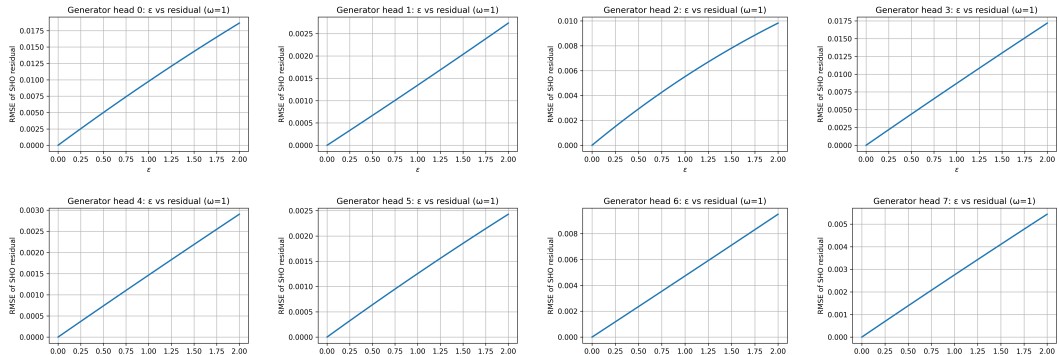

Figure 12: Harmonic Oscillator: $\epsilon$ versus after-flow residuals.

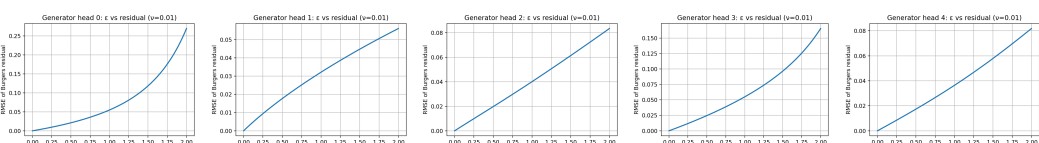

Figure 13: 1D Burgers' Equation: $\epsilon$ versus after-flow residuals.

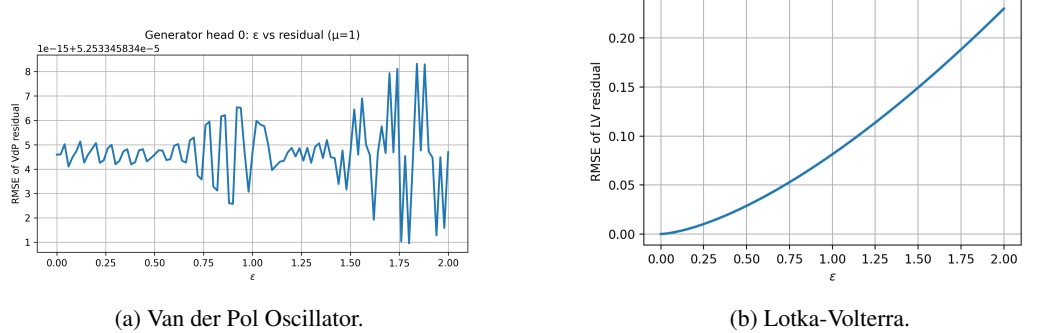

(a) Van der Pol Oscillator.

(b) Lotka-Volterra.

Figure 14: $\epsilon$ versus after-flow residuals: (a)VdP and (b) LV.

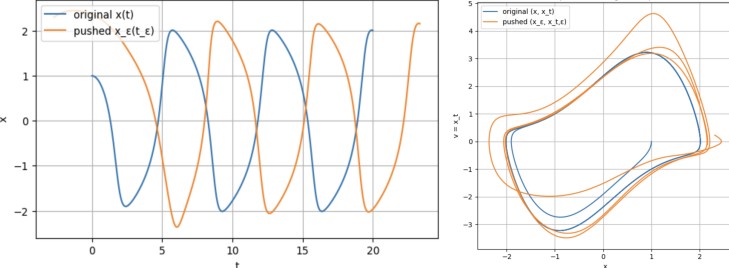

Figure 15: **Visualization of after-flow trajectories under learned symmetry generator for VdP.** (a) The panel on the left shows the original trajectory and transformed trajectory via the learned symmetry generator with $\epsilon = 4.0$. The left panel shows that the ground-truth time-translation symmetry is recovered. (b) The panel on the right is the representative phase-portrait of $(x, \dot{x})$.

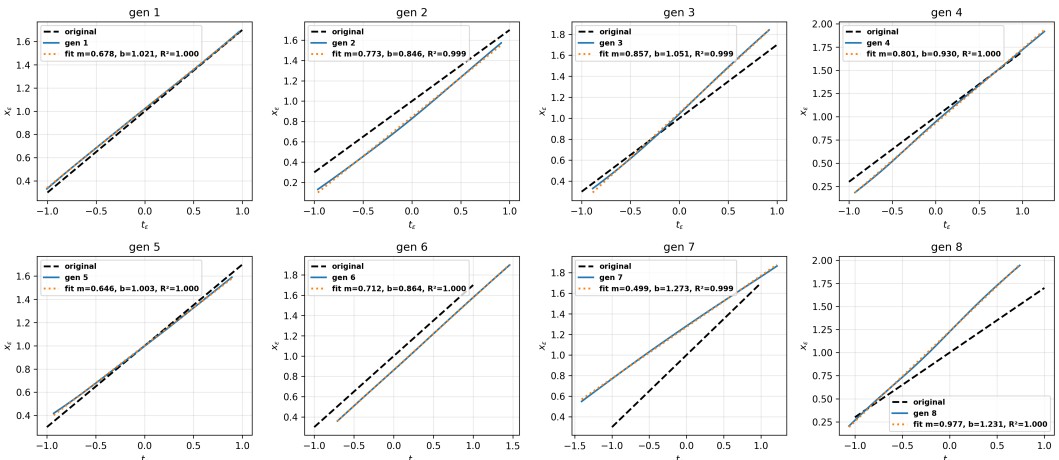

**Figure 16: Visualization of after–flow trajectories under learned symmetry generator for FP.** Transformed trajectories under the learned Lie generators with total deformation $\epsilon = 1$. Starting from the free–particle trajectory $x(t) = a\,t + b$ (black dashed), we integrate the $\epsilon$–flow $\frac{dt}{d\epsilon} = \xi_k(t, x)$, $\frac{dx}{d\epsilon} = \phi_k(t, x)$ using RK4 to obtain $(t_\epsilon, x_\epsilon)$ for each generator $k$ (one panel per $k$ on a $4 \times 4$ grid). Colored curves show the transformed trajectories ; dotted lines overlay the least-squares fit $x \approx m_k t + c_k$ with $R^2$ reported in the legend (boldface). The transformed trajectories preserve the straight shape of FP's solution.

**Baseline Comparison.** To clarify how our principal-angle metric reflects algebra alignment quality, we focus on the most challenging setting in our experiments, namely the 1D viscous Burgers equation, where the learned principal angles are largest relative those from other experiments. We compare our method against the symmetry-discovery approach of Ko et al. (Ko et al., 2024), which also learns symmetry generators for the 1D viscous Burgers equation ($u_t + uu_x = \nu u_{xx}$). Their method recovers four symmetry generators: $v_1 = \partial_x$, $v_2 = \partial_t$, $v_3 = t\,\partial_x + \partial_u$, $v_4 = u\,\partial_u$. For both methods we apply the same algebra-alignment evaluation described before: we compute principal angles between the span of the learned generators and all four-dimensional subspaces of the five-dimensional ground-truth Burgers Lie algebra. Since there are $\binom{5}{4} = 5$ such subsets, we obtain five sets of principal angles. For each subset, we record the maximum principal angle, and we also record the maximum principal angle for our method; these six values are visualized in Figure 17.

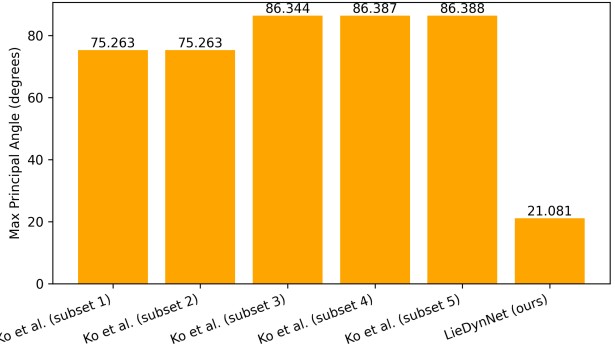

**Figure 17: Maximum principal-angle comparison on the 1D viscous Burgers equation.** Each bar labeled "Ko et al. (subset $k$)" shows the maximum principal angle (in degrees) between the four generators learned by Ko et al. (Ko et al., 2024) and one of the $\binom{5}{4} = 5$ four-dimensional subspaces of the five-dimensional ground-truth Lie algebra. The bar labeled "LieDynNet (ours)" shows the corresponding maximum angle for our method, which is substantially smaller than all Ko et al. subsets, indicating a closer alignment with the ground-truth symmetry algebra.

**Plot of Table 3.**   We use line plots to visualize the principal angles recorded for the Free Particle, Harmonic Oscillator, and Burgers' equation experiments (data taken from Table 3).

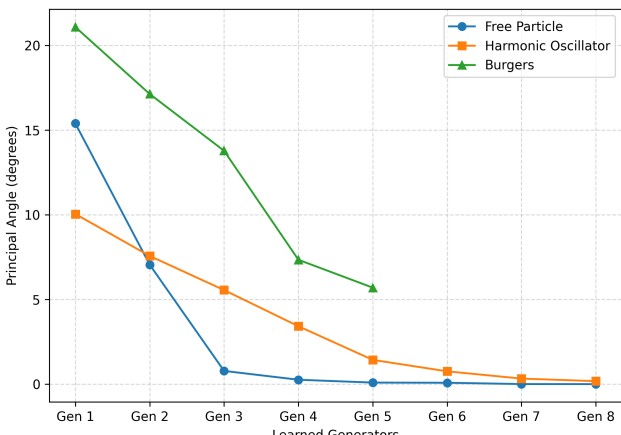

Figure 18:  Principal angles for the Free Particle, Harmonic Oscillator, and Burgers' equation experiments across learned generators (corresponding to Table 3).

**2D Viscous Burgers' Results.**   In this section, we report the relevant results for the experiments done on 2D viscous Burgers' equation.

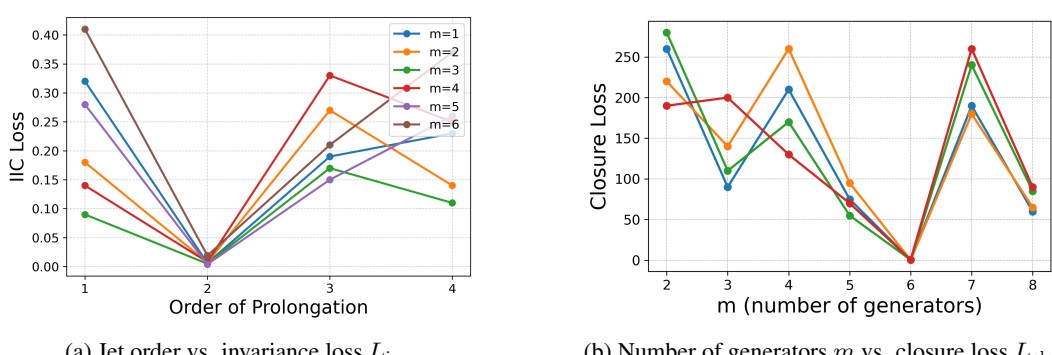

(a) Jet order vs. invariance loss $L_{\mathrm{inv}}$

(b) Number of generators $m$ vs. closure loss $L_{\mathrm{clo}}$

Figure 19: **2D Burgers' results.** (a) $L_{\mathrm{inv}}$ vs. jet order $k$. (b) $L_{\mathrm{clo}}$ vs. number of generators $m$.

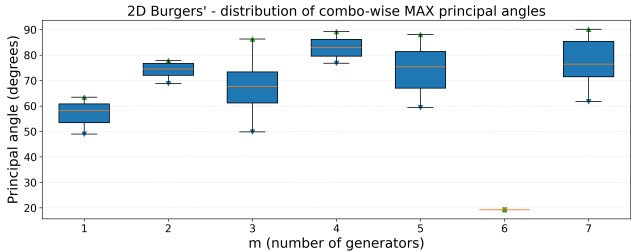

Figure 20: **Distribution of combination-wise maximum principal angles; same notation as in Figure 2**

# D TRAINING DETAILS

Table 8: **Compute summary for the main experiments.** For each system, we report the number of trainable parameters (#Params), wall-clock time per epoch, total training time, and peak GPU memory usage. All measurements are obtained on a single GPU ([GPU model]). Abbreviations: Free-Particle (FP), Simple Harmonic Oscillator (SHO), Van-der Pol Oscillator (VdP), Lotka-Volterra (LV), Burgers Equation (BE), Neural ODE (NO), Generator Network (GN), Neural PDE (NP).

| Experiment | #Params | Time / epoch (s) | Total training time (h) | Peak GPU mem. (GB) |
|---|---|---|---|---|
| FP | NO(4481), GN(18960) | 0.195 | 0.603 | 59.99 |
| SHO | NO(4481), GN (18960) | 0.147 | 0.295 | 30.17 |
| VdP | NO (4481), GN(17154) | 0.034 | 0.079 | 59.97 |
| LV | NO(4482), GN(17411) | 0.032 | 0.073 | 59.93 |
| BE | NP(17153), GN(18959) | 0.065 | 0.282 | 30.18 |

## D.1 SECOND–ORDER SYSTEMS (FREE PARTICLE, HARMONIC OSCILLATOR, VAN DER POL)

For the one–dimensional mechanical benchmarks (free particle, simple harmonic oscillator, and Van der Pol oscillator) we use a common neural–ODE pipeline based on a scalar second–order surrogate

$$\ddot{x}(t) \; = \; f_\theta\big(x(t), \dot{x}(t), t\big),$$

with system–specific ground–truth dynamics:

$$\text{Free particle:} \quad \ddot{x} = 0, \tag{1}$$

$$\text{SHO:} \quad \ddot{x} = -\omega^2 x, \tag{2}$$

$$\text{Van der Pol:} \quad \ddot{x} = \mu(1 - x^2)\dot{x} - x. \tag{3}$$

**Synthetic trajectory data.** For each system we generate a batch of synthetic trajectories $\{x^{(i)}(t_n), \dot{x}^{(i)}(t_n)\}$ on a uniform time grid $t_n = n\,\Delta t$, $n = 0, \ldots, T-1$, starting from randomly sampled initial conditions $(x_0^{(i)}, \dot{x}_0^{(i)})$. For the free particle and SHO we use closed–form solutions ($x(t) = x_0 + \dot{x}_0 t$, $\dot{x}(t) = \dot{x}_0$ for the free particle; the standard sinusoidal formulas for the SHO), while for the Van der Pol oscillator we integrate the first–order system $\dot{x} = \dot{x}$, $\dot{v} = \mu(1 - x^2)v - x$ with an RK4 time–stepping scheme using small time step $\Delta t$. Independent Gaussian noise is added to the observed positions and velocities to test robustness.

Given the velocity time series, we form supervised "pseudo–labels" for the acceleration at interior times via a central finite–difference approximation

$$a_{\text{FD}}^{(i)}(t_n) \; \approx \; \frac{\dot{x}^{(i)}(t_{n+1}) - \dot{x}^{(i)}(t_{n-1})}{2\Delta t}, \qquad n = 1, \ldots, T-2,$$

and build a regression dataset by pairing mid–point states and times with these finite–difference accelerations,

$$X = \big(x^{(i)}(t_n), \dot{x}^{(i)}(t_n), t_n\big) \in \mathbb{R}^3, \qquad y = a_{\text{FD}}^{(i)}(t_n) \in \mathbb{R},$$

and reshaping over all trajectories and time indices.

**Prior–free second–order neural surrogate.** The acceleration field $f_\theta$ is represented by a fully–connected MLP

$$f_\theta : \mathbb{R}^3 \to \mathbb{R}, \qquad (x, \dot{x}, t) \mapsto f_\theta(x, \dot{x}, t),$$

with input $(x, \dot{x}, t)$, two hidden layers of width 64, and a scalar output, using $\tanh$ nonlinearities on all hidden layers. No physics prior (e.g. polynomial, Hamiltonian, or parametric form) is imposed: the network is a generic MLP on the raw state and time coordinates.

Before training, we apply per–feature z–score normalization to the inputs and outputs:

$$\tilde{X} = (X - \mu_X)/\sigma_X, \qquad \tilde{y} = (y - \mu_y)/\sigma_y,$$

with means and standard deviations computed over the training set. These statistics are stored and reused at evaluation time.

We train $f_\theta$ by minimizing the mean–squared error between predicted and target accelerations on minibatches of the normalized dataset,

$$\mathcal{L}_{\text{MSE}}(\theta) \;=\; \mathbb{E}_{(X,y)}\Big[\big(f_\theta(\tilde{X}) - \tilde{y}\big)^2\Big],$$

using the Adam optimizer with learning rate $3 \times 10^{-3}$, and 2,000 gradient steps (JAX + Optax; we fall back to a simple momentum SGD implementation if Optax is unavailable). Gradients are obtained via automatic differentiation (no hand–coded adjoints), and we periodically evaluate the full–dataset MSE to monitor convergence.

**Learned–ODE rollout and densified data.**   To verify dynamical fidelity, we integrate the learned second–order neural ODE

$$\dot{x} = v, \qquad \dot{v} = f_\theta(x, v, t)$$

forward in time using an RK4 integrator, starting from unseen initial conditions, and compare the resulting trajectories $(x(t), v(t))$ against analytic (free particle, SHO) or high–accuracy numerical (Van der Pol) references. All ODE rollouts use the same time step $\Delta t$ as the data generator, and the neural surrogate is always evaluated on normalized inputs and then un–normalized back to physical acceleration.

In addition, for the symmetry–learning experiments we use the trained $\ddot{x} = f_\theta$ model itself to generate a denser coverage of the phase–space box by sampling many initial conditions and rolling out the learned dynamics while enforcing simple bounding heuristics on $|x(t)|$. These model–generated trajectories are post–processed exactly as above (central difference on $\dot{x}$ to obtain accelerations at interior time points), forming a large cloud of labeled points in $(x, \dot{x}, t, a)$ space that is then used to train the symmetry generators.

**Neural symmetry generators.**   For all three second–order benchmarks we parameterize $n$ Lie point symmetry generators on the *configuration space* $(t, x)$ as

$$v_i(t, x) \;=\; \xi_i(t, x)\, \partial_t + \phi_i(t, x)\, \partial_x, \qquad i = 1, \ldots, n,$$

where $n = 8$ for the free particle and harmonic oscillator, and $n = 1$ for the Van der Pol oscillator. All generators share a common Haiku MLP "trunk" of width 128 with $\tanh$ activations, followed by $n$ separate linear heads, so that the network

$$g_\psi : (t, x) \mapsto \big[\xi_i(t, x), \phi_i(t, x)\big]_{i=1}^{n} \in \mathbb{R}^{n \times 2}$$

produces the generator coefficients at any point $(t, x)$. Before feeding $(t, x)$ to the network we apply a fixed affine rescaling to map the training domain into a compact box.

**Generator losses and three–stage schedule.**   To train $g_\psi$ we use a family of seven loss terms (we merged $L_{\text{clo}}$ and $L_{\text{const}}$ into one loss), evaluated on two types of batches:

- **Algebraic batches:** random points $z = (t, x)$ in a rectangular box, used in losses $L_1$–$L_5$.
- **On–shell jet batches:** points $(x, v, t, a)$ lying on the learned equation $\Delta(x, v, t, a) = f_\theta(x, v, t) - a = 0$, obtained by rolling out the neural ODE and using the surrogate $f_\theta$ to define $a$. These are used in losses $L_6$ and $L_7$.

The individual losses are:

- $L_1$: *Lie bracket closure and constancy.* At each $z$ we form Lie brackets $[v_i, v_j](z)$ using Jacobians computed by autodiff, and fit structure constants $c_{ij}^{k}$ via a small Tikhonov–regularized least–squares problem $[v_i, v_j](z) \approx \sum_k c_{ij}^{k} v_k(z)$. We penalize both the projection residual and the variance of $c_{ij}^{k}$ across $z$, encouraging a constant finite–dimensional Lie algebra.

- $L_2$: *Jacobi identity.* Using the fitted $c_{ij}^{k}$, we penalize violations of the Jacobi identities $\sum_{\text{cyc}} c_{ij}^{m} c_{mk}^{\ell} = 0$.

- $L_3$: *Skew–symmetry.* We penalize deviations from $c_{ij}^k + c_{ji}^k = 0$.

- $L_4$: *Bilinearity.* We sample random linear combinations of generators and enforce that the bracket is bilinear by comparing brackets of sums with sums of brackets.

- $L_5$: *Column independence.* We form the empirical Gram matrix of $\{v_i(z)\}_{i=1}^n$ over a batch and penalize small singular values, encouraging the generator family to span an $n$–dimensional subspace rather than collapsing.

- $L_6$: *Infinitesimal invariance of the second–order ODE.* We consider the residual $\Delta(x, v, t, a) = f_\theta(x, v, t) - a$ and construct the second prolongation $\mathrm{pr}^{(2)} v_i$ of each generator on the jet space $(t, x, v, a)$. Using autodiff for all derivatives of $f_\theta$ and $v_i$, we evaluate $\mathrm{pr}^{(2)} v_i(\Delta)$ on on–shell jets ($\Delta = 0$) and penalize its magnitude, enforcing infinitesimal invariance of the learned equation $\ddot{x} = f_\theta$.

- $L_7$: *Flow–based invariance in jet space.* Starting from an on–shell jet $(x, v, t, a)$, we integrate the jet ODE induced by the prolonged generator $v_i$ with a small group parameter $\varepsilon$ (one Heun/RK2 step on $(x, v, t, a) \mapsto (x', v', t', a')$), and penalize the post–flow residual $|f_\theta(x', v', t') - a'|$. This complements $L_6$ with a finite–$\varepsilon$ test.

In the free particle experiment we additionally use an eighth, purely unsupervised span regularizer $L_8$, which evaluates the generators on a dense grid in $(t, x)$, forms the associated Gram matrix, and promotes large, isotropic span (via a log–determinant and conditioning penalty). This encourages the learned generators to occupy an 8–dimensional subspace without collapsing, while remaining fully prior–free (no access to analytic generators).

We optimize the generator parameters $\psi$ with a three–stage curriculum. In all cases we use Adam, JAX–jitted training steps, and global gradient clipping, but the loss weights differ by stage:

- **Stage 1 (algebra pre–training).** We emphasize the algebraic losses with modest invariance: weights $(w_1, \ldots, w_7) \approx (1, 1, 1, 1, 1, 0.2, 0)$. For the free particle, $w_{\mathrm{span}} = 0$ in this stage.

- **Stage 2 (Dynamics–aware refinement).** We increase the weights on the invariance losses $L_6, L_7$ (e.g. $w_6 \approx 0.7$, $w_7 \approx 0.5$), keeping the algebraic constraints active. For the free particle, we also turn on a small span weight $w_8 > 0$.

- **Stage 3 (cooldown and stabilization).** We further increase the invariance weights and, in the free particle case, the span weight. This stage also uses a simple "cooldown" mechanism: if any individual loss blows up beyond a large threshold, we revert to the best parameters so far, shrink the learning rate, and restart the optimizer.

Across all three benchmarks, all derivatives (for both the neural ODE $f_\theta$ and the generators $v_i$) are obtained via automatic differentiation, and all loss terms are fully vectorized over batches of $(t, x)$ or $(x, v, t, a)$ points. This yields learned generators that simultaneously (i) form a finite–dimensional Lie algebra and (ii) act as continuous symmetries of the learned second–order dynamics $\ddot{x} = f_\theta(x, \dot{x}, t)$.

## D.2 LOTKA-VOLTERRA

We consider the classical Lotka–Volterra system

$$\dot{x} = \alpha x - \beta xy, \qquad \dot{y} = -\gamma y + \delta xy, \tag{4}$$

with parameters $(\alpha, \beta, \gamma, \delta) = (1.5, 1.0, 3.0, 1.0)$ and state $z = (x, y) \in \mathbb{R}^2$.[2]

**Stage 0: synthetic LV trajectories.** We draw $B = 32$ i.i.d. initial conditions $z_0^{(b)} \sim \mathcal{U}([x_0^{\min}, x_0^{\max}]^2)$ with $x_0^{\min} = 0.5$, $x_0^{\max} = 1.5$, and integrate the ground–truth LV ODE on $[0, T_{\mathrm{data}}]$ with $T_{\mathrm{data}} = 20$ using a fourth–order Runge–Kutta scheme with step $\Delta t = 0.05$:

$$z_{n+1}^{(b)} = \mathrm{RK4}\big(f_{\mathrm{LV}}, z_n^{(b)}, \Delta t\big), \qquad n = 0, \ldots, N_{\mathrm{data}} - 1, \quad N_{\mathrm{data}} = T_{\mathrm{data}}/\Delta t + 1. \tag{5}$$

This yields a tensor $z_{\mathrm{true}} \in \mathbb{R}^{B \times T \times 2}$ of prey–predator trajectories.

---

[2]Implementation details follow the released JAX/Haiku code.

We normalize the state componentwise,

$$s = \frac{z - \mu_z}{\sigma_z}, \qquad \mu_z = \mathbb{E}[z_{\text{true}}], \quad \sigma_z = \text{Std}[z_{\text{true}}] + 10^{-6}, \tag{6}$$

and work henceforth in the normalized coordinates $s = (s_x, s_y)$.

**Stage 1–2: prior–free neural LV surrogate.** We approximate the (unknown to the algorithm) LV vector field by a purely data–driven neural ODE

$$f_\theta : \mathbb{R}^2 \to \mathbb{R}^2, \qquad \dot{s} \approx f_\theta(s), \tag{7}$$

implemented as an MLP with $\tanh$ activations, hidden width $64$, and two hidden layers. Importantly, $f_\theta$ is *prior free*: we do not hard–code any LV structure (no polynomial form, no positivity constraints, etc.); the only "prior" is that the dynamics depends smoothly on the state $s$.

We train $f_\theta$ in two stages:

- **Stage 1 (multi–shooting on short segments).** From the normalized trajectories $s_{\text{true}}$ we extract overlapping segments of length $L$ corresponding to a physical window $T_{\text{segment}} = 3$, with stride $L/2$. Let $s^{(m)} \in \mathbb{R}^{L \times 2}$ denote the $m$–th segment and $s_0^{(m)} = s_0^{(m)}$ its initial state. For each segment we roll out the neural ODE

$$s_{\text{pred}}^{(m)} = \text{rollout}\big(f_\theta, s_0^{(m)}, \Delta t, L\big), \tag{8}$$

using the same RK4 scheme as above, and minimize the mean–squared segment mismatch

$$\mathcal{L}_{\text{seg}}(\theta) = \frac{1}{ML} \sum_{m=1}^{M} \sum_{n=0}^{L-1} \big\| s_{\text{pred}}^{(m)}(t_n) - s_{\text{true}}^{(m)}(t_n) \big\|_2^2. \tag{9}$$

  We optimize $\theta$ with Adam (lr $= 10^{-3}$) for $8000$ steps.

- **Stage 2 (full–trajectory fine–tuning).** We then fine–tune on the full normalized trajectories $s_{\text{true}}$ by rolling out from each normalized initial condition $s_0^{(b)}$ for the entire horizon and minimizing

$$\mathcal{L}_{\text{full}}(\theta) = \frac{1}{BT} \sum_{b=1}^{B} \sum_{n=0}^{T-1} \big\| s_{\text{pred}}^{(b)}(t_n) - s_{\text{true}}^{(b)}(t_n) \big\|_2^2, \tag{10}$$

  again using RK4 and Adam with a smaller learning rate $10^{-4}$ for $3000$ steps.

After Stage 2, the parameters $\hat{\theta}$ are frozen and define the learned LV surrogate $f_{\hat{\theta}}$; all symmetry learning uses this frozen neural ODE.

**Stage 3: supervised dataset from the learned LV ODE.** To obtain dense vector–field samples, we roll out the *learned* neural ODE in the original (unnormalized) coordinates. We draw $n_{\text{traj}}$ new initial conditions $z_0^{(b)} \sim \mathcal{U}([x_0^{\min}, x_0^{\max}]^2)$, normalize to $s_0^{(b)}$, and integrate

$$\dot{s} = f_{\hat{\theta}}(s), \qquad z = \mu_z + \sigma_z \odot s, \tag{11}$$

for $T$ time steps with step $\Delta t$. At each time we evaluate the normalized RHS $f_{\hat{\theta}}(s)$ and rescale to obtain the physical derivatives

$$\dot{z} = \sigma_z \odot f_{\hat{\theta}}\left(\frac{z - \mu_z}{\sigma_z}\right). \tag{12}$$

From these rollouts we construct a supervised dataset over midpoints $t_k$:

$$X = (t, x, y) \in \mathbb{R}^3, \qquad y_{\text{targets}} = (\dot{x}, \dot{y}) \in \mathbb{R}^2, \tag{13}$$

by stacking $(t, x, y)$ and the corresponding $(\dot{x}, \dot{y})$ across all trajectories and times.

**Stage 4: neural symmetry generators for LV.** We parameterize $n$ candidate Lie point symmetry generators in the extended space $(t, x, y)$ as

$$v_i(t, x, y) = \tau_i(t, x, y)\, \partial_t + \xi_i(t, x, y)\, \partial_x + \phi_i(t, x, y)\, \partial_y, \qquad i = 1, \ldots, n. \qquad (14)$$

In code we use a shared "trunk" network plus $n$ linear heads (Gen8): an MLP with two hidden layers of width $128$ and $\tanh$ activation maps $(t, x, y)$ to a tensor $[\tau_i, \xi_i, \phi_i]_{i=1}^n \in \mathbb{R}^{n \times 3}$.

Let $F_\psi(t, x, y)$ denote the stacked generator outputs and let $J_\psi(t, x, y)$ denote their Jacobian with respect to $(t, x, y)$, both obtained by automatic differentiation.

We then train $\psi$ using *only* two invariance losses:

- **$L_{\text{inv}}$: first–prolongation invariance.** For each generator $v_i$ we form its first prolongation in the jet space $(t, x, y, p, q)$ with $p = \dot{x}$, $q = \dot{y}$, using standard formulas and the LV field $W$.[3] Evaluated on on–shell jets $(t, x, y, p, q)$ with $(p, q)$ given by the frozen LV surrogate, we penalize the squared (or absolute) residual of the infinitesimal invariance condition

$$\mathrm{pr}^{(1)} v_i\big(\dot{z} - f_{\hat{\theta}}(z)\big) = 0,$$

  normalized by local scales and regularized by small penalties on $\|\tau_i, \xi_i, \phi_i\|$ and their Jacobians.

- **$L_{\text{flow}}$: flow–based invariance (Heun in jet space).** We also consider the finite–$\varepsilon$ flow of each generator in jet space. Starting from an on–shell jet $(t, x, y, p, q)$, we integrate the jet–space ODE for $v_i$ with a small step $\varepsilon$ (Heun/RK2, typically one step), obtaining $(t', x', y', p', q')$. We then enforce that the pushed–forward jet remains on the LV manifold by penalizing

$$\big\|(p', q') - f_{\hat{\theta}}(x', y')\big\|,$$

  averaged over generators, batch points, and initial jets.

In practice we use $n = 1$ generator for the LV experiment and train in three stages with Adam and global gradient clipping: Stage 1 uses only $L_6$; Stages 2 and 3 gradually increase the weight on $L_7$ while keeping $L_6$ active. All structure losses on the Lie algebra (closure, Jacobi, bilinearity, column independence) are disabled for LV ($w_1 = \cdots = w_5 = 0$); the generator is therefore identified purely by its invariance to the learned LV dynamics.

### D.3 Burgers' Equation

We study the viscous Burgers equation on a periodic domain,

$$u_t + u\, u_x = \nu u_{xx}, \qquad (t, x) \in [0, 5] \times [-6, 6], \qquad \nu = 10^{-2}, \qquad (15)$$

with smooth localized initial data. The ground–truth trajectories $u(t, x)$ are generated using a spectral Fourier method with wavenumbers $k = 2\pi\, \text{fftfreq}(N_x)$ and a fourth–order Runge–Kutta integrator:

$$u_x = \mathcal{F}^{-1}\big(ik\, \mathcal{F}[u]\big), \quad u_{xx} = \mathcal{F}^{-1}\big(-k^2\, \mathcal{F}[u]\big), \qquad (16)$$

$$u_t = -u\, u_x + \nu u_{xx}, \qquad (17)$$

where $\mathcal{F}$ denotes the Fourier transform. This yields pairs $\big(u(t, x), u_t(t, x)\big)$ on a space–time grid.

**Prior–free neural PDE surrogate.** From these trajectories we build a supervised dataset of local jets

$$X = \big(u,\, u_x,\, u_{xx}\big) \in \mathbb{R}^3, \qquad y = u_t \in \mathbb{R},$$

followed by standard normalization $X \mapsto (X - \mu_X)/\sigma_X$, $y \mapsto (y - \mu_y)/\sigma_y$. We then train a fully–connected MLP

$$f_\theta : \mathbb{R}^3 \to \mathbb{R}, \qquad u_t \approx f_\theta(u, u_x, u_{xx}), \qquad (18)$$

with $\tanh$ activations and hidden width $128$, by minimizing the mean–squared error

$$\mathcal{L}_{\text{PDE}}(\theta) = \frac{1}{N} \sum_{n=1}^{N} \big(f_\theta(X_n) - y_n\big)^2, \qquad (19)$$

using Adam. No physics prior is hard–coded into $f_\theta$; it is a purely data–driven surrogate for the Burgers right–hand side. After training, $\theta$ is frozen and $f_\theta$ is used as a differentiable black–box PDE.

---

[3]All jet coefficients and derivatives are obtained via JAX automatic differentiation.

**On–shell jet dataset for invariance.** To train symmetry generators we require *on–shell* jets of the solution manifold. We therefore roll out $f_\theta$ from a family of smooth periodic initial conditions $\{u_0^{(b)}(x)\}_{b=1}^B$ (sines/cosines with different amplitudes and wavenumbers) on a periodic domain $x \in [0, L)$, again using a spectral RK4 integrator. For each saved time $t_k$ and grid point $x_j$ we assemble

$$u = u(t_k, x_j), \qquad u_x = u_x(t_k, x_j), \quad u_{xx} = u_{xx}(t_k, x_j), \tag{20}$$

$$u_{xxx} = u_{xxx}(t_k, x_j), \qquad x = x_j, \qquad t = t_k, \qquad u_t = f_\theta(u, u_x, u_{xx}), \tag{21}$$

where all spatial derivatives are computed spectrally. This yields a jet dataset

$$\mathbf{X} = \big(t, u, u_x, u_{xx}, u_{xxx}, x\big) \in \mathbb{R}^6, \qquad \mathbf{y} = u_t \in \mathbb{R}, \tag{22}$$

sampling the 2–jet bundle of the learned PDE.

**Neural symmetry generators.** We parameterize $n$ Lie point symmetry generators of the form

$$v_i(t, x, u) = \xi_i(t, x, u)\, \partial_t + \phi_i(t, x, u)\, \partial_x + \tau_i(t, x, u)\, \partial_u, \qquad i = 1, \ldots, n. \tag{23}$$

All $n$ generators share a common neural "trunk" with hidden width 128, and have $n$ separate linear heads. Concretely, we learn a map

$$g_\psi : \mathbb{R}^3 \to \mathbb{R}^{n \times 3}, \qquad (t, x, u) \mapsto \big[\xi_i(t, x, u), \phi_i(t, x, u), \tau_i(t, x, u)\big]_{i=1}^n, \tag{24}$$

implemented as a Haiku MLP with two hidden layers and $\tanh$ activation, preceded by an affine rescaling of $(t, x, u)$ onto a compact box.

**Algebraic structure losses.** Given the generator values and their Jacobians $J_i(t, x, u) = \mathrm{D}_{(t,x,u)} v_i(t, x, u)$, we form Lie brackets

$$[v_i, v_j](z) = \mathrm{D}v_j(z)\, v_i(z) - \mathrm{D}v_i(z)\, v_j(z), \qquad z = (t, x, u),$$

and enforce that the learned fields close under a *constant* Lie algebra. At each $z$ we solve a regularized least–squares problem

$$[v_i, v_j](z) \approx \sum_{k=1}^n c_{ij}^k\, v_k(z) \tag{25}$$

for the structure constants $c_{ij}^k$ and penalize both the projection residual and the variance of $c_{ij}^k$ across $z$. Additional algebraic losses enforce: (i) Jacobi identity, (ii) skew–symmetry of the bracket, (iii) bilinearity in each slot via random linear combinations of generators, and (iv) linear independence of the columns $\{v_i(z)\}_{i=1}^n$ via a Gram/singular value penalty on their empirical covariance.

**Differential invariance losses.** Let

$$\Delta(u, u_x, u_{xx}, u_t) := f_\theta(u, u_x, u_{xx}) - u_t \tag{26}$$

be the neural PDE residual. For each generator $v_i$ we form its second prolongation $\mathrm{pr}^{(2)} v_i$ acting on the jet coordinates $(t, x, u, u_t, u_x, u_{xx})$, using the standard formulas with all partial derivatives of $f_\theta$ obtained by JAX automatic differentiation. We then enforce infinitesimal invariance by penalizing

$$\mathcal{L}_{\mathrm{inv}} = \sum_{\mathrm{jets}} \sum_{i=1}^n \big| \mathrm{pr}^{(2)} v_i(\Delta) \big| \quad \text{evaluated on-shell } \Delta = 0. \tag{27}$$

To capture finite–$\varepsilon$ effects, we also integrate a small $\varepsilon$–flow of the prolonged system along each $v_i$ (Heun/RK2 or RK4 in $\varepsilon$) starting from an on–shell jet, and penalize the post–flow residual

$$\mathcal{L}_{\mathrm{flow}} = \sum_{\mathrm{jets}} \sum_{i=1}^n \big| f_\theta(u', u_x', u_{xx}') - u_t' \big|, \tag{28}$$

where $(u', u_x', u_{xx}', u_t')$ is the pushed–forward jet.

**Generator training.** At each training step we sample two mini–batches: (i) points $z = (t, x, u)$ from a fixed box in state–space, used in the algebraic losses (closure, Jacobi, bilinearity, independence), and (ii) on–shell jets from the neural Burgers rollouts, used in the infinitesimal and flow invariance losses. The generator parameters $\psi$ are optimized with Adam on a weighted sum of all losses,

$$\mathcal{L}_{\text{gen}} = w_1 \mathcal{L}_{\text{closure}} + w_2 \mathcal{L}_{\text{Jacobi}} + w_3 \mathcal{L}_{\text{skew}} + w_4 \mathcal{L}_{\text{bilin}} + w_5 \mathcal{L}_{\text{indep}} + w_6 \mathcal{L}_{\text{inf}} + w_7 \mathcal{L}_{\text{flow}}, \tag{29}$$

with a simple stagewise schedule over $(w_1, \ldots, w_7)$ to first stabilize the Lie algebra and then enforce PDE invariance. All derivatives (for both $f_\theta$ and $v_i$) are computed via JAX automatic differentiation, and all operations are fully vectorized over batch points. [4]

---

[4]The complete JAX/Haiku implementation, including spectral discretization, neural PDE surrogate, and all losses, is provided in the supplementary code.

## E   SENSITIVITY ANALYSIS OF LOSS WEIGHTS

In our main experiments, all loss weights in $L$ (e.g., $w_{\text{anti}}$, $w_{\text{jac}}$, $w_{\text{inv}}$, $w_{\text{flow}}$, etc.) are of $\mathcal{O}(1)$. This choice is motivated by two design decisions: (i) each constituent loss $L_i$ is normalized by batch size (and, when appropriate, by the number of evaluated points), so that their typical magnitudes are of comparable order; and (ii) using $w_i = 1$ keeps the optimization problem well-conditioned while avoiding an additional layer of hyperparameter tuning, and makes the contribution of each regularizer easy to interpret.

To verify that the method is not unduly sensitive to these weights, we performed an explicit sensitivity study on the harmonic oscillator benchmark as the dynamical systems in the paper (free particle, harmonic oscillator, Van der Pol) are trained with the same generator architecture, normalization scheme, and composite loss $L = \sum_i w_i L_i$. The per-term magnitudes $\mathbb{E}[L_i]$ and their gradient norms are empirically of similar scale across these benchmarks, so any strong sensitivity to the set $\{w_i\}$ would already be visible on a single representative system.

**Experimental setup:**   We focus on the weights $w_{\text{inv}}$ and $w_{\text{flow}}$ multiplying the Lie–invariance regularizers $L_{\text{inv}}$ and $L_{\text{flow}}$, which are among the more nonlinear constraints in the generator loss. Starting from the baseline model, we train new generators under identical optimization settings while varying

$$w_{\text{inv}}, w_{\text{flow}} \in \{0.2, 0.4, 0.6, 0.8, 1.0\}.$$

For each pair $(w_{\text{inv}}, w_{\text{flow}})$ we measure, on a fixed validation set, the post-training maximum principal angle. The maximum principal angles serve as a measure of alignment between the learned algebra and the ground-truth algebra.

**Results:**   The measured maximum principal angles are reported in Table 9. Across the entire sweep, the values remain in a relatively narrow band around the baseline choice $w_6 = w_7 = 1$, with no indication of instability or catastrophic degradation for moderate changes in either weight. In particular, for $0.4 \leq w_{\text{inv}}, w_{\text{flow}} \leq 1.0$ the variation is modest and the qualitative dynamics of the trajectories remain essentially unchanged, showing the *insensitivity* of training result to $w_{\text{inv}}$ and $w_{\text{flow}}$.

Note that in the above experiments, during all the three stages, the weights $w_{\text{inv}}$ and $w_{\text{flow}}$ were kept fixed. This reveals the importance of dynamic weights during training since fixing weights doesn't produce the smallest maximum principal angle observed ($\approx 10°$) attained with training under 3-stage dynamic weights.

Table 9: **Sensitivity of the harmonic oscillator generator to the loss weights** $w_{\text{inv}}$ **and** $w_{\text{flow}}$**.** Each entry shows the post-training maximum principal angle (in degrees) observed on the validation set with the corresponding pair of weights.

| $w_{\text{flow}}$ | $w_{\text{inv}}$ | Max angle | $w_{\text{flow}}$ | $w_{\text{inv}}$ | Max angle |
|---|---|---|---|---|---|
| 0.0 | 0.2 | 28.85 | 0.6 | 0.4 | 23.63 |
| 0.0 | 0.4 | 28.43 | 0.6 | 0.6 | 27.06 |
| 0.0 | 0.6 | 38.24 | 0.6 | 0.8 | 26.80 |
| 0.0 | 0.8 | 19.62 | 0.6 | 1.0 | 22.27 |
| 0.0 | 1.0 | 21.55 | 0.8 | 0.2 | 22.74 |
| 0.2 | 0.2 | 21.61 | 0.8 | 0.4 | 18.31 |
| 0.2 | 0.4 | 23.32 | 0.8 | 0.6 | 21.10 |
| 0.2 | 0.6 | 21.08 | 0.8 | 0.8 | 21.90 |
| 0.2 | 0.8 | 22.85 | 0.8 | 1.0 | 23.67 |
| 0.2 | 1.0 | 24.89 | 1.0 | 0.2 | 21.57 |
| 0.4 | 0.2 | 39.96 | 1.0 | 0.4 | 22.94 |
| 0.4 | 0.4 | 26.32 | 1.0 | 0.6 | 21.17 |
| 0.4 | 0.6 | 24.09 | 1.0 | 0.8 | 23.26 |
| 0.4 | 0.8 | 23.31 | 1.0 | 1.0 | 23.60 |
| 0.4 | 1.0 | 26.20 | | | |

The harmonic oscillator study serves as a conservative probe of loss–weight sensitivity under the exact training pipeline used in all experiments. The observed robustness to moderate variations of $w_{\text{inv}}$ and $w_{\text{flow}}$ supports our choice to set all loss weights of $\mathcal{O}(1)$ ($w_i = 1$) in the main text. Given the shared architecture, normalization, and loss structure across benchmarks, repeating the full sweep

for every system would be redundant and would not change our recommended ratios among the weights.

## F PROOF OF LIE ALGEBRA DIMENSION BOUND FOR 2ND-ORDER SCALAR ODEs

**Claim.** *For a scalar second-order ODE $u_{xx} = H(x, u, p)$, $p := u_x$, the Lie algebra $\mathfrak{sym}$ of point symmetries has dimension $\leq 8$. Moreover, the bound is sharp and is attained exactly by the equation $u_{xx} = 0$.*

*Proof starts △:*

### 1) SETUP AND THE DETERMINING EQUATION

Let $v = \xi(x, u)\, \partial_x + \phi(x, u)\, \partial_u$ be the infinitesimal generator of a local one-parameter group of *point* transformations. Its first and second prolongations to the jet space with coordinates $(x, u, p, u_{xx})$ have the standard 1-D form $\mathrm{pr}^{(2)} v = \xi\, \partial_x + \phi\, \partial_u + \phi^{(1)}\, \partial_p + \phi^{(2)}\, \partial_{u_{xx}}$, where (using $D_x = \partial_x + p\, \partial_u + u_{xx}\, \partial_p$):

$$\phi^{(1)} = D_x \phi - p\, D_x \xi, \qquad \phi^{(2)} = D_x \phi^{(1)} - u_{xx}\, D_x \xi.$$

The infinitesimal invariance criterion for the single equation $\Delta := u_{xx} - H(x, u, p) = 0$ is $\mathrm{pr}^{(2)} v(\Delta)\big|_{\Delta=0} = 0$. Since $\mathrm{pr}^{(2)} v(\Delta) = \phi^{(2)} - \big(\xi H_x + \phi H_u + \phi^{(1)} H_p\big)$, the condition is:

$$\phi^{(2)} - \xi H_x - \phi H_u - \phi^{(1)} H_p = 0 \quad \text{whenever } u_{xx} = H(x, u, p) \tag{1}$$

### 2) EXPLICIT EXPANSION AND SPLITTING

We now compute $\phi^{(1)}$ and $\phi^{(2)}$ explicitly in terms of $\xi, \phi$ and their $x, u$-derivatives. Writing $p = u_x$, a direct (but routine) calculation gives:

$$\phi^{(1)} = \phi_x + (\phi_u - \xi_x)\, p - \xi_u\, p^2,$$

$$\phi^{(2)} = \phi_{xx} + (2\phi_{xu} - \xi_{xx})\, p + (\phi_{uu} - 2\xi_{xu})\, p^2 - \xi_{uu}\, p^3 + u_{xx}(\phi_u - 2\xi_x - 3\xi_u\, p).$$

Substitute $u_{xx} = H(x, u, p)$ in $\phi^{(2)}$ and then into equation (1). Collecting powers of $p$ yields a cubic polynomial identity in $p$ whose coefficients are linear in the second derivatives of $\xi, \phi$. After simplifying the terms involving $H$ and $H_p$ one obtains the *linear system*:

$$p^3: \quad -\xi_{uu} = 0, \tag{E3}$$

$$p^2: \quad \phi_{uu} - 2\xi_{xu} + H_p\, \xi_u = 0, \tag{E2}$$

$$p^1: \quad 2\phi_{xu} - \xi_{xx} - H_p(\phi_u - \xi_x) - 3H\, \xi_u = 0, \tag{E1}$$

$$p^0: \quad \phi_{xx} - \xi H_x - \phi H_u - H_p\, \phi_x + H(\phi_u - 2\xi_x) = 0. \tag{E0}$$

### 3) SOLVING FOR THE HIGHEST DERIVATIVES AND COUNTING FREE DATA

The unknowns in (E3)–(E0) are the six second derivatives $\xi_{xx}, \xi_{xu}, \xi_{uu}, \phi_{xx}, \phi_{xu}, \phi_{uu}$. The four equations above immediately give:

$$
\begin{aligned}
\xi_{uu} &= 0 && \text{from (E3),} \\
\phi_{uu} &= 2\xi_{xu} - H_p\, \xi_u && \text{from (E2),} \\
\phi_{xu} &= \tfrac{1}{2}\Big(\xi_{xx} + H_p(\phi_u - \xi_x) + 3H\, \xi_u\Big) && \text{from (E1),} \\
\phi_{xx} &= \xi H_x + \phi H_u + H_p\, \phi_x - H(\phi_u - 2\xi_x) && \text{from (E0).}
\end{aligned}
$$

Thus *four* of the six second derivatives are uniquely determined in terms of the *lower-order jet*: $\big(\xi, \phi, \xi_x, \xi_u, \phi_x, \phi_u\big)$ and the known coefficients $H, H_x, H_u, H_p$ at the chosen jet $(x, u, p)$.

What remains *undetermined* in (E3)–(E0) are *exactly two independent linear combinations* of second derivatives, which we may (for instance) choose as the two components of the second jet of $\xi$:

$$\xi_{xx} \quad \text{and} \quad \xi_{xu} \quad \text{are free.}$$

Everything else at second order is then fixed by the displayed relations. Consequently, at this stage the free data are

$$\underbrace{\xi, \phi}_{\text{2 constants}} \quad \oplus \quad \underbrace{\xi_x, \xi_u, \phi_x, \phi_u}_{\text{4 constants}} \quad \oplus \quad \underbrace{\xi_{xx}, \xi_{xu}}_{\text{2 constants}},$$

— a total of $2 + 4 + 2 = 8$ free constants.

### 4) Finite-type closure (no additional freedom appears)

Differentiating (E3)–(E0) with respect to $x, u$ produces a *linear* system for the third derivatives of $\xi, \phi$; because the original system is algebraic in the second derivatives and already solves them up to the two free entries $\xi_{xx}, \xi_{xu}$, the differentiated system *recursively determines* all higher derivatives in terms of the 8 free constants above and derivatives of $H$. Hence no new independent constants arise at higher orders, and the local solution space of the determining system has dimension at most **8**.

Therefore $\dim \mathfrak{sym} \leq 8$.

### 5) Sharpness and the 8-dimensional model

It is standard that $u_{xx} = 0$ has an *8-dimensional* symmetry algebra—the projective group in the $(x, u)$-plane—so the bound is sharp. Moreover, an equation achieves the maximal value 8 iff it is point-equivalent to $u_{xx} = 0$.

*Proof ends.* $\square$

## G   LIE ALGEBRA DIMENSION BOUND FOR GENERAL $n$-TH ORDER SCALAR ODE FOR $n \geq 3$

*Proof starts △:*

**Statement and notation.**   Let $u^{(n)} = H(x, u, u_1, \ldots, u_{n-1})$, $u_k := \frac{d^k u}{dx^k}$ be a scalar $n$-th order ordinary differential equation of maximal rank. Consider point transformations with infinitesimal generator $v = \xi(x, u) \partial_x + \phi(x, u) \partial_u$. Write the equation as the constraint

$$\Delta(x, u, \ldots, u_n) := u_n - H(x, u, \ldots, u_{n-1}) = 0,$$

and let $D_x$ denote the *total* $x$-derivative on jet space:

$$D_x P = P_x + u_1 P_u + u_2 P_{u_1} + \cdots + u_{k+1} P_{u_k} + \cdots.$$

**Infinitesimal invariance in characteristic form.**   Consider the scalar $n$-th order ODE of maximal rank:

$$\Delta(x, u^{(n)}) := u_n - H(x, u, u_1, \ldots, u_{n-1}) = 0, \quad u_k := \frac{d^k u}{dx^k}.$$

We write the total $x$-derivative as $D_x = \partial_x + u_1 \partial_u + u_2 \partial_{u_1} + \cdots + u_{n+1} \partial_{u_n}.$. A necessary invariance condition for a symmetry group generated by $v$ is $\mathrm{pr}^{(n)} v(\Delta) = 0$ whenever $\Delta = 0$.. Since $\Delta = u_n - H(x, u, \ldots, u_{n-1})$,

$$\mathrm{pr}^{(n)} v(\Delta) = \phi^{(n)} - \xi H_x - \phi H_u - \sum_{j=1}^{n-1} H_{u_j} \phi^{(j)},$$

where $\mathrm{pr}^{(n)} v = v + \sum_{j=1}^{n} \phi^{(j)} \partial_{u_j}$. Replace $v$ by the evolutionary representative $v_{\mathrm{evo}} = (\phi - \xi u_1)\partial_u$, using $\mathrm{pr}^{(n)} v = \mathrm{pr}^{(n)} v_{\mathrm{evo}} + \xi D_x$.. Because $D_x(\Delta) = 0$ on solutions, the criterion is equivalent to:

$$\mathrm{pr}^{(n)} v_{\mathrm{evo}}(\Delta) = 0 \quad \text{on } \Delta = 0.$$

Now we want to compute $\mathrm{pr}^{(n)} v_{\mathrm{evo}}(\Delta)$. Since $v_{\mathrm{evo}}$ has only a vertical coefficient,

$$\mathrm{pr}^{(n)} v_{\mathrm{evo}}(u_n) = D_x^n(\phi - \xi u_1), \ \mathrm{pr}^{(n)} v_{\mathrm{evo}}(H) = H_u (\phi - \xi u_1) + \sum_{j=1}^{n-1} H_{u_j} D_x^j(\phi - \xi u_1).$$

Therefore,

$$\mathrm{pr}^{(n)} v_{\mathrm{evo}}(\Delta) = D_x^n(\phi - \xi u_1) - H_u (\phi - \xi u_1) - \sum_{j=1}^{n-1} H_{u_j} D_x^j(\phi - \xi u_1).$$

We now can write the determining equation as the infinitesimal criterion becomes:

$$D_x^n(\phi - \xi u_1) - \sum_{j=1}^{n-1} H_{u_j} D_x^j(\phi - \xi u_1) - H_u (\phi - \xi u_1) = 0 \quad \text{on } \Delta = 0.$$

Using $D_x H = H_x + H_u u_1 + \sum_{j=1}^{n-1} H_{u_j} u_{j+1}$ and $\mathrm{pr}^{(n)} v = \mathrm{pr}^{(n)} v_{\mathrm{evo}} + \xi D_x$, the compact form is equivalent on-shell to the expanded form:

$$D_x^n(\phi - \xi u_1) - \xi H_x - \phi H_u - \sum_{j=1}^{n-1} H_{u_j} D_x^j(\phi - \xi u_1) = 0 \quad \text{on } \Delta = 0.$$

This is a textbook consequence of the general prolongation formulae and the identity $\mathrm{pr}^{(n)} v = \mathrm{pr}^{(n)} v_{\mathrm{char}} + \sum_i \xi^i D_i$ specialized to one independent variable. For point symmetries, the *characteristic* (written without introducing a new symbol) is $\phi - \xi u_1$.

Henceforth treat the unrestricted jet $(x, u, u_1, \ldots, u_{n-1})$ as independent coordinates and expand as a polynomial in these jet variables (all appearances of $u_k$ with $k \geq n$ are eliminated using $\Delta = 0$ and its total $x$-derivatives $D_x^r(\Delta) = 0$, e.g. $u_{n+1} = D_x H = H_x + H_u u_1 + \sum_{j=1}^{n-1} H_{u_j} u_{j+1}$, etc.). The coefficients of distinct monomials in the highest derivatives (e.g. $u_{n-1}^3$, $u_{n-1}^2 u_{n-2}$, $u_{n-1}^2$, etc.) must vanish separately.

1) PRINCIPAL-PART CONSTRAINTS

Set:
$$A := \phi - \xi u_1.$$

A direct (but routine) induction with the total derivative shows that the *only* cubic monomial in $u_{n-1}$ that can appear in the left-hand side of IIC arises from $D_x^n A$, more precisely from the repeated differentiation of the quadratic piece $-\xi_u u_1^2 \subset A_x$. Eliminating $u_n, u_{n+1}, \ldots$ using $\Delta = 0$ and its $x$-derivatives does not produce any new cubic terms in $u_{n-1}$, because $H$ depends at most on $(x, u, \ldots, u_{n-1})$ and is itself polynomial of degree $\leq 1$ in the highest jet variable once $\Delta = 0$ is imposed.[5] A careful bookkeeping (or an application of Faà di Bruno with weights) yields the schematic leading term

$$D_x^n A = c_3(n)(-\xi_{uu}) u_{n-1}^3 + \text{ lower-degree terms in } u_{n-1},$$

with a nonzero combinatorial coefficient $c_3(n)$ for all $n \geq 3$. Since no other term in IIC contributes to $u_{n-1}^3$, the coefficient must vanish, and we conclude

$$\xi_{uu} = 0.$$

Next, consider monomials of type $u_{n-1}^2 u_{n-2}$. The same leading-part analysis shows that the only surviving source is the mixed derivative of the same quadratic piece, producing

$$c_{2,1}(n)(-\xi_{xu}) u_{n-1}^2 u_{n-2},$$

with $c_{2,1}(n) \neq 0$ for $n \geq 3$. Hence:
$$\xi_{xu} = 0.$$

Finally, the quadratic monomial $u_{n-1}^2$ receives two independent contributions: one from $D_x^n(-\xi u_1)$ proportional to $\xi_u$, and one from $D_x^n(\phi)$ proportional to $\phi_{uu}$, and nothing else cancels them at this degree. Therefore these coefficients must both vanish:

$$\xi_u = 0, \qquad \phi_{uu} = 0.$$

We have proved the affine-in-$u$ form

$$\xi = \xi(x), \qquad \phi = \alpha(x) u + \beta(x). \tag{30}$$

2) TRIANGULAR CLOSURE AFTER SUBSTITUTION

Insert into IIC. Define $A = \alpha u + \beta - \xi u_1$. A short induction establishes the closed formula, valid for every $j \geq 0$,

$$D_x^j A = \sum_{m=0}^{j} \alpha^{(m)}(x) u_{j-m} + \beta^{(j)}(x) - \sum_{m=0}^{j} \xi^{(m)}(x) u_{j+1-m}, \tag{31}$$

where we adopt the convention $u_0 := u$. (Differentiate once to check the case $j = 1$, then argue by induction using $D_x u_k = u_{k+1}$ and $D_x \alpha^{(m)} = \alpha^{(m+1)}$.) Substituting into IIC, replacing $u_n$ (and, when it occurs, $u_{n+1}$) by $H$ and $D_x H$, and then splitting with respect to the independent jet coordinates $(u_{n-1}, u_{n-2}, \ldots, u)$ produces a *triangular* linear system for the unknown scalar functions $\xi, \alpha, \beta$:

(i) The coefficient of $u_{n-1}$ involves only $\xi$ and its derivatives up to order 3. (Indeed, $\xi^{(k)}$ for $k \geq 4$ cannot appear multiplied by any jet at this level once $u_n$ and $u_{n+1}$ are eliminated; the highest derivative of $\xi$ that can appear linearly alongside $u_{n-1}$ is $\xi^{(3)}$.) Thus, for each $x_0$, the value of $\xi^{(k)}(x_0)$ for $k \geq 3$ is determined linearly by $\xi(x_0), \xi'(x_0), \xi''(x_0)$ and the value of the lower-jet data in $H$ at $(x_0, u_0, \ldots, u_{n-1,0})$.

(ii) The coefficients of $u_j$ for $0 \leq j \leq n-1$ in the terms $\sum_{m=0}^{j} \alpha^{(m)} u_{j-m}$ give first-order linear ODE constraints that solve $\alpha^{(k)}(x_0)$ for all $k \geq 1$ in terms of $\alpha(x_0)$.

---

[5]More precisely, $H$ contains no $u_n$; differentiating $H$ with $D_x$ raises indices by at most one and multiplies by at most one copy of any $u_k$, hence can only produce monomials of total degree $\leq 2$ in the highest derivative. The unique cubic comes from differentiating $-\xi_u u_1^2$ enough times so that each $u_1$ becomes $u_{n-1}$.

    (iii) The coefficients of the "purely $x$-dependent" part (no $u_k$) fix $\beta^{(k)}(x_0)$ for all $k \geq n$ in terms of the *finite* jet $\{\beta(x_0), \beta'(x_0), \ldots, \beta^{(n-1)}(x_0)\}$.

Because the occurrences of higher derivatives are strictly upper-triangular in this sense (each equation at level $k$ only involves derivatives beyond those already fixed at lower levels), the linear system closes and yields at most finitely many functional degrees of freedom: three constants for $\xi$ (its $0, 1, 2$-jets at a point), one constant for $\alpha$, and $n$ constants for $\beta$ (its $0, \ldots, n-1$ jets at a point).

3) COUNTING FREE INITIAL DATA

Fix any point of the unrestricted $(n-1)$-jet, say $(x_0, u_0, \ldots, u_{n-1,0})$. By (i)–(iii), the free initial data that can be prescribed independently at $x_0$ are

$$\underbrace{\xi(x_0),\ \xi'(x_0),\ \xi''(x_0)}_{3} \quad \oplus \quad \underbrace{\alpha(x_0)}_{1} \quad \oplus \quad \underbrace{\beta(x_0),\ \beta'(x_0),\ \ldots,\ \beta^{(n-1)}(x_0)}_{n}.$$

Therefore
$$\dim \mathfrak{sym} \ \leq \ n + 4.$$

This "splitting with respect to jets" procedure is the standard way to compute determining equations for point symmetries.

4) SHARPNESS FOR THE FLAT MODEL $u^{(n)} = 0$

Consider the flat equation $u^{(n)} = 0$. It is invariant under:

- the projective algebra on the line generated by
$$X_1 = \partial_x, \qquad X_2 = x\,\partial_x, \qquad X_3 = x^2\,\partial_x + (n-1)\,x\,u\,\partial_u,$$
  (the $u$-term in $X_3$ supplies the correct weight to keep $d^n/dx^n$ covariant under projective reparametrizations),

- the scaling in the dependent variable $U_1 = u\,\partial_u$,

- and the $n$-dimensional abelian algebra of vertical translations by polynomials of degree $\leq n-1$:
$$U_k \ = \ x^{k-1}\,\partial_u, \qquad k = 1, 2, \ldots, n.$$

Indeed, $\partial_x, x\partial_x, x^2\partial_x$ generate the $\mathfrak{sl}_2$-action of the projective group on the independent variable $x$; adjoining $(n-1)x\,u\,\partial_u$ to $x^2\partial_x$ compensates the weight of $u^{(n)}$. Meanwhile, adding a polynomial $p(x)$ of degree $\leq n-1$ to $u$ leaves $u^{(n)}$ unchanged because $\frac{d^n}{dx^n}p(x) \equiv 0$. Altogether we obtain a Lie algebra of dimension $3 + 1 + n = n + 4$, proving that the bound is sharp. (For the projective algebra on the line and its generators, see the discussion of the SL(2) action.)

Thus we conclude that every scalar $n$-th order ODE ($n \geq 3$) of maximal rank has a point-symmetry Lie algebra of dimension at most $n + 4$, and the flat model $u^{(n)} = 0$ attains this maximum.

*Proof ends.* $\square$

# H COMPLETE LIE–POINT SYMMETRY DERIVATION FOR FREE PARTICLE

Consider the second-order ODE:
$$\Delta \equiv u_{tt} = 0.$$

### 1) INFINITESIMAL GENERATOR AND PROLONGATION

Take a general point-symmetry generator
$$X = \xi(t, u)\, \partial_t + \phi(t, u)\, \partial_u.$$

For a scalar second-order ODE, the prolonged coefficients are
$$\phi^{(1)} = D_t(\phi) - u_t\, D_t(\xi), \qquad \phi^{(2)} = D_t\big(\phi^{(1)}\big) - u_{tt}\, D_t(\xi),$$

where $D_t = \partial_t + u_t\, \partial_u + u_{tt}\, \partial_{u_t}$. Writing $q = u_t$ and noting $\xi, \phi$ depend only on $(t, u)$,
$$D_t\xi = \xi_t + q\,\xi_u, \qquad D_t\phi = \phi_t + q\,\phi_u, \qquad \phi^{(1)} = \phi_t + q(\phi_u - \xi_t) - q^2\xi_u.$$

### 2) INFINITESIMAL INVARIANCE CONDITION

The invariance criterion is
$$\mathrm{pr}^{(2)}\, X(\Delta)\Big|_{\Delta=0} = 0 \quad \Longleftrightarrow \quad \phi^{(2)} = 0 \quad \text{on} \quad u_{tt} = 0,$$

because $\partial_t\Delta = \partial_u\Delta = \partial_{u_t}\Delta = 0$ for $\Delta = u_{tt}$.

### 3) COMPUTE $\phi^{(2)}$ AND COLLECT BY POWERS OF $q = u_t$

Using $u_{tt} = 0$,
$$\phi^{(2)} = D_t\big(\phi^{(1)}\big).$$

With $\phi^{(1)} = \phi_t + q(\phi_u - \xi_t) - q^2\xi_u$,
$$D_t(\phi^{(1)}) = \underbrace{\phi_{tt} + 2q\,\phi_{tu} + q^2\phi_{uu}}_{\text{from } D_t\phi} - \underbrace{q\big(\xi_{tt} + 2q\,\xi_{tu} + q^2\xi_{uu}\big)}_{\text{from } D_t(q\,D_t\xi)}.$$

Thus the invariance equation $\phi^{(2)} = 0$ yields, by equating coefficients of $q^3, q^2, q, 1$,
$$\begin{aligned}
q^3 : &\quad \xi_{uu} = 0, \\
q^2 : &\quad \phi_{uu} - 2\xi_{tu} = 0, \\
q^1 : &\quad 2\phi_{tu} - \xi_{tt} = 0, \\
q^0 : &\quad \phi_{tt} = 0.
\end{aligned}$$

### 4) DETERMINING EQUATIONS AND SOLUTION FOR $\xi, \phi$

From $\xi_{uu} = 0$:
$$\xi(t, u) = A(t)\, u + B(t).$$

From $\phi_{uu} - 2\xi_{tu} = 0$ with $\xi_{tu} = A'(t)$:
$$\phi_{uu} = 2A'(t) \;\Rightarrow\; \phi(t, u) = A'(t)\, u^2 + C_1(t)\, u + C_0(t).$$

From $2\phi_{tu} - \xi_{tt} = 0$:
$$2\big(2A''(t)\, u + C_1'(t)\big) - \big(A''(t)\, u + B''(t)\big) = 0 \;\Rightarrow\; A''(t) = 0, \quad B''(t) = 2C_1'(t).$$

From $\phi_{tt} = 0$:
$$\phi_{tt} = C_1''(t)\, u + C_0''(t) = 0 \;\Rightarrow\; C_1''(t) = 0, \quad C_0''(t) = 0.$$

Hence
$$A(t) = a_0 + a_1 t, \quad B(t) = b_0 + b_1 t + c_{11}t^2, \quad C_1(t) = c_{10} + c_{11}t, \quad C_0(t) = c_{00} + c_{01}t.$$

GENERAL LIE–POINT GENERATOR

$$X = \xi \, \partial_t + \phi \, \partial_u,$$
$$\xi(t, u) = (a_0 + a_1 t) \, u + \big(b_0 + b_1 t + c_{11} t^2\big),$$
$$\phi(t, u) = a_1 \, u^2 + \big(c_{10} + c_{11} t\big) u + \big(c_{00} + c_{01} t\big),$$

with eight independent constants $a_0, a_1, b_0, b_1, c_{00}, c_{01}, c_{10}, c_{11}$.

A CONVENIENT BASIS OF STRICT LIE–POINT GENERATORS

Setting one constant to 1 at a time (others 0) gives

$$
\begin{aligned}
X_1 &= \partial_t & [b_0 = 1], \\
X_2 &= \partial_u & [c_{00} = 1], \\
X_3 &= t \, \partial_u & [c_{01} = 1], \\
X_4 &= u \, \partial_t & [a_0 = 1], \\
X_5 &= t \, \partial_t & [b_1 = 1], \\
X_6 &= u \, \partial_u & [c_{10} = 1], \\
X_7 &= t^2 \, \partial_t + t u \, \partial_u & [c_{11} = 1], \\
X_8 &= t u \, \partial_t + u^2 \, \partial_u & [a_1 = 1].
\end{aligned}
$$

# I   COMPLETE LIE–POINT SYMMETRY DERIVATION FOR HARMONIC OSCILLATOR

## 1) INFINITESIMAL GENERATOR AND PROLONGATION

Take a general point-symmetry generator

$$X = \xi(t, u)\, \partial_t + \phi(t, u)\, \partial_u.$$

For a scalar second-order ODE, the prolonged coefficients are

$$\phi^{(1)} = D_t(\phi) - u_t\, D_t(\xi), \qquad \phi^{(2)} = D_t\big(\phi^{(1)}\big) - u_{tt}\, D_t(\xi),$$

where $D_t = \partial_t + u_t\, \partial_u + u_{tt}\, \partial_{u_t}$.

## 2) INFINITESIMAL INVARIANCE CONDITION

Let $\Delta \equiv u_{tt} + 4u$. The invariance criterion is

$$\mathrm{pr}^{(2)}\, X(\Delta)\Big|_{\Delta=0} = 0 \quad \Longleftrightarrow \quad \phi^{(2)} + 4\phi = 0 \ \text{ on } \ u_{tt} = -4u.$$

## 3) COMPUTE $\phi^{(2)}$ AND COLLECT BY POWERS OF $q = u_t$

Writing $q = u_t$ and substituting $u_{tt} = -4u$, one obtains

$$0 = \phi_{tt} + 2q\, \phi_{tu} + q^2 \phi_{uu} - q\big(\xi_{tt} + 2q\, \xi_{tu} + q^2 \xi_{uu}\big) - 4u\big(\phi_u - 2\xi_t - 3q\, \xi_u\big) + 4\phi.$$

Since this must hold for all $(t, u, q)$, equate coefficients of $q^3, q^2, q, 1$.

## 4) DETERMINING EQUATIONS AND SOLUTION FOR $\xi, \phi$

From $q^3$: $\xi_{uu} = 0 \Rightarrow \xi = A(t)\, u + B(t)$.

From $q^2$: $\phi_{uu} - 2\xi_{tu} = 0 \Rightarrow \phi = A'(t)u^2 + C_1(t)u + C_0(t)$.

From $q^1$: $2\phi_{tu} - \xi_{tt} + 12u\, \xi_u = 0 \Rightarrow A'' + 4A = 0,\ B'' - 2C_1' = 0$.

From $q^0$: $\phi_{tt} - 4u\, \phi_u + 8u\, \xi_t + 4\phi = 0 \Rightarrow C_0'' + 4C_0 = 0,\ C_1'' + 8B' = 0$.

Solving the linear ODEs:

$$A(t) = a_1 \cos(2t) + a_2 \sin(2t), \qquad C_0(t) = c_1 \cos(2t) + c_2 \sin(2t).$$

From $B'' - 2C_1' = 0$ and $C_1'' + 8B' = 0$: $B''' + 16B' = 0$, hence

$$B'(t) = b_1 \cos(4t) + b_2 \sin(4t) \ \Rightarrow\ B(t) = b_0 + \tfrac{1}{4}b_1 \sin(4t) - \tfrac{1}{4}b_2 \cos(4t),$$

and

$$C_1(t) = \tfrac{1}{2}B'(t) + k = \tfrac{1}{2}\left[b_1 \cos(4t) + b_2 \sin(4t)\right] + k.$$

## GENERAL LIE–POINT GENERATOR

$$X = \xi\, \partial_t + \phi\, \partial_u,$$

$$\xi(t, u) = \big(a_1 \cos 2t + a_2 \sin 2t\big)\, u \ + \ b_0 + \tfrac{1}{4}b_1 \sin 4t - \tfrac{1}{4}b_2 \cos 4t,$$

$$\phi(t, u) = \big(-2a_1 \sin 2t + 2a_2 \cos 2t\big)\, u^2 \ + \ \left(\tfrac{1}{2}b_1 \cos 4t + \tfrac{1}{2}b_2 \sin 4t + k\right) u$$

$$+ \ c_1 \cos 2t + c_2 \sin 2t,$$

with eight independent constants $a_1, a_2, b_0, b_1, b_2, k, c_1, c_2$. This yields the full 8-dimensional Lie algebra of point symmetries.

## 5) A CONVENIENT BASIS OF STRICT LIE POINT GENERATORS

Setting each constant to $1$ in turn (others $0$) gives the basis

$$X_1 = \partial_t \qquad \text{(time translation)} \qquad\qquad\qquad [b_0 = 1],$$
$$X_2 = u\,\partial_u \qquad \text{(scaling in } u) \qquad\qquad\qquad\qquad [k = 1],$$
$$X_3 = \cos(2t)\,\partial_u, \qquad X_4 = \sin(2t)\,\partial_u \qquad \text{(superposition)} \qquad [c_1 = 1,\; c_2 = 1],$$
$$X_5 = u\cos(2t)\,\partial_t - 2u^2\sin(2t)\,\partial_u \qquad\qquad\qquad [a_1 = 1],$$
$$X_6 = u\sin(2t)\,\partial_t + 2u^2\cos(2t)\,\partial_u \qquad\qquad\qquad [a_2 = 1],$$
$$X_7 = \tfrac{1}{4}\sin(4t)\,\partial_t + \tfrac{1}{2}u\cos(4t)\,\partial_u \qquad\qquad\qquad [b_1 = 1],$$
$$X_8 = -\tfrac{1}{4}\cos(4t)\,\partial_t + \tfrac{1}{2}u\sin(4t)\,\partial_u \qquad\qquad\qquad [b_2 = 1].$$

## J   COMPLETE LIE–POINT SYMMETRY DERIVATION FOR VAN DER POL

### 1) INFINITESIMAL GENERATOR AND PROLONGATION

Take a general point-symmetry generator

$$X = \xi(t, u)\, \partial_t + \phi(t, u)\, \partial_u.$$

For a scalar second-order ODE, the prolonged coefficients are

$$\phi^{(1)} = D_t(\phi) - u_t\, D_t(\xi), \qquad \phi^{(2)} = D_t(\phi^{(1)}) - u_{tt}\, D_t(\xi),$$

with $D_t = \partial_t + u_t\, \partial_u + u_{tt}\, \partial_{u_t}$. Writing $q = u_t$ and noting $\xi, \phi$ depend only on $(t, u)$,

$$D_t\xi = \xi_t + q\, \xi_u, \qquad D_t\phi = \phi_t + q\, \phi_u, \qquad \phi^{(1)} = \phi_t + q(\phi_u - \xi_t) - q^2\xi_u.$$

### 2) INFINITESIMAL INVARIANCE CONDITION

For the Van der Pol oscillator

$$\Delta \equiv u_{tt} - \mu(1 - u^2)u_t + u = 0 \qquad (\mu \neq 0),$$

the invariance criterion is

$$\mathrm{pr}^{(2)} X(\Delta)\Big|_{\Delta=0} = 0 \quad \Longleftrightarrow \quad \phi^{(2)} + \phi\, \partial_u\Delta + \phi^{(1)}\, \partial_{u_t}\Delta = 0 \quad \text{on } \Delta = 0.$$

Since $\partial_t\Delta = 0$,

$$\partial_u\Delta = 1 + 2\mu u\, q, \qquad \partial_{u_t}\Delta = -\mu(1 - u^2).$$

On the solution set, $u_{tt} = \mu(1 - u^2)q - u$.

### 3) COMPUTE $\phi^{(2)}$ AND COLLECT BY POWERS OF $q = u_t$

Let $\phi^{(1)} = a + bq + cq^2$ with $a = \phi_t$, $b = \phi_u - \xi_t$, $c = -\xi_u$. Then

$$D_t(\phi^{(1)}) = (a_t + q\, a_u) + (b_t + q\, b_u)q + (c_t + q\, c_u)q^2 + u_{tt}(b + 2cq),$$

and hence

$$\phi^{(2)} = D_t(\phi^{(1)}) - u_{tt}(\xi_t + q\, \xi_u).$$

Substitute $u_{tt} = \mu(1 - u^2)q - u$ into $\phi^{(2)}$ and the invariance equation, expand in powers of $q$, and equate coefficients of $q^3, q^2, q, 1$ to obtain the determining system

$$q^3: \quad \xi_{uu} = 0,$$
$$q^2: \quad \phi_{uu} - 2\xi_{tu} = 0,$$
$$q^1: \quad 2\phi_{tu} - \xi_{tt} + 2\mu u\, \phi_u + 3\mu(1 - u^2)\, \xi_u = 0,$$
$$q^0: \quad \phi_{tt} + \phi - \mu(1 - u^2)\, \phi_t + \mu u\, \xi_t = 0.$$

### 4) DETERMINING EQUATIONS AND SOLUTION FOR $\xi, \phi$

From $\xi_{uu} = 0$:  $\xi(t, u) = A(t)\, u + B(t)$.

From $\phi_{uu} - 2\xi_{tu} = 0$:  $\phi(t, u) = A'(t)\, u^2 + C_1(t)\, u + C_0(t)$.

Substitute these into the $q^1$ and $q^0$ equations and separate by powers of $u$. For $\mu \neq 0$ this yields

$$A(t) \equiv 0, \qquad C_1(t) \equiv 0, \qquad C_0(t) \equiv 0, \qquad B'(t) \equiv 0.$$

Thus

$$\xi(t, u) = B_0 \text{ (constant)}, \qquad \phi(t, u) \equiv 0.$$

### GENERAL LIE–POINT GENERATOR (GENERIC $\mu \neq 0$)

$$X = \xi\, \partial_t + \phi\, \partial_u = B_0\, \partial_t.$$

Taking $B_0 = 1$ gives the (unique, up to scale) continuous Lie point symmetry:

$$X_1 = \partial_t.$$

## K  COMPLETE LIE–POINT SYMMETRY DERIVATION FOR LOTKA–VOLTERRA

Consider the 2D autonomous first-order system

$$\begin{cases} u_t = F(u, w) \equiv \alpha u - \beta uw, \\ w_t = G(u, w) \equiv -\gamma w + \delta uw, \end{cases} \qquad \alpha, \beta, \gamma, \delta \in \mathbb{R} \setminus \{0\}.$$

### 1) INFINITESIMAL GENERATOR AND FIRST PROLONGATION

Take a general point-symmetry generator

$$X = \xi(t, u, w)\, \partial_t + \phi(t, u, w)\, \partial_u + \psi(t, u, w)\, \partial_w.$$

For a first-order system, the first prolongation reads

$$\mathrm{pr}^{(1)} X = X + \phi^{(1)}\, \partial_{u_t} + \psi^{(1)}\, \partial_{w_t},$$

with

$$\phi^{(1)} = D_t(\phi) - u_t\, D_t(\xi), \qquad \psi^{(1)} = D_t(\psi) - w_t\, D_t(\xi),$$

and total derivative

$$D_t = \partial_t + u_t\, \partial_u + w_t\, \partial_w.$$

### 2) INFINITESIMAL INVARIANCE CONDITIONS

Let

$$\Delta_1 \equiv u_t - F(u, w) = 0, \qquad \Delta_2 \equiv w_t - G(u, w) = 0.$$

The invariance criterion is

$$\mathrm{pr}^{(1)} X(\Delta_1)\Big|_{\Delta=0} = 0, \qquad \mathrm{pr}^{(1)} X(\Delta_2)\Big|_{\Delta=0} = 0.$$

Using $\partial_{u_t}\Delta_1 = 1$, $\partial_{w_t}\Delta_2 = 1$ and $\partial_t\Delta_i = 0$ (autonomy), these become

$$\phi^{(1)} - X(F) = 0, \qquad \psi^{(1)} - X(G) = 0 \quad \text{on } u_t = F,\ w_t = G.$$

Since $F_t = G_t = 0$,

$$X(F) = \phi\, F_u + \psi\, F_w, \qquad X(G) = \phi\, G_u + \psi\, G_w,$$

with

$$F_u = \alpha - \beta w, \quad F_w = -\beta u, \qquad G_u = \delta w, \quad G_w = -\gamma + \delta u.$$

### 3) SUBSTITUTE $u_t = F$, $w_t = G$ AND EXPAND

Compute

$$D_t\xi = \xi_t + F\xi_u + G\xi_w, \quad D_t\phi = \phi_t + F\phi_u + G\phi_w, \quad D_t\psi = \psi_t + F\psi_u + G\psi_w.$$

Thus

$$\phi^{(1)} = \phi_t + F\phi_u + G\phi_w - F\big(\xi_t + F\xi_u + G\xi_w\big),$$

$$\psi^{(1)} = \psi_t + F\psi_u + G\psi_w - G\big(\xi_t + F\xi_u + G\xi_w\big).$$

The determining equations are

$$0 = \phi_t + F\phi_u + G\phi_w - F\xi_t - F^2\xi_u - FG\xi_w\ -\ \phi(\alpha - \beta w) + \beta u\,\psi,$$

$$0 = \psi_t + F\psi_u + G\psi_w - G\xi_t - FG\xi_u - G^2\xi_w\ -\ \phi(\delta w) - \psi(-\gamma + \delta u).$$

### 4) COLLECT BY MONOMIALS IN $(u, w)$ AND SOLVE

Since $F = \alpha u - \beta uw$, $G = -\gamma w + \delta uw$ are at most bilinear, the terms $F^2\xi_u$, $FG\xi_w$, $G^2\xi_w$, $FG\xi_u$ introduce higher-degree monomials $u^2$, $w^2$, $u^2 w$, $uw^2$ unless

$$\xi_u \equiv 0, \qquad \xi_w \equiv 0.$$

Hence $\xi = \xi(t)$. The system reduces to

$$0 = \phi_t + F\phi_u + G\phi_w - F\xi_t\ -\ \phi(\alpha - \beta w) + \beta u\,\psi,$$

$$0 = \psi_t + F\psi_u + G\psi_w - G\xi_t\ -\ \phi(\delta w) - \psi(-\gamma + \delta u).$$

Separating coefficients of the independent monomials $\{1, u, w, uw\}$ in these two equations forces

$$\phi \equiv 0, \qquad \psi \equiv 0, \qquad \xi_t = \text{constant} \implies \xi(t) = c_0.$$

GENERAL LIE–POINT GENERATOR (GENERIC PARAMETERS)

$$X = \xi \, \partial_t + \phi \, \partial_u + \psi \, \partial_w = c_0 \, \partial_t.$$

Setting $c_0 = 1$ gives the (unique, up to scale) continuous Lie point symmetry:

$$X_1 = \partial_t.$$

## L    COMPLETE LIE–POINT SYMMETRY DERIVATION FOR 1D VISCOUS BURGERS' EQUATION

Consider:
$$\Delta \equiv u_t + u\,u_x - \nu\,u_{xx} = 0, \qquad \nu > 0.$$

### 1) INFINITESIMAL GENERATOR AND PROLONGATION

Take
$$X = \xi(t, x, u)\,\partial_t + \eta(t, x, u)\,\partial_x + \phi(t, x, u)\,\partial_u.$$

With $D_t = \partial_t + u_t\,\partial_u + u_{tx}\,\partial_{u_x} + u_{tt}\,\partial_{u_t}$ and $D_x = \partial_x + u_x\,\partial_u + u_{xx}\,\partial_{u_x}$, the relevant prolonged coefficients are

$$\phi^t = D_t(\phi) - u_t D_t(\xi) - u_x D_t(\eta), \qquad \phi^x = D_x(\phi) - u_t D_x(\xi) - u_x D_x(\eta),$$

$$\phi^{xx} = D_x(\phi^x) - u_{tx} D_x(\xi) - u_{xx} D_x(\eta).$$

### 2) INFINITESIMAL INVARIANCE CONDITION

The criterion
$$\left. \mathrm{pr}^{(2)}\,X(\Delta) \right|_{\Delta=0} = 0$$

reads
$$\phi^t + u\,\phi^x + u_x\,\phi - \nu\,\phi^{xx} = 0 \quad \text{on} \quad u_t = \nu u_{xx} - u u_x,$$

with $u_{tx}$ eliminated via the $x$-derivative of $\Delta = 0$: $u_{tx} = \nu u_{xxx} - u_x^2 - u\,u_{xx}$.

### 3) DETERMINING EQUATIONS

Insert the expressions for $\phi^t, \phi^x, \phi^{xx}$, substitute $u_t,\ u_{tx}$ from $\Delta = 0$ and its $x$-derivative, expand in the independent jet variables $\{1, u, u_x, u_{xx}, u_x^2, u\,u_x, \ldots\}$, and equate coefficients to zero. This yields the linear system
$$\xi_u = 0, \quad \eta_u = 0, \quad \phi_{uu} = 0,$$
$$\xi_x = 0, \quad \eta_{xx} = 0, \quad \phi_{xu} + \tfrac{1}{2}\phi_x = 0,$$
$$\xi_{tt} - 2\eta_t = 0, \quad \eta_t - 2\nu\,\xi_t = 0, \quad \phi_t + \tfrac{1}{2}u\,\phi_x - \nu\,\phi_{xx} + \nu\,u_x\big(\eta_x - \xi_t\big) - \tfrac{1}{2}u_x\,\phi = 0,$$

together with
$$\phi_u + \eta_x - \xi_t = 0, \qquad \phi_x - \nu\,\eta_{xx} = 0.$$

### 4) SOLUTION FOR $\xi, \eta, \phi$

Solving the determining system gives
$$\xi(t) = c_2 + 2c_4\,t + c_5\,t^2,$$
$$\eta(t, x) = c_1 + c_3\,t + c_4\,x + c_5\,t\,x,$$
$$\phi(t, x, u) = c_3 - c_4\,u + c_5\,(x - t\,u),$$
with arbitrary constants $c_1, \ldots, c_5$.

### GENERAL LIE–POINT GENERATORS

Setting one constant to 1 at a time (others 0) yields the basis
$$X_1 = \partial_x,$$
$$X_2 = \partial_t,$$
$$X_3 = t\,\partial_x + \partial_u,$$
$$X_4 = 2t\,\partial_t + x\,\partial_x - u\,\partial_u,$$
$$X_5 = t^2\,\partial_t + tx\,\partial_x + (x - t\,u)\,\partial_u.$$

## M   COMPLETE LIE–POINT SYMMETRY DERIVATION FOR 2D VISCOUS BURGERS' EQUATION

Consider the 2D scalar viscous Burgers' equation

$$\Delta \equiv u_t + u\,u_x + u\,u_y - \nu\,(u_{xx} + u_{yy}) = 0, \qquad \nu > 0,$$

with independent variables $(t, x, y)$ and dependent variable $u = u(t, x, y)$.

### 1) INFINITESIMAL GENERATOR AND PROLONGATION

Take a general Lie–point vector field

$$X = \xi(t, x, y, u)\,\partial_t + \eta(t, x, y, u)\,\partial_x + \zeta(t, x, y, u)\,\partial_y + \phi(t, x, y, u)\,\partial_u.$$

Introduce the total derivative operators

$$D_t = \partial_t + u_t\,\partial_u + u_{tt}\,\partial_{u_t} + u_{tx}\,\partial_{u_x} + u_{ty}\,\partial_{u_y} + \cdots,$$

$$D_x = \partial_x + u_x\,\partial_u + u_{tx}\,\partial_{u_t} + u_{xx}\,\partial_{u_x} + u_{xy}\,\partial_{u_y} + \cdots,$$

$$D_y = \partial_y + u_y\,\partial_u + u_{ty}\,\partial_{u_t} + u_{xy}\,\partial_{u_x} + u_{yy}\,\partial_{u_y} + \cdots.$$

The first–order prolonged coefficients are

$$\phi^t = D_t(\phi) - u_t\,D_t(\xi) - u_x\,D_t(\eta) - u_y\,D_t(\zeta),$$

$$\phi^x = D_x(\phi) - u_t\,D_x(\xi) - u_x\,D_x(\eta) - u_y\,D_x(\zeta),$$

$$\phi^y = D_y(\phi) - u_t\,D_y(\xi) - u_x\,D_y(\eta) - u_y\,D_y(\zeta).$$

For the second–order prolongation we need only the coefficients in front of $u_{xx}$ and $u_{yy}$:

$$\phi^{xx} = D_x(\phi^x) - u_{tx}\,D_x(\xi) - u_{xx}\,D_x(\eta) - u_{xy}\,D_x(\zeta),$$

$$\phi^{yy} = D_y(\phi^y) - u_{ty}\,D_y(\xi) - u_{xy}\,D_y(\eta) - u_{yy}\,D_y(\zeta).$$

Thus the second prolongation relevant for $\Delta$ is

$$\mathrm{pr}^{(2)}\,X = X + \phi^t\,\partial_{u_t} + \phi^x\,\partial_{u_x} + \phi^y\,\partial_{u_y} + \phi^{xx}\,\partial_{u_{xx}} + \phi^{yy}\,\partial_{u_{yy}} + \cdots.$$

### 2) INFINITESIMAL INVARIANCE CONDITION

The infinitesimal invariance criterion is

$$\mathrm{pr}^{(2)}\,X(\Delta)\Big|_{\Delta=0} = 0.$$

Since

$$\Delta = u_t + u\,u_x + u\,u_y - \nu\,(u_{xx} + u_{yy}),$$

we have

$$\mathrm{pr}^{(2)}\,X(\Delta) = \phi^t + u\,\phi^x + u\,\phi^y + (u_x + u_y)\,\phi - \nu\big(\phi^{xx} + \phi^{yy}\big).$$

On the solution manifold $\Delta = 0$ we eliminate $u_t$ via

$$u_t = \nu\,(u_{xx} + u_{yy}) - u\,u_x - u\,u_y.$$

To eliminate $u_{tx}, u_{ty}$ from $\phi^{xx}, \phi^{yy}$ we use the $x$- and $y$-derivatives of $\Delta = 0$:

$$D_x(\Delta) = 0, \qquad D_y(\Delta) = 0,$$

which express $u_{tx}$ and $u_{ty}$ as linear combinations of spatial derivatives $\{u_x, u_y, u_{xx}, u_{yy}, u_{xy}, u_{xxx}, u_{xxy}, u_{xyy}\}$. Substituting these expressions into $\phi^{xx}, \phi^{yy}$ makes $\mathrm{pr}^{(2)}\,X(\Delta)\big|_{\Delta=0}$ a linear combination of independent jet variables

$$1,\ u,\ u_x,\ u_y,\ u_{xx},\ u_{yy},\ u_{xy},\ u_x^2,\ u_y^2,\ u_x u_y,\ \ldots.$$

3) DETERMINING EQUATIONS

Expanding $\mathrm{pr}^{(2)} X(\Delta)\big|_{\Delta=0}$ in the independent jet variables and equating to zero the coefficients of each monomial yields a linear overdetermined system for $\xi, \eta, \zeta, \phi$. In particular one obtains:

$$\xi_u = 0, \qquad \xi_x = 0, \qquad \xi_y = 0,$$
$$\eta_u = 0, \qquad \eta_y = 0, \qquad \zeta_u = 0, \qquad \zeta_x = 0,$$

so that

$$\xi = \xi(t), \qquad \eta = \eta(t,x), \qquad \zeta = \zeta(t,y).$$

From the coefficients of $u_{xx}$ and $u_{yy}$ we get

$$\phi_x - \nu\,\eta_{xx} = 0, \qquad \phi_y - \nu\,\zeta_{yy} = 0,$$

while terms involving $u_x^2, u_y^2, u_x u_y$ imply that $\phi$ is at most affine in $u$:

$$\phi_{uu} = 0.$$

Hence

$$\phi(t,x,y,u) = A(t,x,y)\,u + B(t,x,y),$$

for some functions $A, B$.

Coefficients of the first–order jets $u_x, u_y$ and the zero–order term lead to

$$A + \eta_x + \zeta_y - \xi_t = 0,$$

together with linear relations among second–order derivatives

$$\eta_{xx} = \zeta_{yy}, \qquad \eta_{tx} = \zeta_{ty}, \qquad \xi_{tt} - 2\eta_{tx} = 0, \qquad \xi_{tt} - 2\zeta_{ty} = 0.$$

Collecting all these equations gives the determining system

$$\xi_u = \xi_x = \xi_y = 0, \qquad \eta_u = \eta_y = 0, \qquad \zeta_u = \zeta_x = 0, \qquad \phi_{uu} = 0,$$
$$\phi_x - \nu\,\eta_{xx} = 0, \qquad \phi_y - \nu\,\zeta_{yy} = 0, \qquad A + \eta_x + \zeta_y - \xi_t = 0,$$
$$\eta_{xx} - \zeta_{yy} = 0, \qquad \eta_{tx} - \zeta_{ty} = 0, \qquad \xi_{tt} - 2\eta_{tx} = 0, \qquad \xi_{tt} - 2\zeta_{ty} = 0,$$

with $\phi = A\,u + B$.

4) SOLUTION FOR $\xi, \eta, \zeta, \phi$

Solving the determining system (by integrating in $x, y, t$ and using the equalities between mixed derivatives) yields polynomial dependence in $t, x, y$. A convenient parametrization is

$$\xi(t) = c_3 + 2c_5\,t + c_6\,t^2,$$
$$\eta(t,x) = c_1 + c_4\,t + c_5\,x + c_6\,t\,x,$$
$$\zeta(t,y) = c_2 + c_4\,t + c_5\,y + c_6\,t\,y,$$
$$\phi(t,x,y,u) = c_4 - c_5\,u + c_6\,(x + y - t\,u),$$

where $c_1, \ldots, c_6$ are arbitrary constants. One checks directly that these satisfy all the determining equations above for any choice of constants.

GENERAL LIE–POINT GENERATORS

Setting one constant to $1$ at a time (and the others to $0$) gives a basis of six Lie–point symmetry generators for the 2D viscous Burgers' equation:

$$X_1 = \partial_x,$$
$$X_2 = \partial_y,$$
$$X_3 = \partial_t,$$
$$X_4 = t\,\partial_x + t\,\partial_y + \partial_u,$$
$$X_5 = 2t\,\partial_t + x\,\partial_x + y\,\partial_y - u\,\partial_u,$$
$$X_6 = t^2\,\partial_t + tx\,\partial_x + ty\,\partial_y + (x + y - t\,u)\,\partial_u.$$

These generators form a 6-dimensional Lie algebra of point symmetries for $\Delta = u_t + u\,u_x + u\,u_y - \nu\,(u_{xx} + u_{yy}) = 0$ and reduce to the standard 1D viscous Burgers' symmetries upon restricting to $y$-independent solutions and dropping the $\partial_y$ component.

