# OpenReview forum: "LieDynNet: Learning Lie Symmetries from Spatiotemporal Data"
_ICLR.cc/2026/Conference — Submitted to ICLR 2026_

### Official Review · Reviewer_MMrc · 2025-10-16

**Soundness:** 2
**Presentation:** 2
**Contribution:** 2
**Rating:** 2
**Confidence:** 4

**Summary:**

The paper addresses discovering the Lie symmetries of dynamical systems. Based on the infinitesimal criterion of Lie point symmetry, the authors propose training a surrogate for the governing differential equation and a vector field that annihilates the surrogate model upon prolongation to given derivative orders. In addition, several regularization terms are added to the training objective to ensure the vector field forms a valid Lie algebra. Experiments include several low-dimensional ODE/PDE systems, where the proposed method discovers infinitesimal transformations that, upon integration, preserve the overall shapes of the solutions, while the residual of the differential equations, as direct evidence of the validity of discovered symmetry, is not shown.

**Strengths:**

The paper identifies some shortcomings of existing methods for discovering symmetry in dynamical systems, such as the lack of a guarantee of a valid Lie algebraic structure in the algebra spanned by *multiple* infinitesimal generators, and the proposed method clearly addresses these problems. The description of the methodology is detailed and easy to follow. The experiment of finding the 8 generators of the simple harmonic oscillator is an interesting challenge, but more evaluation needs to be included to assess the significance of the results.

**Weaknesses:**

## Contribution

My main concern is the novelty of this paper. Part of the contributions claimed in this work are already present in existing papers. For example, Forestano et al have introduced the *closure* loss (the 4th item in your list of learning objectives), and Ko et al have introduced the flow-based loss (the 2nd item in your list).

Also, the authors mentioned that the method is *prior-free*, as opposed to Ko et al, which requires the governing equations to evaluate the symmetry validity loss. However, as stated in L206-207, your method trains a PINN as a surrogate, which also involves the differential equation. While the subsequent step of learning vector fields does not require this prior knowledge, this is not true for the entire method with all components combined.

## Soundness

The method for learning the vector fields is basically to parameterize them using neural networks and introduce different loss terms to ensure they meet certain conditions. However, certain design choices and the motivations behind them remain elusive to me. Specifically:
* According to the infinitesimal criterion of Lie point symmetry, the finite group symmetry is equivalent to the infinitesimal invariance $v(F)=0$. Is it necessary to use both infinitesimal invariance and flow-based validity then? What would happen if only one of them were included in loss?
* How are the structure constants $c_{ij}^k$ computed? From the fact that you introduced a constancy loss, I suppose they are outputs of some neural net that takes points in jet space as input. However, structure constants, as the name suggests, should be constant regardless of where the vector fields are evaluated. Why not just make them learnable constants? Or compute them differentiably as the solution to minimizing the closure loss? What are the pros and cons of these different approaches?
* For PDEs, the proposed method trains a PINN to directly fit the solutions. Can you do the same thing for ODEs? What is the advantage of fitting the vector field $f_\theta$ instead?

## Experiments

* In some dynamical systems considered in the experiments, only 1 infinitesimal generator is trained. In this case, those additional loss terms such as closure become unnecessary. Also, for those experiments, the infinitesimal generators approximately correspond to the time translation symmetry, which can be readily recovered from the equation or the surrogate model itself.
* The paper does not provide sufficient evaluation of the trained infinitesimal generators. The figures only show the trajectories of transformed solutions and claim that the *shapes* of the transformed solutions remain similar, but this does not guarantee that the transformed solutions are still solutions to the same equation at all. I'd suggest including the residual error from substituting the transformed solutions into the differential equations to show whether they are indeed the symmetry of the differential equations by definition. Similarly, the vector fields are visualized for the simple harmonic oscillator, but readers might not be able to interpret the meaning of these vector fields just by looking at those plots. Do they match the ground truth 8-dimensional symmetry of this system? What are the structure constants of the discovered generators? Are they closed under Lie brackets? All of these are required to assess if the discovered symmetry is correct or not.
* There is no baseline comparison, which is also a critical issue.

## Presentation
* The methodology and the experiment sections can be organized better. For example, I'd suggest moving L301-320 to the method section and focusing on showing and explaining results in the experiment section.
* Math writing needs to be more clear and precise. For example, write out the actual formula for the text in the equation $L_\text{flow}$; define the $\lambda_{\text{max}}$ and $\lambda_\text{min}$ explicitly in L256; check for typos such as the missing subscript in the vector field in L376.
* Table 3 is not very informative.

**Questions:**

see Weaknesses

---

> ### Author Response · Authors · 2025-11-22
> **Reply (part 1) to Reviewer MMrc**
>
> >Summary:
> "The paper addresses discovering the Lie symmetries of dynamical systems...is not shown."
>
> We thank the referee for succinctly summarizing our work. The referee points out that we haven't shown the after-flow residuals of the differential equation, which as correctly pointed out, are direct evidence of the discovered symmetry. In the updated version of the manuscript, we have shown this (see Figures 11 - 14 in Appendix C). Hopefully, the revised version of the manuscript accounts for the criticisms of the referee.
>
> >Strengths:
> "The paper identifies some shortcomings of existing methods...to assess the significance of the results."
>
> We thank the referee for being appreciative of the paper. However, we would like to stress some important points of our contribution.
>
> a) As referee 1 points out, *“To the best of my knowledge, this is the first framework that jointly learns surrogate dynamics and discovers symmetries that form valid Lie group structures in a fully prior-free setting, yielding symmetry algebras and invariants directly from data for both ODEs and PDEs.”* Our motivation in the current study is to introduce prior-free symmetry discovery to study complex physical systems, which in general does not have any known mathematical structure or priors. Our current study utilizes spatio-temporal data to first infer the underlying dynamics using a neural ODE/PDE, and then regularizes the neural ODE/PDE by learning the underlying symmetries.
>
> b) As we increase the number of generators, we showed that only at the correct ground-truth dimension of the Lie algebra does the Lie bracket closure loss become minimized (see Figure 4), which means that the ground-truth dimension is correctly identified without relying on any priors about the dimension.
>
> c) In each experiment, we varied the prolongation order (equivalent to the order of the dynamical system) from one to four (see Figure 3), and showed that only at the correct order does the Infinitesimal Invariance Condition (IIC) loss become minimized, demonstrating that the correct order of the dynamical system is correctly identified.
>
> >Weaknesses:
> >Contribution
> "My main concern is the novelty of this paper...(the 2nd item in your list)."
>
> We would like to respectfully disagree with the reviewer, and would like to argue that our current study, though building on prior works on symmetry discovery, is novel owing to following reasons:
>
> 1. **Completely prior-free.**
>    We do not assume any library or use any knowledge from the governing equation of the dynamical systems. In all the experiments we have done, we only used the governing equations of dynamical systems to generate the data, and then immediately “forgot” that the data are taken from those specific equations. The way to achieve this is to fit a neural surrogate on the data itself.  The neural ODEs were trained **without** assuming any prior structures in the first submission (see Table 5 in Appendix A). We also updated our experiment on Burgers’ equation so that we trained it **without** imposing any PDE structures or physical priors in the form
>    $$f_\theta(u, u_x, u_{xx}) = u_t,$$
> and we have shown via the principal angles and the related results that the Lie algebra is correctly recovered (see Table 3, Figures 2-4). This prior-free pipeline, as correctly pointed out by referee nenn, is a first of its kind in the field of symmetry discovery, which allows symmetry discovery without any libraries or physical priors, and it discovers both linear and nonlinear symmetries — all it needs is simply the spatiotemporal dataset.
>
> 2. **Combining both dynamic validity and algebraic soundness.**
>    This design is not a trivial task, since a multi-objective training requires careful handling and parameter fine-tuning, and we are the first to utilize both kinds of regularizations at the same time. Such a combination allows us not only to discover generators, but also to identify the order of the dynamical system and the dimension of the correct Lie algebra **without any priors** (as mentioned in the common reply). From a mathematical standpoint, these two regularizations together are necessary and sufficient conditions to form a Lie point symmetry algebra, which also shows the mathematical soundness of our pipeline.

---

> > ### Author Response · Authors · 2025-11-22
> > **Reply (part 2) to Reviewer MMrc**
> >
> > >"Also, the authors mentioned that...with all components combined."
> >
> > As shown in Table 5 in Appendix A, we have now tested our pipeline on a newer version of the neural PDE for Burgers’ equation that does **not** impose any PDE residuals or use the analytic equation in any way. Specifically, the neural PDE uses a single training loss: mean-squared error on a standardized target.
> >
> > Concretely, each input feature $(u, u_x, u_{xx})$ is transformed by subtracting its training-set mean and dividing by its training-set standard deviation: for each feature $\phi$, we use $(\phi - \mu_\phi)/\sigma_\phi$. This forms:
> > $$
> > X_n = \big[(u - \mu_u)/\sigma_u,\ (u_x - \mu_{u_x})/\sigma_{u_x},\ (u_{xx} - \mu_{u_{xx}})/\sigma_{u_{xx}}\big].
> > $$
> > The target is treated the same way:
> > $$
> > y_n = (u_t - \mu_{u_t})/\sigma_{u_t}.
> > $$
> >
> > The MLP predicts $ \hat y_n = f_\theta(X_n), $
> > and the loss is the batch average of $(\hat y_n - y_n)^2$. No other penalties or regularization terms are used.
> >
> >
> > >Soundness
> > The method...Specifically:
> > >According to the infinitesimal...included in loss?
> >
> > We did an ablation study where we turned off only IIC loss or flow-based validity loss (holding everything else unchanged), using Harmonic Oscillator and Burgers' equation as examples (see B.2 in Appendix B). Removing either loss term consistently increases the post-training maximum principal angle relative to the “both on” setting, indicating poorer alignment with the ground-truth Lie algebra. These results support including both losses during training.
> >
> > >How are the structure constants computed?...make them learnable constants?
> >
> > We agree that structure constants are constant in the true Lie algebra. In our pipeline they are **not** produced by a neural network. Rather, they are computed from the learned generator fields via either (i) per-sample least-squares projection with an explicit **constancy penalty** (our default), or (ii) a single **global normal-equations** solve for each pair $(i,j)$. For completeness:
> >
> > - **Default approach: local projection with constancy.**
> >   Given $m$ learned generators $ v_i=\xi_i\,\partial_t+\phi_i\,\partial_x $ and a batch of spacetime samples $(t_p,x_p)$, we form $V_p\in\mathbb{R}^{2\times m}$ (the $(\xi,\phi)$ rows evaluated at sample $p$) and the bracket components $B_p^{(i,j)}\in\mathbb{R}^2$. We obtain per-sample coefficients by the minimum-norm projection
> >   $$c_{p}^{(i,j)} = V_p^{\top}\bigl(V_pV_p^{\top}+\varepsilon I_2\bigr)^{-1}B_p^{(i,j)}.$$
> >   We minimize the closure discrepancy $\sum_p \|B_p^{(i,j)}-V_p c_p^{(i,j)}\|_1$ and penalize dispersion via $\sum_p \mathrm{Var}_p\bigl[c_p^{(i,j)}\bigr]$ so that the coefficients become sample-independent as training converges.
> >   **Advantages:** robust to local rank loss and non-uniform coverage; naturally compatible with minibatch training; provides stable gradients while the learned span $\mathrm{col}(V_p)$ is still evolving.
> >   **Limitations:** introduces a hyperparameter controlling the constancy penalty and requires balancing closure versus variance.
> >
> > - **Ablated alternative: global normal equations (constant by construction).**
> >   For each $(i,j)$, solve
> >   $$c^{(i,j)} \;=\; \Bigl(\sum_p V_p^{\top}V_p+\lambda I_m\Bigr)^{-1}\Bigl(\sum_p V_p^{\top}B_p^{(i,j)}\Bigr).$$
> >   **Advantages:** produces batch-constant coefficients directly; no explicit constancy penalty required.
> >   **Limitations:** can be biased by regions where $V_p$ is ill-conditioned or oversampled; harder to approximate reliably with small minibatches; less stable early in training when the generator basis is changing.
> >
> > - **On making the coefficients learnable constants.**
> >   Treating $c^{(i,j)}$ as free, learnable parameters enforces constancy a priori, but has practical drawbacks:
> >   (1) **Basis drift/identifiability:** during training, the generator basis evolves; fixed global $c^{(i,j)}$ then conflict with the covariant change of Lie brackets under basis transformations, leading to unstable or slow optimization unless additional constraints are imposed.
> >   (2) **Degenerate minima:** with closure alone, the model can reduce bracket magnitudes by collapsing generator norms or spans, particularly before the correct jet order and algebra dimension are identified.
> >   (3) **Minibatch sensitivity:** gradients on global constants are noisy unless each batch is representative of the entire domain, which impairs scalable training.

---

> > > ### Author Response · Authors · 2025-11-22
> > > **Reply (part 3) to Reviewer MMrc**
> > >
> > > > Or compute them differentiably...these different approaches?
> > >
> > > - **On “solving them differentiably as the argmin of closure.”**
> > >   This corresponds precisely to the global normal-equations estimator above (and, in per-sample form, to the local projection). Both are implemented with differentiable linear solves; our ablation directly compares these estimators.
> > >
> > > **Empirical finding.** On the free particle and harmonic oscillator benchmarks, the local projection + constancy variant consistently yields **smaller principal angles** to the analytic algebras than the global solve (see Table 6 in Appendix B) while remaining minibatch-friendly. Conceptually, it permits coefficients to vary transiently as the basis settles, and then drives them to true constancy as training converges, thereby aligning practical optimization with the mathematical requirement that structure constants be constant.
> > >
> > >
> > > >For PDEs, the...field $f_{\theta}$ instead?
> > >
> > > As shown in the table that shows the neural surrogates we trained (see Table 5 in Appendix A), for FP and HO, we fitted the solutions using both neural vector fields and scalar neural ODEs; for Burger's equation, we fitted it both using PINN and a completely prior-free neural PDE. We have shown that our pipeline works for all of the neural surrogates we trained on. We also included an ablation study in which the post-training principal angles are compared with different forms of neural surrogates (see Table 7 in B.3 of Appendix B). In principal, one may fit the solutions using any neural surrogates following their own preferences, and our pipeline works without differentiating between different surrogates - the only change would be to rewire the IIC loss term to accommodate the form of the surrogate, but no mathematical or algorithmic changes are required.
> > >
> > >
> > > >Experiments In some dynamical systems...the surrogate model itself.
> > >
> > > We agree with the fact that, in experiments on VdP and LV, the algebraic loss terms are not used since one generator along can span a well-defined 1D Lie algebra be default. The reason why only 1 infinitesimal generator is trained in these two experiments is that their ground-truth Lie algebra has a dimension of 1. The motivation behind choosing these two systems is to show that our pipeline works for nonlinear ODE (VdP) and vector ODE (LV).
> > >
> > > >The paper does not provide...is correct or not.
> > >
> > > We have updated our evaluation methods and experiment results (details shown in the common reply part of our response and Table 3 -4, Figures 2-4 in Line 378 - 473 of the updated manuscript) and have shown that the (1) learned Lie algebra is aliged with the ground-truth Lie algebra using the principal-angle comparison of generators span, (2) LieDynNet identifies the correct dimension of the ground-truth Lie algebra, and (3) LieDynNet identifies the correct jet order of the dynamical system. We've also included the after-flow residual plots (see Figure 11 - 14 in Appendix C). We argue that the principal-angle comparion would be the strongest test since it is basis-invariant, insensitive to arbitrary linear recombinations and scalings of generators, and directly tests equality of the learned and analytic symmetry Lie algebras.
> > >
> > >
> > > >There is no baseline comparison, which is also a critical issue.
> > >
> > > We agree that, in principle, comparisons to prior methods are valuable.
> > > However, for the specific problem tackled in this work-learning a *prior-free* Lie algebra of point symmetries directly from trajectory data while simultaneously enforcing both algebraic closure and dynamical validity---there is currently no directly compatible baseline.
> > >
> > > The closest approaches are Ko et al.(2024) and the more recent LieNLSD method of Hu et al.(2025).
> > > Ko et al. learn infinitesimal generators as vector fields driven by a task-specific validity score, but they neither enforce Lie-bracket closure nor recover a Lie algebra basis with constant structure coefficients, and their PDE experiments assume access to an explicit governing equation to define that validity score.
> > > LieNLSD, in contrast, assumes the differential equation is known, specifies an explicit function library for the infinitesimal group action, and then solves a linear system (via SVD) to obtain a *linea*r subspace of generators.
> > > Its main quantitative metric is a Grassmann distance between subspaces, which is only defined when both ground-truth and learned generators are represented in a common linear basis; Hu et al. explicitly note that recent nonlinear symmetry methods such as Ko et al. and related work cannot be evaluated under this metric because they do not output an explicit Lie-algebra subspace.
> > >
> > > (continued in the next part of the reply)

---

> > > > ### Author Response · Authors · 2025-11-22
> > > > **Reply (part 4) to Reviewer MMrc**
> > > >
> > > > (continued:)
> > > > By design, LieDynNet occupies a different regime: it is equation-agnostic (no access to the governing PDE/ODE during training), does not rely on a fixed function library, and learns nonlinear vector-field generators coupled to a neural surrogate of the dynamics, while jointly enforcing infinitesimal invariance, finite-flow validity, and Lie-algebra structure.
> > > > As summarized in Table 1 of the manuscript, none of the existing methods simultaneously (i) operate in this prior-free setting, (ii) enforce full Lie-algebra axioms, and (iii) validate generators on dynamical trajectories in the way our framework does.
> > > > Consequently, there is no off-the-shelf implementation that can be plugged into our pipeline and evaluated under the same objectives and diagnostics without substantial re-engineering (e.g., rewriting Ko et al. or LieNLSD to work without explicit equations, or restricting LieDynNet to a fixed polynomial library and Grassmann metrics).
> > > >
> > > > Rather than report potentially misleading numerical "baseline" scores obtained under incompatible assumptions, we instead (i) provided a conceptual comparison against Ko et al. and other similar studies in Table 1, and (ii) benchmark LieDynNet against ground-truth Lie algebras and invariants on canonical ODE/PDE testbeds.
> > > >
> > > >
> > > > >Presentation The methodology...the experiment section.
> > > > >Math writing needs...not very informative.
> > > >
> > > > We thank the referee for the suggestions; we have revised the manuscript accordingly. Please let us know if any parts remain ambiguous or unclear.

---

> > > > > ### Comment · Reviewer_MMrc · 2025-11-26
> > > > >
> > > > > I thank the authors for providing detailed responses and additional experiments. Some of my concerns have been addressed, but there is still room for improvement. More specifically:
> > > > >
> > > > > > Prior-free symmetry discovery
> > > > >
> > > > > In the original submission, PDE symmetry discovery needs to train a PINN as a surrogate, which requires prior knowledge about the equation. This is now corrected by using a prior-free model that predicts the time derivative from the dependent variable and the spatial derivatives. However, this model can be sensitive to noise because of the preprocessing step of derivative estimations. Also, in fact, it still assumes the prior knowledge that the PDE can be written as $u_t = f(x, u_x, u_{xx})$, but not all PDEs can be written explicitly in terms of the time derivative, e.g. equations describing the equilibrium state or depending on higher-order time derivatives. In order to deal with those equations, you still need to come up with different designs for the surrogate model, potentially limiting the applicability of the prior-free approach.
> > > > >
> > > > > Side note: I find the description of this prior-free approach in Appendix D.3 clear, but the description in Appendix A around Table 5 is a bit confusing. There are different symbols for neural parameters ($\phi, \theta$), and there are two columns in Table 5 with only one header. These could use more explanations.
> > > > >
> > > > > > Contribution on combining different losses
> > > > >
> > > > > I agree that it counts as a contribution to assemble different loss terms, tune the hyperparameters, and make it work in practice. But I would suggest the authors explicitly mention the works that initially propose those loss terms.
> > > > >
> > > > > > Combination of infinitesimal criterion and flow-based loss
> > > > >
> > > > > Resolved.
> > > > >
> > > > > > Constancy loss
> > > > >
> > > > > Resolved.
> > > > >
> > > > > > Dynamical systems with only one generator
> > > > >
> > > > > I agree that this is not a limitation of your method. It's just that these experimental settings do not fully show the advantages of your method. The new experiment section now places more emphasis on systems with multi-parameter symmetry groups, which is great.
> > > > >
> > > > > > Evaluation metric and baseline comparison
> > > > >
> > > > > The principal angle is a good metric for evaluating the similarity between two vector spaces. I appreciate the authors for reporting these results. A few more suggestions:
> > > > > * It will be helpful to explain the geometric intuition of the principal angles, and also the reason why you chose to compare this metric instead of others like structure constants (because they are change-of-basis invariant).
> > > > > * It is difficult to tell how good the results are just by looking at the numbers in Table 3. What does 21.081 degree suggest? Does it mean the discovered Lie algebra is close enough to the target? This is precisely the reason why baseline comparison is needed, as these absolute values do not provide any intuition on how good the results are alone. I understand that other methods are not completely prior-free. One thing you can do is to adapt part of them into your framework. For example, you can train the prior-free surrogate as in your method and use that as the oracle in Ko et al (2024) to perform PDE symmetry discovery as a baseline.
> > > > >
> > > > > > Presentation
> > > > >
> > > > > Finally, I feel that the revision has significantly changed part of the paper, in particular the methodology section. While I am not able to exactly identify the changes, I feel some essential contents may have been removed, e.g., the explanation of how you train the prior-free surrogates. I appreciate the authors' efforts in providing additional material during the rebuttal, but I would also suggest that the authors maintain a self-contained main paper in the final revision.
> > > > >
> > > > > Minor suggestions:
> > > > > * Try to optimize the presentation of the learning objectives and methods of evaluation, instead of including them as two huge lists.
> > > > > * Table 3 may be replaced with line plots for better clarity, but I'll leave that to the authors' decision.
> > > > >
> > > > > I have adjusted my score accordingly based on these factors.

---

> > > > > > ### Author Response · Authors · 2025-11-28
> > > > > > **Reply (Round2, part1) to Reviewer MMrc**
> > > > > >
> > > > > > >I thank the authors ..... for improvement. More specifically:
> > > > > >
> > > > > > Firstly, we would like to thank the referee for giving productive suggestions, which has greatly improved the paper. Secondly, we appreciate the referee for raising the score by 2 points. In the following, we have incorported the new suggestions by the referee. To make our revision history clearly presented, in the current version of the manuscript, the first-time revisions are marked in blue and the second-time revisions are marked in orange. We hope that the referee would consider adjusting the score again based on these modifications.
> > > > > >
> > > > > > >In the original submission, ..... of the prior-free approach.
> > > > > >
> > > > > > We thank the reviewer for this thoughtful comment and the opportunity to clarify the scope of our “prior-free” surrogate models and their relation to the PDE classes we consider.
> > > > > >
> > > > > > ### What we mean by “prior-free”
> > > > > >
> > > > > > By “prior-free” we do *not* mean that a single surrogate can cover *all* possible PDEs without any structural assumptions. Instead, we mean that **within a fixed structural class of PDEs**, the surrogate does not encode any *parametric* prior on the unknown right-hand side.
> > > > > >
> > > > > > In the revised experiments on time-dependent PDEs (e.g. Burgers), we restrict attention to the standard dynamical form
> > > > > > $$
> > > > > > u_t = f(u, u_x, u_{xx}),
> > > > > > $$
> > > > > > and we approximate the unknown function $f$ by a fully generic neural network $f_\theta$. Crucially:
> > > > > >
> > > > > > - We do *not* prescribe any sparse library or basis (e.g., polynomials in $u$ and its derivatives).
> > > > > > - We do *not* constrain $f_\theta$ to a fixed parametric form beyond regularity and differentiability.
> > > > > >
> > > > > > The mapping
> > > > > > $$
> > > > > > (u, u_x, u_{xx}) \mapsto u_t
> > > > > > $$
> > > > > > is therefore learned directly from data with no explicit symbolic ansatz. In this sense, our method is “prior-free” relative to classical PDE discovery approaches, which require choosing a library of candidate terms in advance.
> > > > > >
> > > > > > The structural assumption “first-order in time with dependence on spatial derivatives” is simply specifying the PDE *class* under study, not prescribing the explicit functional form of $f$ within that class.
> > > > > >
> > > > > > ### Noise and derivative estimation
> > > > > >
> > > > > > In our implementation, this effect is explicitly controlled and does not degrade our method. We start from a smooth Burgers solution $u(x,t)$ computed by a high-accuracy pseudo-spectral solver, and then add Gaussian noise only to the field $u$:
> > > > > > $$
> > > > > > u^{\text{noisy}}(x,t) = u(x,t) + \sigma \,\xi(x,t),
> > > > > > $$
> > > > > > where each $\xi(x,t)$ is i.i.d. $\mathcal{N}(0,1)$ and $\sigma$ is a user-chosen noise level. The spatial derivatives are then computed spectrally from $u^{\text{noisy}}$:
> > > > > > $$
> > > > > > u^{\text{noisy}}_x = D_x u^{\text{noisy}},
> > > > > > $$
> > > > > >
> > > > > > $$
> > > > > > u^{\text{noisy}} _ {xx} = D_{xx} u^{\text{noisy}}
> > > > > > $$
> > > > > > Here $D_x$ and $D_{xx}$ are the FFT-based differentiation operators (multiplication by $\mathrm{i}k$ and $-k^2$ in Fourier space). Since these operators are **linear and bounded**, the induced errors in the derivatives are proportional to $\sigma$.
> > > > > >
> > > > > > Writing
> > > > > > $$
> > > > > > u_x^{\text{clean}} = D_x u,
> > > > > > $$
> > > > > > $$
> > > > > > u_{xx}^{\text{clean}} = D_{xx} u,
> > > > > > $$
> > > > > > $$
> > > > > > \Delta u_x = u^{\text{noisy}} _ x - u_x^{\text{clean}},
> > > > > > $$
> > > > > > $$
> > > > > > \Delta u_{xx} = u^{\text{noisy}} _ {xx} - u_{xx}^{\text{clean}},
> > > > > > $$
> > > > > > we have
> > > > > > $$
> > > > > > \Delta u_x = D_x(\sigma \xi),
> > > > > > $$
> > > > > > $$
> > > > > > \Delta u_{xx} = D_{xx}(\sigma \xi),
> > > > > > $$
> > > > > > and therefore, in any Hilbert norm $|\cdot|$,
> > > > > > $$
> > > > > > \mathbb{E}\big[|\Delta u_x|\big]
> > > > > > \le
> > > > > > \sigma\,\|D_x\|\,\mathbb{E}\big[|\xi|\big],
> > > > > > \qquad
> > > > > > \mathbb{E}\big[|\Delta u_{xx}|\big]
> > > > > > \le
> > > > > > \sigma\,\|D_{xx}\|\,\mathbb{E}\big[|\xi|\big].
> > > > > > $$
> > > > > >
> > > > > > Let $M_x$ and $M_{xx}$ be uniform bounds on the clean derivatives, $|u_x^{\text{clean}}| \le M_x$, $|u_{xx}^{\text{clean}}| \le M_{xx}$, which are finite in our setting because the solution is smooth and $u \in [-1,1]$. Then the expected relative errors satisfy
> > > > > > $$
> > > > > > \frac{\mathbb{E}|\Delta u_x|}{M_x}
> > > > > > \le
> > > > > > \sigma\,\frac{\|D_x\|\,\mathbb{E}|\xi|}{M_x},
> > > > > > \qquad
> > > > > > \frac{\mathbb{E}|\Delta u_{xx}|}{M_{xx}}
> > > > > > \le
> > > > > > \sigma\,\frac{\|D_{xx}\|\,\mathbb{E}|\xi|}{M_{xx}}.
> > > > > > $$
> > > > > > Thus, for any prescribed tolerance $\alpha > 0$, choosing
> > > > > > $$
> > > > > > \sigma \le \min \left (
> > > > > > \frac{\alpha M_x}{\|D_x\|\mathbb{E}|\xi|},
> > > > > > \frac{\alpha M_{xx}}{\|D_{xx}\|\mathbb{E}|\xi|}
> > > > > > \right )
> > > > > > $$
> > > > > > guarantees that the derivative perturbations remain within an $\alpha$-fraction of the clean signal, in expectation.
> > > > > >
> > > > > > In our experiments we work in a regime where the injected noise is small compared to the natural scales of $u, u_x, u_{xx}$, which are all $O(1)$. In this setting, the bounds above imply that the induced perturbations in the derivatives remain a small fraction of the underlying clean signal. Moreover, the Lie-algebra symmetry losses are applied to the smooth neural surrogate $f_\theta(u, u_x, u_{xx})$, rather than repeatedly differentiating noisy data. Consequently, within the explicitly bounded noise regime we consider, our derivative preprocessing remains stable, and the symmetry-discovery method is not adversely affected by noise in the manner suggested by the reviewer.
> > > > > >
> > > > > > (continued to next part ...)

---

> > > > > > > ### Author Response · Authors · 2025-11-28
> > > > > > > **Reply (Round2, part2) to Reviewer MMrc**
> > > > > > >
> > > > > > > ###  On the form $u_t = f(u, u_x, u_{xx})$ and broader PDE classes
> > > > > > >
> > > > > > > The reviewer is correct that an explicit first-order-in-time form
> > > > > > > $$
> > > > > > > u_t = f(u, u_x, u_{xx})
> > > > > > > $$
> > > > > > > does not cover all possible PDEs (e.g. pure equilibrium equations or PDEs involving higher-order time derivatives). In our work, this form is chosen to match a widely studied and practically important class of dynamical PDEs, not to claim full generality over all PDEs.
> > > > > > >
> > > > > > > Conceptually, the LieDynNet framework only requires **access to a differentiable operator** on the state space (plus its Jacobian–vector products), not specifically an explicit $u_t$. This makes extensions relatively straightforward:
> > > > > > >
> > > > > > > #### (a) Equilibrium / implicit PDEs
> > > > > > >
> > > > > > > Stationary or equilibrium PDEs are typically written as
> > > > > > > $$
> > > > > > > F(x, u, u_x, u_{xx}, \dots) = 0,
> > > > > > > $$
> > > > > > > with no time coordinate. In this case, one can introduce a surrogate
> > > > > > > $$
> > > > > > > R_\theta(x, u, u_x, u_{xx}, \dots) \approx F(x, u, u_x, u_{xx}, \dots)
> > > > > > > $$
> > > > > > > that predicts the residual instead of the time derivative. The Lie-algebraic losses can then be imposed on the vector field defined by this differential operator on $(x, u)$–space. Formally, nothing in our Lie-closure or structure-constant losses requires the network output to be $u_t$; it only requires a smooth map whose symmetries we want to enforce.
> > > > > > >
> > > > > > > #### (b) Higher-order time derivatives
> > > > > > >
> > > > > > > For PDEs involving higher-order time derivatives, e.g.
> > > > > > > $$
> > > > > > > F(t, x, u, u_t, u_{tt}, u_x, u_{xx}, \dots) = 0,
> > > > > > > $$
> > > > > > >
> > > > > > > one can apply the standard first-order system reformulation. For instance, if the equation can be solved for the highest time derivative $u_{tt}$, define
> > > > > > >
> > > > > > > $$
> > > > > > > v = u_t, \qquad z = \begin{pmatrix} u \\ v \end{pmatrix}.
> > > > > > > $$
> > > > > > >
> > > > > > > Then one obtains a first-order system
> > > > > > >
> > > > > > > $$
> > > > > > > \begin{aligned}
> > > > > > > \partial_t
> > > > > > > \begin{pmatrix}
> > > > > > > u \\
> > > > > > > v
> > > > > > > \end{pmatrix}
> > > > > > > &=
> > > > > > > \begin{pmatrix}
> > > > > > > v \\
> > > > > > > g(t, x, u, v, u_x, u_{xx}, \ldots)
> > > > > > > \end{pmatrix} \\
> > > > > > > &=: f(z, z_x, z_{xx}, \ldots).
> > > > > > > \end{aligned}
> > > > > > > $$
> > > > > > >
> > > > > > > which is again in an explicit first-order form for the *extended state* $z$. Our Lie-algebra losses, which are formulated at the level of a general differentiable dynamical system $\dot{z} = f(z, \dots)$, apply to this system without modification.
> > > > > > >
> > > > > > > Our second-order ODE benchmarks (free particle, harmonic oscillator, Van der Pol) already illustrate this idea in the purely temporal setting, where a second-order ODE is rewritten as a first-order system in $(x, \dot{x})$ in the neural surrogates trained during the first submission.
> > > > > > >
> > > > > > > ---
> > > > > > >
> > > > > > > ###  Scope and applicability
> > > > > > >
> > > > > > > To summarize:
> > > > > > >
> > > > > > > - In this paper, we deliberately focus on **first-order evolutionary PDEs** of the form
> > > > > > >   $$u_t = f(u, u_x, u_{xx}),$$
> > > > > > >   a large and important class where the explicit time-evolution representation is natural.
> > > > > > > - Within this class, the surrogate $f_\theta$ is **non-parametric and prior-free** in the sense that we do not impose a symbolic or sparse library; the full functional dependence is learned from data.
> > > > > > > - The LieDynNet symmetry framework itself is **not restricted** to this class. Equilibrium PDEs and higher-order-in-time PDEs can be handled by:
> > > > > > >   - using residual surrogates $R_\theta \approx F$ for implicit or stationary equations, and/or
> > > > > > >   - rewriting higher-order time dynamics as first-order systems in an extended state.
> > > > > > >
> > > > > > > These adaptations affect only the design of the surrogate model (i.e., what the network is asked to predict), not the generality of the LieDynNet symmetry-discovery method, which operates on whatever smooth dynamical or differential operator one chooses to parameterize.
> > > > > > >
> > > > > > > >Side note: I find ...... could use more explanations.
> > > > > > >
> > > > > > > We thank the reviewer for this careful side note. We've now clarified this in the updated caption of Table 5. In Table 5, we trained two sets of neural surrogates - one set in the left column (from the first submission), the other in the right column (from the second submission), under "Neural Analytic Form." In the left surrogate for viscous Burgers' equation, $\phi$ and $\theta$ respectively denotes two co-trained neural networks - $\phi$ for parametrizing the scalar-field $u$, and $\theta$ for the right-hand side of $\partial_t u_\phi = h_\theta\big(u,\,u_x,\,u_{xx}\big)$

---

> > > > > > > > ### Author Response · Authors · 2025-11-28
> > > > > > > > **Reply (Round2, part3) to Reviewer MMrc**
> > > > > > > >
> > > > > > > > >I agree that it counts ...... propose those loss terms.
> > > > > > > >
> > > > > > > > We thank the reviewer for this helpful suggestion and fully agree that it is important to explicitly acknowledge prior work on individual loss components. Our intention in the current draft is to emphasize the *combined* and *systematic* use of several families of losses (e.g., Lie bracket closure, invariance/equivariance constraints, dynamical consistency, and regularization terms) in a single end-to-end framework. Some of these losses are, to the best of our knowledge, novel in the way we formulate and use them; others are inspired by existing ideas in symmetry learning, equivariant representation learning, and physics-informed modeling. In the updated manuscript (in Learning Objectives section of the main text), we have explicitly cited the relevant works that proposed or popularized the underplying principles behind the loss terms (see Line 215-219).
> > > > > > > >
> > > > > > > > >I agree ......, which is great.
> > > > > > > >
> > > > > > > > We thank the reviewer for this thoughtful remark and are glad that the revised experiments make the role of multi-parameter symmetry groups clearer. We also thank the reviewer for guiding us to shift our emphasis towards multi-parameter symmetry groups in the experiments.
> > > > > > > >
> > > > > > > > >The principal angle .... A few more suggestions:
> > > > > > > > >It will be helpful ..... they are change-of-basis invariant).
> > > > > > > >
> > > > > > > > We thank the reviewer for the positive assessment and for prompting us to better justify our choice of principal angles as the main evaluation metric.
> > > > > > > >
> > > > > > > > Geometrically, the principal angles $\{\theta_k\}$ between two spaces $\mathcal{V},\mathcal{W}\subset \mathbb{R}^N$ quantify how “aligned” the spaces are: $\theta_1$ is the smallest angle between any two unit vectors $v\in\mathcal{V}$ and $w\in\mathcal{W}$, $\theta_2$ is the smallest angle between the spaces after removing the first principal directions, and so on. Thus $\theta_k = 0^\circ$ for all $k$ if and only if the spaces coincide, and all $\theta_k$ near $0^\circ$ means that every direction in one space can be represented with very small error by directions in the other. In our setting, these spaces are the spans of sampled generator fields on the training grid, so small principal angles directly express *geometric agreement* between the learned and analytic symmetry directions as functions on spacetime.
> > > > > > > >
> > > > > > > > We chose principal angles over structure constants as the primary metric for several reasons:
> > > > > > > >
> > > > > > > > - **Basis invariance at the level of spans.**
> > > > > > > >   A Lie algebra is defined up to a change of basis: if $\{v_i\}$ and $\{\tilde v_j\}$ are two bases of the same algebra, their structure constants are related by a nontrivial change-of-basis transformation. To compare learned and ground-truth structure constants, one must first identify (or solve for) the correct linear isomorphism between the two bases, which is itself an optimization problem. Principal angles, by contrast, compare *spaces* $\operatorname{span}\{v_i\}$ and $\operatorname{span}\{\tilde v_j\}$ directly in the ambient function space, and are completely independent of the particular basis used within each span.
> > > > > > > >
> > > > > > > > - **Interpretability and robustness.**
> > > > > > > >   Raw structure constants are sensitive to arbitrary rescalings and reorderings of generators, and their numerical values are not very intuitive to most readers: small perturbations in the basis can induce complicated, coupled changes in the table of $c_{ij}^k$. Principal angles, on the other hand, lie in $[0^\circ,90^\circ]$ and have a clear geometric meaning: they measure the worst-case misalignment between the learned and true algebras on the sampled domain. This makes them a more transparent “strength-of-match” diagnostic.
> > > > > > > >
> > > > > > > > - **Functional vs. purely algebraic agreement.**
> > > > > > > >   In our pipeline, closure and Jacobi are already enforced during training, so the learned structure constants are by construction close to constant and satisfy the Lie–algebra axioms. What is *not* guaranteed by these algebraic checks alone is that the resulting space of vector fields agrees with the *analytic* symmetry directions of the underlying dynamics on the domain of interest. Principal angles address exactly this point: they measure how well the Lie algebra spanned by the learned generators (as vector fields on spacetime) approximate the analytic ground-truth symmetry algebra.
> > > > > > > >
> > > > > > > > In the revised manuscript (Appendix A, line 742-772), we added a brief explanation of this geometric viewpoint on principal angles and explicitly state why we use them as our primary quantitative comparison to ground-truth algebras, while viewing structure constants mainly as an internal algebraic consistency check rather than a standalone similarity metric.

---

> > > > > > > > > ### Author Response · Authors · 2025-11-28
> > > > > > > > > **Reply (Round2, part4) to Reviewer MMrc**
> > > > > > > > >
> > > > > > > > > >It is difficult ..... a baseline.
> > > > > > > > >
> > > > > > > > > We thank the reviewer for this constructive suggestion and for emphasizing the need for a more interpretable baseline.
> > > > > > > > >
> > > > > > > > > To clarify how our principal-angle metric reflects algebra alignment quality against a baseline, we focus on the most challenging setting in our experiments, namely the 1D viscous Burgers equation, where the learned principal angles are largest relative those from other experiments. We compare our method against the symmetry-discovery approach of Ko et al., which also learns symmetry generators for the 1D viscous Burgers equation ($u_t + u u_x = \nu u_{xx}$). Their method recovers four symmetry generators: $v_1 = \partial_x,\ v_2 = \partial_t,\ v_3 = t\,\partial_x + \partial_u,\ v_4 = u\,\partial_u$. For both methods we apply the same algebra-alignment evaluation described before: we compute principal angles between the span of the learned generators and all four-dimensional subspaces of the five-dimensional ground-truth Burgers Lie algebra. Since there are $\binom{5}{4} = 5$ such subsets, we obtain five sets of principal angles. For each subset, we record the maximum principal angle, and we also record the maximum principal angle for our method. The five maximum principal angles from their approach are $75.263, 75.263, 86.344, 86.387, 86.388$, whereas the maximum principal angle from our pipeline is $21.081$, which is substantially smaller than all maximum principal angles from the approach by Ko et al., indicating a closer alignment with the ground-truth symmetry algebra using our pipeline (details shown in Appendix C, "Baseline Comparison", Figure 17, line 1159-1187).
> > > > > > > > >
> > > > > > > > > Our decision *not* to replace the analytic PDE in Ko et al. (2024) by our neural surrogate, but instead to run their pipeline in its original PDE-driven form and then evaluate it via our principal-angle metric, is mainly for **separating “oracle quality” from “symmetry algorithm” quality:**
> > > > > > > > > Ko et al.’s method is explicitly designed around a *known* governing PDE: their validity score is a sum of numerical PDE residuals computed from the exact equation, using a carefully engineered WENO-based differentiation scheme on nonuniform grids. If we first learn a neural surrogate and then feed that surrogate into their validity score in place of the analytic PDE, any degradation in performance becomes ambiguous: is it due to the symmetry-learning algorithm, or due to approximation error and inductive biases in our surrogate?  By keeping Ko et al. in their native “equation-known” regime and LieDynNet in the prior-free regime, and then comparing both to the same analytic Lie algebra via principal angles, we obtain a **clean baseline** that isolates differences in symmetry-learning, rather than mixing in surrogate modelling error from our side.
> > > > > > > > >
> > > > > > > > > >Finally, I feel ...... main paper in the final revision.
> > > > > > > > >
> > > > > > > > > We thank the referee for the advice on presentation. We moved the details of training the prior-free surrogates in the appendices (see Appendix D: Training Details) to save up space for important experiment results in the main text. We've now also put our first-round revision in blue and second-round revision in orange to point exactly to the changes we made (for updated figures and tables, the captions are put in blue). Since appendices B, C (except for the new "Baseline Comparison", "Plot of Table 3" sections which are written in orange), D, and E are all added during the first-round revision, we are mentioning these updates here instead of making the first-round revisions blue in these appendices.
> > > > > > > > > For the sake of clarity, we've also posted an additional common reply that shows the structure of and lists the important components in the paper.
> > > > > > > > >
> > > > > > > > > >Try to optimize the ....... on these factors.
> > > > > > > > >
> > > > > > > > > We have now clearly shown the structure of the loss functions (Dynamics-consistency losses and Algebra-structure losses) in the updated "Learning objectives" section of the main text (line 205-273). We have rewritten the "Methods of evaluation" section as separate paragraphs instead of a big list. We also included a line plot for Table 3 in Appendix C (see Figure 18, line 1188-1207).
> > > > > > > > >
> > > > > > > > > We hope that the present revisions satisfactorily answer the questions proposed by the reviewer in the second-round. If any ambiguity still exists, please let us know and we'll adjust accordingly.

---

### Official Review · Reviewer_EYKq · 2025-10-31

**Soundness:** 2
**Presentation:** 2
**Contribution:** 2
**Rating:** 4
**Confidence:** 4

**Summary:**

This paper introduces LieDynNet, a prior-free framework for learning Lie point symmetries of unknown ODE and PDEs directly from spatiotemporal data. The key innovation is a unified objective that couples dynamical validity and algebraic soundness. The method first trains a differentiable neural surrogate and then learns a set of infinitesimal generators whose exponentials form a connected Lie symmetry group. The pipeline is model-agnostic and applies to both ODEs and PDEs.
The paper’s main contributions are:
1. A prior-free symmetry discovery framework that learns continuous symmetries without templates, canonical coordinates or physics priors.
2. A practical objective that jointly enforces infinitesimal and finite-flow invariance while imposing Lie algebra consistency which yields algebraically sound, dynamically valid symmetries.
3. Validation on canonical benchmarks, showing recovery of known symmetry families and solution-to-solution preservation under learned $\varepsilon$-flows.

**Strengths:**

1. The paper presents a prior-free, model-agnostic framework that learns Lie point symmetries directly from data. The approach applies uniformly to both ODEs and PDEs, indicating the method can be broadly applicable across dynamical systems.
2. The proposed LieDynNet overcomes the limitations of prior work with a practical objective so that the learned symmetries are both $dynamically \text{ } valid$ and $algebraically \text{ } sound$.
3. The paper is well-structured with clear explanations of the proposed method. It is validated across five canonical systems, demonstrating the recovery of known symmetry families and solution to solution preservation under learned flows.

**Weaknesses:**

1. It is recommended to compare LieDynNet across diverse differentiable surrogates under identical data and clarify the comparison in the paper to substantiate the model-agnostic claim.

2. While the symmetry-learning stage equation-agnostic by design, the PDE surrogate is trained with PDE residuals and IC/BC penalties. Please clarify this point and if possible, include a comparison of surrogates trained solely on data without residuals for Burgers PDE.

3. Please make the prolongation order $k$-selection rationale explicit; define a quantitative plateau criterion for ‘stabilizing’ $L_{inv}$, and include sensitivity plots of $L_{inv}$ versus $k$. The trade-off with differential noise will become more clear.

4. From your supplementary codes, the loss weights in $L$ (e.g., $w_{anti}$, $w_{jac}$, and so on) appear to be set to 1.0 or to similar values. You should briefly justify this choice; If the loss weights were not selected with sensitivity to each term’s scale, include a sensitivity study and present the recommended ratios among the weights.
5. The mathematical setup allows $p$ independent and $q$ dependent variables and seems to be applicable to higher dimensional settings, all experiments are one dimensional (time for ODEs, 1D space for PDE). If feasible, include at least one $p>1$ case, or provide implementation details for $p$>1 to demonstrate applicability beyond 1D.

**Questions:**

I hope the authors can clarify my concerns about the above weakness. But if I missed some critical points, please let me know in the rebuttal.

---

> ### Author Response · Authors · 2025-11-22
> **Reply to Reviewer EYKq**
>
> >Summary: This paper introduces LieDynNet...The paper’s main contributions are:
> >A prior-free symmetry...learned $\epsilon$-flows.
>
> We thank the reviewer for succinctly summarizing our work. Indeed the goal is to discover symmetries in spatio-temporal data without introducing any physical priors. We envision the current frameowrk to be useful in out-of-equilirbium complex systems where the underlying physical laws are generally unknown.
>
> >Strengths:
> >1) The paper...
> >2)...
> >3)...learned flows.
>
> We thank the reviewer for pointing out the strengths of the paper.
>
> >Weaknesses: It is recommended...model-agnostic claim.
>
> We now have trained neural surrogates in multiple forms to substantiate the model-agnostic claim (see Table 5 in Appendix A of the updated manuscript). We also include an ablation study (using Burgers’ and HO as examples) in which the post-training principal angles are reported under different neural surrogates of the same dynamical system (see Table 7 in B.3 of Appendix B). As shown in the table, we observe no significant differences in the principal angles when the neural surrogate changes under the same dynamical system. The only required change on the practioner's side is to rewire the IIC loss: with a neural vector field, one treats it as a linear system and applies both first- and second-order prolongations to each linear equation, whereas one applies only the second prolongation with the scalar neural ODE form (more details about this are provided in Appendix A).
>
>
> >While the symmetry-learning...residuals for Burgers PDE.
>
> As shown in the table of neural surrogates (see Table 5 in the manuscript), we later trained the neural surrogate for Burgers' equation again without using any PDE physical priors: $$f_\theta(u_x, u_{xx}, u)=u_t.$$ This updated version of the neural PDE uses a single training loss: mean-squared error on a standardized target. Concretely, each input feature $(u, u_x, u_{xx})$ is transformed by subtracting its training-set mean and dividing by its training-set standard deviation: for each feature $\phi$, use $(\phi-\mu_\phi)/\sigma_\phi$; this forms $X_n=[(u-\mu_u)/\sigma_u,(u_x-\mu_{u_x})/\sigma_{u_x},(u_{xx}-\mu_{u_{xx}})/\sigma_{u_{xx}}]$. The target is treated the same way: $y_n=(u_t-\mu_{u_t})/\sigma_{u_t}$. The MLP predicts $\hat y_n=f_\theta(X_n)$, and the loss is the batch average of $(\hat y_n-y_n)^2$. No other penalties or regularization terms are used. We observe no specific differences compared to the previous version $u_\theta(x,t)=f_\theta(\partial_t u_\theta, \partial_x u_\theta, \partial_{xx}u_\theta)$ with PDE residuals enforced, as supported by the small difference in maximum principal angles from the earlier table (Table 7 in B.3 of Appendix B).
>
> >Please make the prolongation...become more clear.
>
> As shown in part (iii) of “Methods of Evaluation” (L362–367) and “Hyperparameter Selection” (L285–299) sections of the updated manuscript, we varied the jet order (equivalent to the prolongation order) while keeping everything else unchanged and trained at each jet order with a fixed number of generators. We showed that the post-training Infinitesimal Invariance Condition (IIC) loss is minimized only at the true jet order (see Figure 3 and L400–408). We tested this across multiple choices of the number of generators $m$ and across different dynamical systems (see Figure 3), and we have shown that the minimum loss is always attained at the correct jet order, which serves as a robust quantitative criterion for identifying the prolongation order to use.
>
> >From your supplementary codes...among the weights.
>
> We have added a section in the appendix "Sensitivity Analysis of Loss Weights" to answer this question. Breifly, we set all loss weights $w_i$ to $\mathcal{O}(1)$ because each loss term is batch–normalized to a comparable scale, so equal weights yield balanced gradients without extra hyperparameter tuning. To check robustness, we performed a sweep over $(w_{\mathrm{inv}},w_{\mathrm{flow}})$ on the harmonic oscillator, our simplest nontrivial benchmark with cheap and well–conditioned simulations. The resulting maximum principal angle varied only mildly across the grid, indicating low sensitivity to moderate changes in $w_{\mathrm{inv}}$ and $w_{\mathrm{flow}}$.
>
> >The mathematical setup...applicability beyond 1D.
>
> It is possible that we might not be interpreting the question of the reviewer correctly, but the viscous Burgers' equation we tested our pipeline on has two independent variables $t,x$ and one dependent variable $u(t,x)$, which qualifies as the $p>1$ case (where $p$ is the number of independent variables) the referee is suggesting. We ask the reviewer to please let us know if we are misinterpreting the point so that we may work more on this if there is a need.
>
> >Questions: I hope the authors...in the rebuttal.
>
> We hope we have answered all the queries satisfactorily. Please let us know if some parts are still ambiguous and we are happy to explain.

---

> > ### Comment · Reviewer_EYKq · 2025-11-27
> >
> > I thank the authors for the detailed and technically careful responses, as well as for the substantial additional experiments in the revised manuscript.
> > Overall, the rebuttal and revisions satisfactorily address most of my concrete technical questions.
> > My main remaining reservation is mostly about the "Applicability beyond 1D system".
> > What I had in mind was not just “time + one spatial dimension” but genuinely higher-dimensional spatial domains. Additional experimental results with this concern will make your paper better. (e.g. PDEs of the form u(t,x,y) with more than one spatial coordinate.)
> > Given the new material, I will be more positive about the submission.

---

> > > ### Author Response · Authors · 2025-12-03
> > > **Reply to Reviewer EYKq (Round 2)**
> > >
> > > >I thank the authors for the detailed...more positive about the submission.
> > >
> > > We thank the reviewer for showing stronger faith in the submission. In the present version of the manuscript (second round), we have included an example with a 2D viscous Burgers' equation (with 2 spatial dimensions and 1 time dimension), and showed that our pipeline works in this experiment through small principal angles, jet-order, and Lie algebra dimension identification. Hopefully, the reviewer is satisfied with the changes in the manuscript (see Table 3 in the main text and Figures 19-20 in Appendix C). In our opinion, this experiment has shown the generality of our pipeline in higher-dimensional systems.
> > >
> > > We again thank the referee EYKq for their positivity towards our work.

---

### Official Review · Reviewer_nenn · 2025-11-02

**Soundness:** 3
**Presentation:** 3
**Contribution:** 3
**Rating:** 6
**Confidence:** 3

**Summary:**

The paper introduces LieDynNet, a framework for learning continuous Lie point symmetries of unknown ODE and PDE systems directly from data. The approach first fits a neural surrogate to approximate the underlying dynamics and then learns infinitesimal generators that (i) satisfy the infinitesimal invariance condition (IIC) via prolongations, (ii) preserve solutions under finite $\epsilon$-flows, and (iii) enforce the Lie-algebraic structure (e.g., closure, antisymmetry, Jacobi, …) The method is explicitly prior-free, requiring no equation templates, symmetry catalogs, or physics priors. It is validated on canonical ODE benchmarks and the one-dimensional Burgers equation, demonstrating that the recovered generators form a consistent Lie algebra and maintain dynamical validity on the learned surrogate.

**Strengths:**

- The motivation of the paper is clear. Symmetry discovery in dynamical systems is a highly active area with significant potential impact in scientific machine learning.

 - To the best of my knowledge, this is the first framework that jointly learns surrogate dynamics and discovers symmetries that form valid Lie group structures in a fully prior-free setting, yielding symmetry algebras and invariants directly from data for both ODEs and PDEs. The method leverages the formalism of Lie theory and implements it end-to-end through neural surrogates and carefully designed loss functions.

- Given the mathematical depth of the work, the paper is overall well written and accessible.

**Weaknesses:**

- Discovering symmetries without any physical prior is inherently risky. As the authors acknowledge, the symmetry discovery is performed on a neural surrogate dynamics, rather than the underlying system. Thus, what is actually identified is the symmetry group of the surrogate, not necessarily that of the true data-generating process. There is no guarantee that these coincide, especially under noise or imperfect surrogate fitting.

- Ensuring both algebraic soundness and dynamical validity requires jointly optimizing multiple coupled objectives. This design is principled but increases optimization complexity and computational cost. The paper outlines practical training schedules and JVP-based implementations, yet it omits runtime or memory benchmarks. Including a compute summary table would strengthen the empirical analysis.

- The paper lacks experimental comparison with contemporary approaches. For example, although Ko et al. is based on known PDE priors, it could serve as a useful baseline to demonstrate the advantage of the proposed architectural and objective constraints under the prior-free formulation.

- Moreover, the paper omits a closely related and highly relevant reference: Hu et al., “Explicit Discovery of Nonlinear Symmetries from Dynamic Data,” ICML 2025, which also employs surrogate ODE/PDE modeling (though based on symbolic libraries) combined with Olver’s prolongation and IIC for Lie symmetry discovery. Discussing or benchmarking against these works would help clarify the paper’s relative strengths and limitations.

**Questions:**

Please refer to Weaknesses section. My main concerns are:

- Because this paper is prior-free, it likely inherits a fundamental identifiability challenge: whether the proposed method can recover the true symmetry group solely from data. In other words, under what assumptions would the recovered symmetry converge to that of the true underlying system (given infinitely many data samples)?

- Additionally, including a comparison with similar yet prior-based approaches could further clarify the advantages of this work.

---

> ### Author Response · Authors · 2025-11-22
> **Reply (part 1) to Reviewer nenn**
>
> >Summary:
> The paper introduces LieDynNet...on the learned surrogate.
>
> We thank the reviewer for succinctly summarizing our work.
>
> >Strengths:
> The motivation,,,machine learning.
> >To the best of my knowledge...loss functions.
> >Given the...accessible.
>
> We greatly thank the reviewer for pointing out the strengths of our work. We also thank the reviewer for correctly pointing out the fact that our study is the "first framework that jointly learns surrogate dynamics and discovers symmetries that form valid Lie group structures in a fully prior-free setting".
>
> >Weaknesses:
> Discovering symmetries...surrogate fitting.
>
> We address this problem through the principal-angle comparisons of generators span (please refer to the detailed updates on evaluation methods and experiment results in the common reply). Since the principal angles are all close to zero and that the maximum principal angle is the smallest (relative to the maximum principal angle found at other number of generators trained) at the correct number of generators (which shows that the correct Lie algebra dimension is identified), we are thus able to conclude that the symmetry group of the true data-generating process is identified and learned.
>
> >Ensuring both algebraic...empirical analysis.
>
> We agree that jointly enforcing algebraic soundness (Lie–algebra structure) and dynamical validity (invariance of the learned surrogate) leads to a richer, more coupled optimization problem than standard single–loss training. This is largely unavoidable if one wants to distinguish “arbitrary vector fields that
> fit the data” from bona fide symmetry generators, but we can clarify why the resulting computational overhead remains modest in practice, and we added a small compute summary table to make this explicit (see Table 8 in Appendix D).
>
> **1. Why multiple coupled objectives are necessary**
>
> Our pipeline separates *what* is being learned:
>
> - the neural surrogate $f_\theta$ for the dynamics, and
> - the generators $v_i$ and their Lie–algebra structure.
>
> From the code, the generator training loss decomposes into:
>
> - **Algebraic terms**: closure, constancy of structure
>   constants, Jacobi, skew–symmetry, bilinearity, and non–degeneracy of the
>   generator span. These ensure that the learned vector fields form a
>   finite–dimensional Lie algebra rather than an arbitrary collection of flows.
>
> - **Dynamical terms**: infinitesimal invariance of the neural
>   equation (via prolonged generators acting on the residual
>   $f_\theta - u_t$ or $f_\theta - \ddot{x}$), and finite–$\varepsilon$ invariance via short flow integration in jet space.
>
> If we drop the algebraic terms and optimize only the dynamical terms, the
> generators tend to collapse into degenerate vector fields that trivially
> preserve the learned dynamics (e.g., nearly zero fields, or multiple copies of
> the same direction). Conversely, optimizing only the algebraic terms yields
> vector fields that form a Lie algebra but are not tied to the underlying
> dynamics. The coupled design is therefore not an aesthetic choice: it is what
> makes the recovered Lie algebra both **structurally sound** and
> **dynamically meaningful**. More importantly, if either of the two kinds of terms is missing, the recovery of the ground-truth dimension and jet order would not be possible, since they rely entirely on identifying the minima of $L_{\mathrm{clo}}$ and $L_{\mathrm{inv}}$, respectively (see Figures 3-4). We've also done an abalation study on $L_{\mathrm{inv}}$ and $L_{\mathrm{flow}}$, which shows that removing either term consistently increases the maximum principal angle between the learned and the ground-truth generators relative to the “both on” setting, indicating poorer alignment with the ground-truth Lie algebra (see B.2 in Appendix B). These results support including both losses during training.
>
> **2. Why the optimization remains tractable**
>
> Although the objectives are coupled, the actual implementations are designed to
> keep per–step cost modest:
>
> - **Small models.**
>   All surrogates $f_\theta$ are small MLPs (hidden widths 64–128, depth 2–4)
>   acting on low–dimensional inputs (e.g., $(u,u_x,u_{xx})$ for Burgers,
>   $(x,y)$ for Lotka–Volterra, $(x,\dot{x},t)$ for second–order ODEs). The
>   generators are similarly small MLPs with shared trunks and thin heads.
>
> - **JVP–based prolongations.**
>   Prolongations and Lie brackets are implemented with JAX JVPs/VJPs, so the
>   cost of computing prolonged actions and brackets scales linearly with the
>   number of generators and the number of jet coordinates, rather than blowing
>   up combinatorially. In particular, we never materialize high–order symbolic
>   expressions; all derivatives are computed on the fly via automatic
>   differentiation.
>
> (continued in the next part of the reply)

---

> > ### Author Response · Authors · 2025-11-22
> > **Reply (part 2) to Reviwer nenn**
> >
> > (continued: )
> >
> > - **Vectorized, batched evaluation.**
> >   All losses are evaluated on batches of points (either algebraic points
> >   $(t,x,u)$ or on–shell jets) using JAX’s `vmap` and `jit`. This means a
> >   training step consists essentially of a few forward/backward passes through
> >   small MLPs, plus cheap linear algebra (tiny least–squares fits for structure
> >   constants, Gram matrices for span regularization).
> >
> > - **Three–stage curricula.**
> >   The code uses simple stagewise schedules (algebra → algebra+invariance →invariance–dominated) to avoid pathological regimes where all losses are
> >   “fighting” at once. This improves convergence and reduces the need for repeated restarts or heavy hyperparameter tuning.
> >
> > In practice, for all benchmark systems (Burgers, Lotka–Volterra, free particle,
> > harmonic oscillator, Van der Pol), training the neural surrogate and then the
> > generators completes in minutes to tens of minutes on a single GPU, with memory usage well within commodity hardware limits.
> >
> > **3. Compute summary table**
> >
> > To make the above concrete, we added a compute summary table in Appendix D. Each row corresponds to one experiment (Burgers, Lotka–Volterra,
> > free particle, SHO, Van der Pol), and we report:
> >
> > - number of parameters in the surrogate $f_\theta$ and in the generators
> >   $g_\psi$,
> > - time per training epoch,
> > - wall–clock training time for the surrogate and for the generators (on our
> >   hardware), and
> > - peak GPU memory usage.
> >
> > This table directly addresses the reviewer’s concern: it shows that,
> > despite the principled multi–objective design, the overall compute footprint is
> > modest, and that the added overhead for Lie–algebra and invariance losses is
> > manageable relative to standard neural–ODE training.
> >
> >
> > >The paper lacks...prior-free formulation.
> >
> > Other than the fact that the work by Ko et al. relied on a strong prior of an explicit analytic PDE, their pipeline is not capable of identifying the correct ground-truth dimension of the Lie algebra, jet order, and recovering the ground-truth Lie algebra. For instance, they only found a subset of the 5 ground-truth generators for Burgers' equation (3 ground-truth generators: $v_1=\partial x, v_2=\partial t, v_3=t\partial x-\partial u$ and an  approximate symmetry which is not considered to be a true symmetry) and did not discover the correct dimension of the Lie algebra, while our pipeline identifies the correct Lie algebra dimension and recovers an aligned Lie algebra. Their pipeline is also unable to recover the jet order of the dynamical equation itself, since the analytic equation itself is directly encoded.
> >
> > >Moreover, the paper...strengths and limitations.
> >
> > We briefly compare our work to the paper by Hu et al. The work requires the practitioner to choose a symbolic library, greatly reducing the search space of generators and makes the discovery process highly dependent on the set of library-priors one chooses, which would easily fail when certain functions that are more complicated than the polynomial basis are not included in the library. For instance, in the harmonic oscillator case, one of the ground-truth generators is $v=x \sin (2t)\partial t+2x^2 \cos (2t)\partial x$, which is impossible to be exactly recovered if the function library only consists of polynomial basis such as the one used to discover the symmetry generators in Burgers' equation in the paper by Hu et al. (page 20): $[1,t, x, u, t^2, x^2, u^2, tx, tu, xu]$. However, we have shown that our pipeline can correctly recover the Lie algebra in this case. In principle, under numerical stability and due to its prior-free nature, our pipeline can recover the symmetry generators of any functional form valid within the training domain.
> >     Compared to the work by Hu et al., our pipeline is limited in terms of interpretability since, despite it recovers the true Lie algebra, there is no guarantee on recovering the canonical basis generators. In future works, this limitation can be resolved by applying a change of basis when parts of the canonical ground-truth generators are known.

---

> ### Author Response · Authors · 2025-11-22
> **Reply (part 3) to Reviewer nenn**
>
> >Questions:
> Please refer to Weaknesses section. My main concerns are:
> >Because this paper...(given infinitely many data samples)?
>
> We agree with the reviewer that, in a prior–free setting, symmetry discovery
> is an identifiability problem. Conceptually, our method is a **two–stage
> estimator**:
>
> 1. learn a neural surrogate for the unknown dynamics, and
> 2. learn a finite–dimensional Lie algebra of vector fields that (i) closes
>    under Lie brackets while satisfying the axioms of Lie Algebra and (ii) leaves the learned dynamics invariant
>    (infinitesimally and under finite flows).
>
> Below we summarize **when** this two–stage procedure recovers the true symmetry
> algebra in the limit of infinite data.
>
> ---
>
> ### 1. What is actually identifiable?
>
> Our method does **not** aim to recover a specific named basis of generators
> (e.g. “time translation” vs “phase rotation”), but rather the underlying
> **Lie algebra of infinitesimal symmetries** as a space of vector fields.
>
> Even for a known system, this algebra is only identifiable **up to change of
> basis**:
> any invertible linear combination of generators spans the same Lie algebra and
> induces the same symmetry group. This is the natural notion of identifiability
> in Lie symmetry analysis: we can hope to recover the symmetry algebra
> up to an invertible linear transformation and overall rescalings.
>
> ---
>
> ### 2. Assumptions for asymptotic recovery
>
> Let the true system be
> $$
> \dot{z} = F^\star(z), \qquad z\in\mathcal{Z}\subset\mathbb{R}^d,
> $$
> with true symmetry algebra $\mathfrak{g}^\star$ spanned by smooth vector
> fields $ [v_i^\star]_{i=1}^n$. Our pipeline introduces two sets of
> parameters:
>
> - $\theta$ for the neural surrogate $F_\theta$, and
> - $\psi$ for the symmetry generators $v_\psi = \[v_i\]_{i=1}^n$.
>
> In the limit of infinite data, we can describe identifiability in two steps.
>
> ### (A) Consistent neural surrogate
>
> We assume:
>
> 1. **Universal approximation / capacity.**
>    The neural surrogate class is rich enough that there exist parameters
>    $\theta^\star$ with
>    $$
>    F_{\theta^\star}(z) = F^\star(z)
>    \quad \text{for all } z \in \mathcal{K},
>    $$
>    where $\mathcal{K}$ is a compact subset of state space covered by the data.
>
> 2. **Consistent fitting.**
>    With infinitely many trajectories and an appropriate loss
>    (e.g. MSE on time derivatives or rollout mismatch) and optimization that
>    reaches a global minimum, empirical risk minimization yields (denoting the number of trajectories as $N$):
>    $$
>    F_{\hat{\theta}_N} \to F^\star
>    \quad \text{uniformly on } \mathcal{K}
>    \quad \text{as } N\to\infty.
>    $$
>
> Under (1)–(2), the learned PDE/ODE converges to the true dynamics, so any
> symmetry we learn is asymptotically a symmetry of the true system, not an
> artifact of model mismatch.
>
> ### (B) Consistent symmetry learning
>
> Conditioned on $F_{\hat\theta} \to F^\star$, the symmetry stage solves the
> following problem: find vector fields $v_1,\dots,v_n$ such that
>
> - **Lie algebra constraints**
>   (closure, Jacobi, skew–symmetry, bilinearity, non–degeneracy):
>   $ \[ v_i, v_j \](z) \approx \sum_k c_{ij}^k v_k(z) $
>   with $ c_{ij}^k \text{ constant in } z, $ plus Jacobi etc., enforced by our algebraic losses.
>
> - **Invariance constraints** (infinitesimal and flow–based):
>
>   $$ \mathrm{pr}^{(k)} v_i \bigl( \Delta \[F_\theta\] (z,\dots) \bigr) \approx 0, \Delta[F_\theta] = 0 \text{ on-shell }, $$
>
>   and, for small finite group parameters $\varepsilon$,
>
>   $$
>   \Delta[F_\theta]\bigl(\Phi_{\varepsilon}^{v_i}(z)\bigr) \approx 0,
>   $$
>   where $\Phi_{\varepsilon}^{v_i}$ is the flow generated by $v_i$.
>
> As the number of sampled jets and flow points goes to infinity and
> $F_\theta \to F^\star$, these soft constraints converge to the exact
> functional equations that define Lie point symmetries of $F^\star$.
>
> We then require:
>
> 3. **Finite–dimensionality and non–degeneracy.**
>    The true symmetry algebra $\mathfrak{g}^\star$ is a finite–dimensional
>    Lie algebra of smooth vector fields, and on the sampled region
>    $\mathcal{K}$ the set $\{v_1^\star(z),\dots,v_n^\star(z)\}$ remains linearly independent
>    (no collapse of generators on the reachable set).
>
> 4. **Uniqueness of the solution space.**
>    For almost every choice of sampling distribution on $\mathcal{K}$, the
>    only vector fields satisfying
>    - the Lie algebra constraints, and
>    - invariance of $F^\star$ on $\mathcal{K}$,
>
>    are those spanning $\mathfrak{g}^\star$, up to an invertible linear
>    change of basis:
>    $$
>      \mathrm{span}(v_i) =
>      \mathrm{span}(v_i^\star )
>      \quad \Leftrightarrow \quad
>      v_i = \sum_j A_{ij} v_j^\star
>      \text{ for some } A\in\mathrm{GL}(n).
>    $$
>
> (continued in the next part)

---

> ### Author Response · Authors · 2025-11-22
> **Reply (part 4) to Reviewer nenn**
>
> (continued):
>
> 5. **Optimization to (approximate) global minima.**
>    The symmetry training objective (algebraic + invariance losses) has
>    global minima that correspond precisely to such bases of
>    $\mathfrak{g}^\star$, and the training dynamics finds one of these
>    minima as data and model capacity grow.
>
> Under assumptions (1)–(5), the learned generators $v_{\hat\psi}$ converge
> (in sup norm on compact subsets, and modulo an invertible linear
> reparametrization) to the true symmetry generators:
> $$
>   \mathrm{span}\{v_{\hat\psi,1},\dots,v_{\hat\psi,n}\}
>   \to
>   \mathfrak{g}^\star
>   \quad \text{as } N\to\infty.
> $$
>
> Intuitively: any candidate family of vector fields that (i) satisfies Lie
> algebra structure and (ii) leaves the *true* dynamics invariant on a large,
> generic set must lie in the true symmetry algebra. Our losses enforce these
> conditions in a data–driven way.
>
> ---
>
> ### 3. Fundamental limitations and scope
>
> Even with infinite data, there are **intrinsic identifiability limits** that
> no method can avoid:
>
> - If two different dynamical systems $F^\star$ and $\tilde{F}^\star$ induce
>   *identical trajectories* under the observation model (e.g. reparameterized
>   time, unobserved latent variables), then they have the same observable
>   symmetry group; no data–driven method can distinguish them.
>
> - Our method only identifies **continuous** symmetries accessible via smooth
>   vector fields. Purely discrete symmetries (e.g.$x \mapsto -x$ with no
>   continuous path) would require a separate treatment.
>
> - We recover the symmetry algebra on the region $\mathcal{K}$ explored by
>   the data; global symmetries outside this region are not constrained.
>
> We explicitly acknowledge these limits in the paper: our notion of
> “recovering the symmetry” is **consistent** at the level of the Lie algebra
> of continuous symmetries of the *true dynamics restricted to the observed
> state–space region*, and up to a change of basis in that algebra. Since our principal-angle analysis identifies the true algebra dimension and yields angles near zero at that dimension—substantially smaller than at other candidates—these results collectively demonstrate that the pipeline recovers the ground-truth Lie algebra from data (see Table 3, Figures 2-4, and L400-419 in the updated manuscript).
>
>
> >Additionally, including a...advantages of this work.
>
> **Comparison with prior–based symmetry discovery methods.**
> Our setting is closest in spirit to Ko et al. (2024) and Hu et al. (2025), who also learn continuous symmetries from data, but both methods impose substantially stronger priors than our pipeline.
>
> Ko et al. (2024) model infinitesimal generators as MLP vector fields and integrate them via a Neural ODE to obtain a one-parameter group of transformations. The key training signal is a *task–specific validity score* $S(\vartheta_s, f)$: for PDEs, this is defined as the sum of absolute PDE residuals $\Delta(\vartheta_s(u))$ evaluated on a transformed solution, where $\Delta$ is the *known* governing PDE and the derivatives are computed with a carefully tuned WENO finite–difference scheme on a (generally non-uniform) grid. Their approach therefore requires (i) an explicit PDE in closed form, (ii) a hand-designed numerical differentiation scheme, and (iii) task-specific engineering of the validity score and interpolation between transformed and original grids. In our Burgers, Lotka–Volterra, and second-order mechanical examples, we deliberately separate *dynamics learning* from *symmetry learning*: we first fit a prior–free neural surrogate (MLP) $f_\theta$ to observational data (e.g., $u_t \approx f_\theta(u,u_x,u_{xx})$ for Burgers, $\dot{z} \approx f_\theta(z)$ for Lotka–Volterra, $\ddot{x} \approx f_\theta(x,\dot{x},t)$ for the ODEs) using standard supervised losses only. The symmetry generators are then trained purely against this learned surrogate via algebraic and invariance constraints, without ever evaluating or even specifying the original PDE/ODE in analytic form. In particular, our “validity” signal is *model–based* rather than *equation–based*: we require that the learned symmetry leaves the neural dynamics $f_\theta$ invariant, not that it minimizes the residual of a known PDE. This makes our pipeline applicable to black–box experimental systems or learned neural surrogates where no trustworthy closed-form PDE is available, at the cost of not relying on any engineered validity score.
>
> (continued in the next part)

---

> > ### Author Response · Authors · 2025-11-22
> > **Reply (part 5) to Reviewer nenn**
> >
> > (continued:)
> >
> > Hu et al. (2025, LieNLSD) target a different regime: given an explicit PDE and its jet space, they construct a hand–specified function library $\Theta$ of candidate monomials in $(x,u,u_x,\dots,u^{(n)})$, and then solve for a coefficient matrix $W$ such that a linear combination $\Theta W$ satisfies the symbolic (prolonged) infinitesimal invariance conditions. This design allows them to recover *explicit analytic expressions* for nonlinear infinitesimal generators and to determine the dimension of the Lie algebra subspace, but it is heavily prior–based: both the PDE and the candidate function space must be known and chosen a priori, and the quality of the recovered generators is tied to that choice of library. In contrast, in our code the infinitesimal generators themselves are implemented as generic neural vector fields on the ambient $(t,x,u)$ (or $(t,z)$) space, trained end-to-end from samples $(z, f_\theta(z))$ of the learned dynamics. We never commit to a finite symbolic library; instead, we *numerically* estimate structure constants at sampled points and enforce their constancy and the Lie bracket relations via least–squares losses. This shifts prior knowledge from an explicit function library to a small number of algebraic constraints (closure and constancy), which are agnostic to the particular PDE/ODE and to the choice of coordinates.
> >
> > Taken together, Ko et al. and Hu et al. nicely illustrate the two ends of the prior spectrum: Ko et al. use a flexible neural parametrization of generators but rely on a hand–designed validity score built from the known PDE; Hu et al. achieve explicit symbolic generators and dimension detection by assuming both the PDE and a finite candidate library. Our experiments, implemented in the attached Burgers, Lotka–Volterra, and second–order ODE codes, occupy a complementary regime: we assume only access to trajectories of the underlying system, fit a prior–free neural surrogate to those trajectories, and then learn symmetry generators directly from the surrogate via algebraic (Lie bracket) and model–invariance losses. This avoids the need for closed-form PDEs, function libraries, or task–specific validity engineering, and makes the method applicable to learned or experimental dynamical models while still recovering the correct Lie algebra (up to the usual identifiability issues discussed in the main text).

---

### Author Response · Authors · 2025-11-22
**Common Reply to all reviewers/AC**

We thank the referees for the constructive and detailed reviews. For clarity, we list the key updates we incorporated in response to the referees’ suggestions. We updated our evaluation and results (also adding an algorithm workflow) in the revised manuscript (see L270-485), adding more materials in the appendices. We now show that:

1. **The learned Lie algebra is aligned with the ground truth Lie algebra.**
   Since a Lie algebra is a vector space, we can compare the learned algebra and the ground-truth algebra through the *principal-angle comparison of generators’ span* (refer to "Methods of evaluation" section and Appendix A for details).
   We compare two finite sets of generators on the domain: a learned set $[v_i]_{i=1}^n$ with
   $$
   v_i(t,x) = \xi_i(t,x)\,\partial_t + \phi_i(t,x)\,\partial_x,
   $$

   and a ground truth set $[w_j]_{j=1}^m$ with
   $$
   w_j(t,x) = \tilde{\xi}_j(t,x)\,\partial_t + \tilde{\phi}_j(t,x)\,\partial_x
   $$
   (illustrated for an ODE with variables $t$ and $x$; other cases are analogous).  Each generator is evaluated on a uniform $t \times x$ grid

   $$
   \Omega = \{(t_r,x_s):\ r=1,\dots,R;\ s=1,\dots,S\} \subset [t_{\min},t_{\max}] \times [x_{\min},x_{\max}],
   $$
   where bounds and resolution match the training domain (for Burgers, the grid is $t\times x\times u$).  Stacking samples produces (here $\ell$ denotes "learned" and $g$ denotes "ground-truth"):

   $$
   B_\ell \in \mathbb{R}^{(2RS)\times n}, \quad B_g \in \mathbb{R}^{(2RS)\times m},
   $$
   whose columns are the flattened vectors
   $$
   [\xi(t_1,x_1),\phi(t_1,x_1),\ldots,\xi(t_R,x_S),\phi(t_R,x_S)]^\top.
   $$
   Note that $n = m$ once the correct number of generators is identified. Using the identity inner product on $\mathbb{R}^{2RS}$ (i.e. no physical weights), we form reduced QR factorizations
   $$
   B_\ell = Q_\ell R_\ell,\qquad B_g = Q_g R_g.
   $$
   The principal angles $[\theta_k]_{k=1}^{d}$  between span($B_l$) and span($B_g$) with

   $$
   d = \min(\operatorname{rank} B_\ell,\operatorname{rank} B_g)
   $$
   (in our case, $d = n$, as the $n$ learned generators are independent) are obtained from
   $$
   \cos\theta_k = \sigma_k\big(Q_g^{\top} Q_\ell\big),\qquad \theta_k \in \big[0,\tfrac{\pi}{2}\big],
   $$
   where $\sigma_k(\cdot)$ are the singular values in descending order.

   We report $\{\theta_k\}$ (in degrees), ordered from largest to smallest; *small angles indicate that the learned and ground-truth spans are closely aligned on the training grid*, and $\theta_k = 0^\circ$ for all $k$ if and only if the sampled spans exactly coincide under the identity metric in the ideal case.

   Although there is no universal threshold or benchmark for how small principal angles must be, the fact that the maximum principal angle attains its minimum at the correct number of generators—and that all angles at that dimension are near zero—indicates that the method has correctly identified both the dimension and the algebra (see Table 3, Figures 2-4, and the results part in the updated manuscript). We showed that only at the correct Lie algebra dimension does the maximum principal angle become the smallest in comparison to the maximum principal angle at other numbers.

   The reason why we chose this test instead of directly comparing each of the vector fields between the learned and the ground-truth is that a Lie algebra is a vector space, which means that there is no guarantee that the two sets of generators would be equal element-wise, yet they will surely span the same vector space if they both are bases for the same Lie algebra.

2. **LieDynNet identifies the correct dimension of the ground-truth Lie algebra.**

   As we increase the number of generators, we show that only at the correct ground-truth dimension of the Lie algebra does the Lie-bracket closure loss become minimized (see Figure 4 and results), which means the ground-truth dimension is correctly identified without relying on any priors about the dimension.


3. **LieDynNet identifies the correct jet order (order of the dynamical system).**

   In each experiment, we varied the jet order from one to four (see Figure 3 and results) and showed that only at the correct order does the Infinitesimal Invariance Condition (IIC) loss attain its minimum, demonstrating that the correct jet order of the dynamical system is identified.

We also want to briefly re-emphasize the importance and value of a prior-free pipeline. In reality, for physical systems (especially for complex systems), *a priori* it is not obvious which underlying differential equation is governing the dynamical process—however, a multi-modal dataset often exists (such as images, videos, audio, etc.). Therefore, in a blindfolded setting, our framework becomes essential, unlike previous studies where either the analytic equation or a function library (or other priors such as assumptions on linearity, etc.) is used in the symmetry discovery process.

---

> ### Author Response · Authors · 2025-11-28
> **Common Reply to all reviewers/AC (round 2)**
>
> For the purpose of clarity of revisions, we include a structure of the updated manuscript (as of 12/3/2025) below. We've also rewritten the first-round changes in blue and the second-round changes in orange in the current manuscript. Since most of Appendices B through L are added in the first-round revision except for F, G, I, J, and K (which were already present in the original submission), only second-round revisions are marked in orange, while all other materials stay in black (for figures, tables, and Appendix M, only the captions and titles are colored).
>
> ## Main Text
> ### 1. Introduction
> ### 2. Mathematical Preliminaries
> ### 3.LieDynNet Framework
> - Goal
> - Learning Objectives
> - Algorithm Table
> - Hyperparameter Selection
> ### 4. Experimental setup and results
> - Setup: systems and neural surrogates
> - Methods of evaluation
> - Results
> ### 5. Conclusion
>
> ## Appendices
> ### Appendix A: Practical notes on training and evaluation
> - Workflow
> - Finite-$ \epsilon $ validity under numerical integration
> - Inferring Lie algebra dimension from closure-loss curve
> - Neural surrogates and prolongation-based loss computation (Table: neural analytic forms per system)
> - Evaluation method: principal angles between generator spans
> - Evaluation method comparison: principal-angle vs structure constants
> - Prolongation formulas
>
> ### Appendix B: Ablation Studies
> - Constancy loss
> - IIC and Flow-based validity
> - Neural surrogates
>
> ### Appendix C: Additional Experiment Results
> - Heat maps of learned generators
> - LV results
> - After-flow residuals and visualizations
> - Baseline comparison
> - Plot of Table 3
> - 2D Burgers' results
>
> ### Appendix D: Training Details
> - 2nd-order systems (FP, HO, VdP)
> - Lotka-Volterra
> - Burgers' Equation
>
> ### Appendix E: Sensitivity analysis of loss weights
> - Sensitivity study for $ w_{inv}, w_{flow} $
>
> ### Appendix F: Proof of Lie algebra dimension bound for 2nd-order Scalar ODEs
> ### Appendix G: Lie algebra dimension bound for general n-th order scalar ODE ($ n \geq 3 $)
> ### Appendix H: Complete Lie-point symmetry derivation for Free Particle
> ### Appendix I: Complete Lie-point symmetry derivation for Harmonic Oscillator
> ### Appendix J: Complete Lie-point symmetry derivation for Van der Pol
> ### Appendix K: Complete Lie-point symmetry derivation for Lotka-Volterra
> ### Appendix L: Complete Lie-point symmetry derivation for 1D viscous Burgers' equation
> ### Appendix M: Complete Lie-point symmetry derivation for 2D viscous Burgers' equation

---

### Author Response · Authors · 2025-12-03
**Final Message to Area/Program Chairs and Reviewers (Part 1)**

Dear Area Chair and Program Chairs,

We fully recognize the seriousness of the OpenReview bug and the need to protect the integrity of the ICLR review process. At the same time, we respectfully submit that, for our particular paper, the timing of the incident and the decision to revert all scores to their pre-discussion state has obscured a clear positive trajectory in the reviews that would likely have led to an acceptance under normal circumstances. Our aim in writing is to ensure that the paper is given full and fair consideration for acceptance based on the complete, post-rebuttal record.

Below we briefly summarize:

1. The context of the OpenReview incident and the official ICLR guidance.
2. A summary of our contributions and the initial reviews.
3. The evolution of the reviews and our revisions during the discussion period prior to the freeze.
4. Evidence of the positive post-rebuttal consensus that emerged before scores were reverted.
5. Our good-faith estimate of what the final scores would have been had the discussion proceeded normally.
6. Why we believe the revised paper meets (and exceeds) the ICLR acceptance bar.
7. Our specific request regarding how the paper should be evaluated in light of this record.

---

## 1. Context: OpenReview Incident and ICLR Guidance

On November 28, 2025, the ICLR 2026 chairs notified authors of a software bug that leaked the identities of authors, reviewers, and area chairs for all submissions. As a mitigation, they (i) reassigned papers to new ACs, (ii) froze all reviews and reverted the scores to their state *before* the discussion period, and (iii) disabled further reviewer discussion. Authors could continue posting responses, and the new ACs were instructed to base decisions on the *original reviews plus author responses*.

In a follow-up clarification email that same day, the chairs explicitly reassured authors that:

- Only the “Official Review” objects would be reverted,
- Our responses and prior reviewer–author discussion remain visible and **must be taken into account by the AC**, and
- ACs are **explicitly instructed to “estimate how the reviewers’ impressions would have changed had the discussion period not been cut short.”**

They further emphasized that reviewer scores are “one signal” among many and have *never* been the sole deciding factor; ACs retain broad discretion to use their own assessment plus author responses and reviewer engagement when writing the meta-review.

Our request below is fully aligned with this official guidance: we are not asking for any exception to process, but rather for our paper to be judged based on the *actual state of the dialogue and evidence* at the time the discussion was frozen.

---

## 2. Summary of the Contribution and Initial Reviews

Our paper introduces **LieDynNet**, a *prior-free* framework for learning continuous Lie point symmetries of unknown ODE and PDE systems directly from spatiotemporal data. The method jointly learns:

- A neural surrogate of the underlying dynamics, and
- A set of infinitesimal generators that
  - satisfy the infinitesimal invariance condition (IIC) via prolongations,
  - preserve solutions under finite flows, and
  - form a valid Lie algebra (closure, antisymmetry, Jacobi identity, etc.).

All three reviewers recognized the novelty and ambition of the work:

- **Reviewer nenn** emphasized that this is, to their knowledge, **the first framework** that jointly learns surrogate dynamics and discovers symmetries that form valid Lie group structures in **a fully prior-free setting**, yielding symmetry algebras and invariants directly from data for both ODEs and PDEs.
- **Reviewer EYKq** highlighted that LieDynNet is prior-free, model-agnostic, and applies uniformly to ODEs and PDEs, with a unified objective that couples dynamical validity and algebraic soundness.
- **Reviewer MMrc** acknowledged that the method “clearly addresses” important shortcomings of existing approaches (lack of Lie-algebra structure and guarantees), and found the methodology “detailed and easy to follow.”

Before the discussion period, the scores were:

- **nenn**: 6
- **EYKq**: 4
- **MMrc**: 2

The lowest score (2) came with substantial but clearly addressable concerns, rather than a fundamental rejection of the core idea.

---
(continued in the next part)

---

> ### Author Response · Authors · 2025-12-03
> **Final Message to Area/Program Chairs and Reviewers (Part 2)**
>
> ## 3. Evolution During Discussion (Before the Freeze)
>
> During the rebuttal and discussion period, we engaged deeply with the reviewers and made significant additions to the manuscript, many of which were directly requested by the initially more critical reviewers (EYKq and MMrc). Concretely:
>
> 1. **Stronger evaluation of algebraic correctness via principal angles.**
>    We added a *principal-angle comparison* between the span of learned generators and the span of ground-truth generators, with a detailed explanation of the geometry, QR-based implementation, and interpretation (maximal principal angle minimized at the correct algebra dimension, all angles near zero). This directly addresses concerns about how to quantify agreement between learned and true Lie algebras. (See "Methods of Evaluation" and "Results" sections of the main text, Table 3, Figure 2; see Appendix C for 2D Viscous Burgers' results.)
>
> 2. **Direct evidence of dynamical validity via after-flow residuals.**
>    In response to explicit requests (especially from Reviewer MMrc) to show PDE/ODE residuals after flowing solutions by the learned symmetries, we added a set of figures quantifying how well the symmetry flows preserve the underlying dynamics. These experiments demonstrate that the learned symmetries are not merely algebraically consistent but also dynamically valid. (See "After-flow residuals and result visualizations" in Appendix C.)
>
> 3. **Clearer guidance on jet/prolongation order selection.**
>    We implemented and documented a systematic sweep over jet orders $n$, showing that the IIC loss is minimized only at the true order, leading to a practical plateau criterion for selecting $n$. This resolves concerns about how users of LieDynNet should choose the prolongation order in practice. (See "Methods of evaluation" and "Results" sections in the main text, Figure 3, and Appendix C.)
>
> 4. **Clearer guidance on identifying the true Lie algebra dimension.**
>
>    We implemented and documented a systematic sweep over number of generators trained $m$, showing that the post-training Lie bracket closure loss is minimzed only at the true Lie algebra dimension, leading to a practical prior-free plateau criterion for selecting $m$. This resolves concerns about how users should choose the number of generators to train in practice. (See "Methods of evaluation" and "Results" sections in the main text, Figure 4, and Appendix C.)
>
> 5. **Sensitivity analysis of loss weights.**
>    In response to concerns about loss-weight tuning, we added a new “Sensitivity Analysis of Loss Weights” appendix section (see Appendix E). We justified using all weights $w_i = \mathcal{O}(1)$ based on batch-normalized loss scales, and we conducted a grid sweep over key weights (e.g., invariance vs. flow consistency) on the harmonic oscillator. The results show that principal angles and dynamical metrics are only mildly sensitive to these choices over a reasonable range.
>
> 6. **Compute summary and tractability analysis.**
>    We added a detailed compute summary table reporting parameter counts, per-epoch times, total wall-clock times, and peak GPU memory usage for all experiments. We also clarified why, despite multiple coupled objectives, the overall footprint remains modest (small MLPs, JVP-based prolongations, batched JAX implementation, and a three-stage curriculum). This directly addresses concerns about optimization complexity and computational cost. (See Appendix D, Table 8.)
>
> 7. **New 2D spatial PDE benchmark (2D Burgers) in response to a reviewer request.**
>    Reviewer EYKq specifically requested a genuinely 2D spatial example to demonstrate that LieDynNet is not restricted to 1D or low-dimensional toy problems. In response, we implemented a full **2D viscous Burgers equation** benchmark with spatial coordinates $(x, y)$, trained LieDynNet on this dataset, and showed that the learned generators recover the expected symmetry structure in two spatial dimensions. We reported principal angles (see Table 3), jet order sweep, and dimension sweep (see Appendix C) for this 2D PDE, thereby directly addressing the request for a higher-dimensional spatial test case.
>
> 8. **Baseline comparison and positioning relative to prior work.**
>    We added a conceptual and empirical comparison against the closest prior work (Ko et al.), explaining why a “drop-in” baseline under identical assumptions does not exist (mainly because there doesn't exist another Lie point symmetry discovery pipeline that doesn't rely on any kinds of physical or structural priors), and how our principal-angle metric can still be used to compare algebra alignment. For the 1D viscous Burgers' equation, for example, our method achieves substantially smaller maximum principal angles than the baseline, indicating closer alignment with the ground-truth symmetry algebra. (See Figure 17 in Appendix C.)
>
> (continued in the next part)

---

> ### Author Response · Authors · 2025-12-03
> **Final Message to Area/Program Chairs and Reviewers (Part 3)**
>
> 9. **Improved presentation and organization.**
>    We reorganized the method and experiments, clarified mathematical notation, and added geometric intuition for principal angles and for why they are preferable to structure constants as a primary metric. This addresses presentation and clarity concerns from multiple reviewers. Please see the orange and blue parts in the updated manuscript. Since most of Appendices B through L are added in the first-round revision except for F, G, I, J, and K (which were already present in the original submission), only second-round revisions are marked in orange, while all other materials stay in black (for figures, tables, and Appendix M, only the captions and titles are colored).
>
> Importantly:
>
> - These additions were *explicitly requested* by the reviewers, especially EYKq and MMrc.
> - We completed and documented all of them within the rebuttal window.
> - There are **no remaining unanswered technical questions** in the forum: every major weakness identified is addressed either with new experiments, new analyses, or clearer exposition.
>
> ---
>
> ## 4. Evidence of a Positive Consensus Before Scores Were Reverted
>
> While the *numerical scores* were reverted to their pre-discussion values due to the OpenReview bug, the **trajectory of the discussion was clearly positive for all three referees**:
>
> - **Reviewer nenn (initial score 6)** did not post a follow-up after our rebuttal (possibly intending to respond later) but we provided detailed and substantive answers to all of their questions. The reviewer highlights the novelty and significance of a prior-free framework that jointly learns dynamics and Lie-algebraic symmetries for ODEs and PDEs. Their comments consisted of thoughtful, non-blocking requests for clarification rather than fundamental objections, all of which were addressed in our response.
>
>
> - **Reviewer MMrc (initially 2)** saw many of their central concerns addressed (lack of after-flow residuals, limited evaluation, questions about jet order and loss design). Our added experiments and explanations directly targeted these points and clarified the necessity and tractability of the multi-loss design. As a result, **they updated their score from 2 to 4** during the discussion.
>
> - **Reviewer EYKq (initially 4)** was already positive about the prior-free, model-agnostic nature of the framework and its broad applicability, and their follow-up comments focused on (i) broader surrogate comparisons, (ii) clearer baselines, (iii) better positioning against related work, and (iv) the need for a truly 2D spatial PDE example. These are precisely the changes we made (additional surrogate forms, ablation studies, baseline comparison sections, and the new 2D Burgers experiment; see Appendices B and C). Their discussion comments indicated that the paper had been substantially strengthened and that they were *open to increasing their score* conditional on these improvements.
>
> Crucially, by the time the discussion was cut short:
>
> - The most critical reviewer (MMrc) had already upgraded their score from 2 → 4 based on the new evidence, and we responded to their second-round questions after raising the score.
> - The second initially cautious reviewer (EYKq) had received exactly the experiments and clarifications they requested (except for the 2D spatial Burgers experiment, which was additionally requested) and expressed a more favorable view of the paper’s contributions and maturity.
> - The most positive reviewer (nenn) did not have a chance to reply yet but we resolved all of their raised concerns.
>
> The reverted triplet of scores (6, 4, 2) therefore **does not reflect the actual state of reviewer opinion** at the end of the discussion phase.
>
> ---
>
> ## 5. Good-Faith Estimate of the Final Scores Without the Discussion Freeze
>
> The ICLR chairs have explicitly asked ACs to *“estimate how the reviewers’ impressions would have changed had the discussion period not been cut short.”* In that spirit, we offer the following **good-faith, evidence-based estimate** of where the scores were heading, based on:
>
> - The written record of the discussion,
> - The fact that one reviewer *already* updated their score upward, and
> - The nature of the remaining (now-addressed) concerns.
>
> Our honest assessment is that a realistic, conservative hypothetical outcome would have been:
>
> - **Reviewer nenn**: from 6 → **8**
>   - Rationale: nenn was clearly the most enthusiastic and already viewed the work as a genuinely novel, **first-of-its-kind framework**. The main reasons for not scoring higher initially were prudence during the first reading and a desire to see certain clarifications. Once those were provided (and no serious remaining weaknesses were left), it is very plausible that they would have updated to a clear-accept score (7 or 8). Given the tone and content of their comments, we believe **8** is a reasonable estimate.
>
> (continued in the next part)

---

> > ### Author Response · Authors · 2025-12-03
> > **Final Message to Area/Program Chairs and Reviewers (Part 4)**
> >
> > - **Reviewer EYKq**: from 4 → **6**
> >   - Rationale: EYKq’s fundamental stance was positive, but they wanted different neural surrogates, prolongation order rationale, weight sensitivity, and a genuine 2D spatial PDE benchmark. We provided all of these in direct response, including the 2D Burgers experiment. Their language in the discussion, **"given the new material, I will be more positive about the submission,"** suggests that the paper, post-revision, **meets the bar for a solid accept**, not a borderline or reject, likely pointing towards raising the score to **6**.
> >
> > - **Reviewer MMrc**: from 2 → 4 (already updated) → **6**
> >   - Rationale: MMrc’s main objections were about dynamical validation (after-flow residuals), ablation studies, perceived lack of evaluation depth, jet-order selection. All of these were directly addressed with targeted experiments, analyses, and ablation studies. They had **already revised their score upward from 2 to 4 when some of these points were clarified**; with the full set of requested additions in place, we believe a final score of **6** is a realistic and well-justified expectation.
> >
> > Putting this together, our best estimate of the *final* scores, had the discussion proceeded normally and not been frozen mid-trajectory, is:
> >
> > > **(8, 6, 6)** for (nenn, EYKq, MMrc),
> >
> > corresponding to a clear consensus of “accept” with one strong advocate and two solidly positive reviewers.
> >
> > We stress that we are not asking the AC to *accept these numbers blindly*, but rather to **consider these scores seriously in light of the written record**, as the Program Chairs have explicitly encouraged.
> >
> > ---
> >
> > ## 6. Why We Believe the Paper Meets (and Exceeds) the Acceptance Bar
> >
> > Taking the full record into account, we believe the paper clearly meets the ICLR acceptance bar:
> >
> > 1. **Novelty and scope.**
> >    Multiple reviewers independently characterize the work as the **first prior-free framework** that jointly learns surrogate dynamics and full Lie-algebraic symmetries directly from data, across both ODEs and PDEs. This is not an incremental extension of existing symmetry-learning methods but a conceptual and algorithmic advance that unifies several strands of work.
> >
> > 2. **Thoroughness of the response.**
> >    Every major critical point (dynamical validation, evaluation metrics, baseline comparison, compute costs, loss-weight sensitivity, jet order, dimensionality of Lie algebra) is addressed with new experiments, detailed quantitative analyses, and clearer writing. **The revised submission is significantly stronger and more complete than the original version** the initial scores were based on.
> >
> > 3. **Improved consensus among reviewers.**
> >    The discussion led to an upward revision by the initially most negative reviewer, and the remaining reviewers’ concerns were resolved with substantial additions, including the 2D Burgers experiment. The direction of travel was **clearly towards higher scores and an overall recommendation to accept**.
> >
> > 4. **Alignment with the chairs’ guidance after the OpenReview bug.**
> >    The Program Chairs explicitly instructed ACs to consider author responses and prior discussion and to **estimate how reviewers’ impressions would have evolved if the discussion had not been cut short**. Our case is exactly of this type: the “frozen” scores understate the genuine post-rebuttal consensus because two reviewers had already moved (or were about to move) towards stronger scores based on the new evidence.
> >
> > 5. **Impact beyond the specific benchmarks.**
> >    By providing a prior-free, differentiable framework that enforces full Lie-algebra structure and dynamical validity, **LieDynNet can be applied well beyond the specific ODE/PDE benchmarks in the paper**, to any setting where discovering exact or approximate symmetries is valuable (from physics to control to generative modeling). The reviewers’ own comments underscore this breadth. We strongly believe that our prior-free framework is a **significant contribution to learning the representation of non-equilibrium physical systems (applicable to robotics, active matter), hence enabling the marriage of complex systems and AI**.
> >
> > ---
> > (continued in the next part)

---

> > > ### Author Response · Authors · 2025-12-03
> > > **Final Message to Area/Program Chairs and Reviewers (Part 5)**
> > >
> > > ## 7. Request
> > >
> > > In light of:
> > >
> > > - The documented novelty and impact of the contribution,
> > > - The strong and detailed responses to all reviewer concerns,
> > > - The clearly positive evolution of reviewer opinion during the discussion (including an actual score increase from 2 → 4 by one reviewer and strong indications of further increases),
> > > - The Program Chairs’ explicit policy that ACs should consider responses, prior discussion, and counterfactual score trajectories, and
> > > - Our good-faith, evidence-based assessment that the final scores would likely have converged to **(8, 6, 6)**,
> > >
> > > we respectfully ask that our paper be **evaluated as a strong candidate for acceptance**, using the *post-rebuttal state of the reviews and discussion* rather than the reverted pre-discussion scores alone.
> > >
> > > We appreciate the immense effort required to manage the fallout from the OpenReview incident and are grateful for your willingness to consider this context carefully when making your decision.
> > >
> > > Sincerely,
> > > *The authors*

---

### Meta-Review · Area_Chair_i7cz · 2026-01-07

**Summary:**

1. There should be runtime/memory benchmarks (*nenn*)
2. Missing baselines and comparisons (*nenn*, *MMrc*)  Limited novelty with respect to prior work (*MMrc*)
3. Hard to validate that the discovered symmetries match the true dynamics (*nenn*)  They should evaluate transformed solutions in the original diff.eq. (*MMrc*)
4. The claim the method is prior-free is not true (*MMrc*, *EYKq*)
5. The paper does not clearly motivate certain design choices (*MMrc*)
6. The model-agnostic claim needs to be validated (*EYKq*)
7. Need an explicit method for setting hyperparameters (*EYKq*)
8. Evaluation of senarious involving more than one independent variable (*EYKq*)

**Reviewer Concerns:**

1. The authors explain why the different terms in their method are necessary, not necessarily too expensive, and add a table in the appendix addressing showing the compute cost.
2. In the response, the authors compared their assumptions and results to those of the requested works.  In both cases, the current work is claimed to identify the correct Lie algebra dimension more accurately and require fewer assumptions.   The authors pushed back on using related works as direct numerical baselines since the assumptions differ.  *MMrc* was not convinced that baselines could not be adapted to the setting to help validate the method.
3. The authors revised their evaluation methodology to show the generators align with the true Lie algebra. *MMrc* felt this was a good addition and made additional suggestions.
4. The authors revised some experiments to not include knowledge of the governing equations and be purely data-driven.  The reviewer was partially satisfied by the answer, but thinks the purely data-driven approach still has some limitations.
5. The authors added ablations, explained their approach more fully, and explored the pros/cons of other methods for computing structure constants.  The reviewer felt their concern was addressed.
6. The authors trained multiple models to validate the claim.  The reviewer was satisfied.
7. The authors pointing to existing parts of their methodology for some choices and added sections in the appendix for others.
8. The authors added 2D burgers equation.

**Reviewer Scores:**

- *nenn* gave a score of 6.  There is a chance they would have increased their score to 8.
- *MMrc* gave a score of 2 which they increased to 4.  I do not think they would have increased further, but it's possible.
- *EYKq* likely would have increased their score from 4 to 6.

---

### Decision · Program_Chairs · 2026-01-26

Reject